# Advancing Universal Deep Learning for Electronic-Structure Hamiltonian Prediction of Materials

**Shi Yin**[1]*†, **Zujian Dai**[1]*, **Xinyang Pan**[2]*, **Lixin He**[2,1,3]†

[1] Institute of Artificial Intelligence, Hefei Comprehensive National Science Center
[2] Laboratory of Quantum Information, University of Science and Technology of China
[3] Hefei National Laboratory, University of Science and Technology of China

https://github.com/DavidYin94/NextHAM

## Abstract

Deep learning methods for electronic-structure Hamiltonian prediction have offered significant computational efficiency advantages over traditional density functional theory (DFT), yet the diversity of atomic types, structural patterns, and the high-dimensional complexity of Hamiltonians pose substantial challenges to the generalization performance. In this work, we contribute on both the methodology and dataset sides to advance universal deep learning paradigm for Hamiltonian prediction. On the method side, we propose **NextHAM**, a **n**eural **E**(3)-symmetry and e**x**pressive correc**t**ion method for efficient and generalizable materials electronic-structure **Ham**iltonian prediction. First, we introduce the zeroth-step Hamiltonians, which can be efficiently constructed by the initial charge density of DFT, as informative input descriptors that enable the model to effectively capture prior knowledge of electronic structures. Second, we present a neural Transformer architecture with strict E(3)-symmetry and high non-linear expressiveness for Hamiltonian prediction. Third, we propose a novel training objective to ensure the accuracy performance of Hamiltonians in both real space and reciprocal space, preventing error amplification and the occurrence of "ghost states" caused by the large condition number of the overlap matrix. On the dataset side, we curate a broad-coverage large benchmark, namely **Materials-HAM-SOC**, comprising $17,000$ material structures spanning more than $60$ elements from six rows of the periodic table and explicitly incorporating spin–orbit coupling (SOC) effects, providing high-quality data resources for training and evaluation. Comprehensive experimental results demonstrate that NextHAM achieves excellent accuracy in predicting Hamiltonians and band structures, with spin-off-diagonal blocks reaching the accuracy of sub-$\mu$eV scale. These results establish NextHAM as a universal and highly accurate deep learning model for electronic-structure prediction, delivering DFT-level precision with dramatically improved computational efficiency.

## 1 Introduction

Understanding the electronic structure is fundamental to unraveling how electrons govern the properties of condensed matter systems. This knowledge is essential for predicting a wide range of material characteristics, such as electrical conductivity, magnetism, optical behavior, and chemical activity, which are vital for technologies spanning from electronics to sustainable energy and advanced catalysis. At the heart of these calculations is the challenge of determining the system's Hamiltonian matrix, whose eigenvalues and eigenstates yield important quantities like energy levels, band structures, and electronic wavefunctions. Traditionally, Density Functional Theory (DFT) (Hohenberg & Kohn, 1964; Kohn & Sham, 1965) has been the standard approach for these problems.

---

*Equal contributions.
†Corresponding authors. Emails: shiyin@iai.ustc.edu.cn, helx@ustc.edu.cn

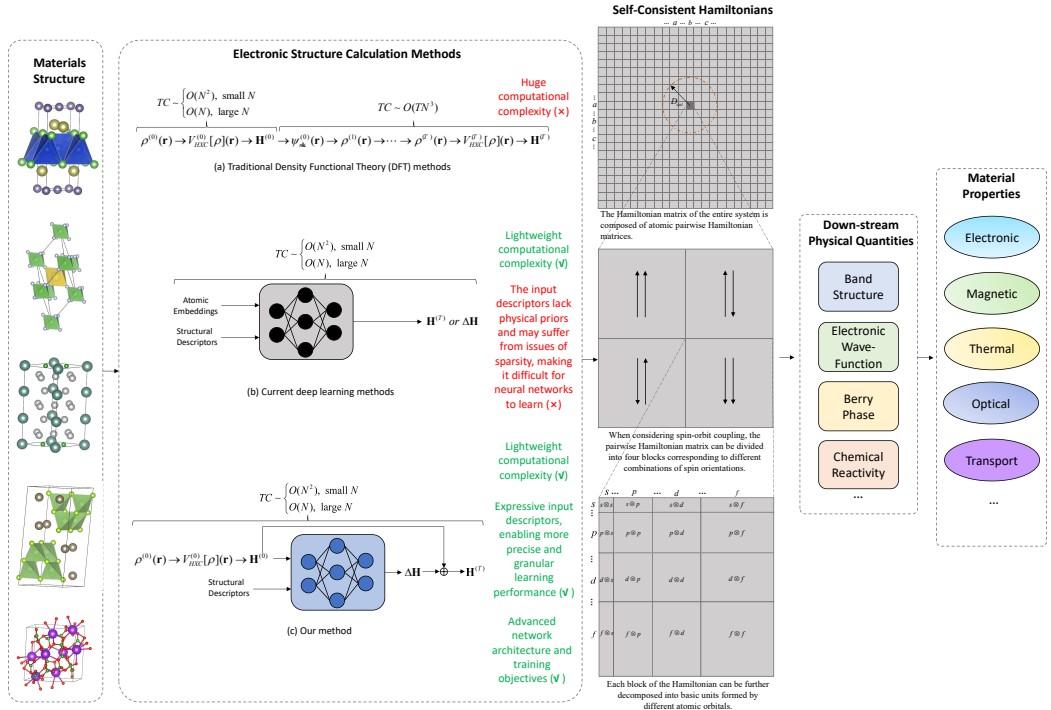

Figure 1: Comparison of paradigms for electronic-structure Hamiltonian calculation, highlighting the fundamental differences between our method and both classical DFT methods and existing deep learning approaches.

However, as shown in Fig. 1 (a), DFT relies heavily on the self-consistent (SC) procedure, which demands repeated (denoted as $T$ turns), computationally intensive diagonalizations of large matrices, each scaling as $\mathcal{O}(N^3)$ with system size $N$, making simulations of large or complex materials extremely resource-consuming. Recently, deep learning has emerged as a powerful tool in the physical sciences (Zhang et al., 2025). As shown in Fig. 1 (b), modern deep neural network methods (Gong et al., 2023; Yu et al., 2023b; Zhang et al., 2024; Wang et al., 2024c; Li et al., 2025; Yin et al., 2025) can predict Hamiltonians, i.e., the core physical quantities in electronic structure calculations, directly from atomic configurations in an efficient way, circumventing the computationally expensive SC loop and dramatically accelerating computations. This paradigm shift lowers the computational barriers associated with electronic structure calculations, unlocking the simulation and design of unprecedentedly large-scale materials systems, driving new innovation in materials discovery and engineering. Please refer to Appendix A for background introduction.

However, deep learning methods still face substantial challenges in achieving accurate and generalizable Hamiltonian prediction, primarily due to the extremely complex and fundamentally difficult nature of the input–output mapping that the neural network must learn, making it difficult to generalize across diverse material systems. Consequently, it has become common practice to constrain the scope, such as limiting the range of supported elements, neglecting spin–orbit coupling (SOC) effects, or reducing the number of orbitals considered, as thoroughly discussed in Appendix C. While such strategies help alleviate modeling burdens, they also restrict the applicability of these methods to the full diversity and complexity of real-world materials. What's more, large open-source materials datasets for the training and evaluation of general Hamiltonian learning models are also rare.

To solve these challenges, in this work, we make contributions on both methodology and benchmark toward advancing universal deep learning for electronic-structure Hamiltonian prediction of materials. On the method side, as shown in Fig. 1 (c), we propose **NextHAM**, a **n**eural **E**(3)-symmetry and e**x**pressive correc**t**ion framework for efficient and accurate Hamiltonian prediction:

First, we dive deeply into the traditional DFT computational process outlined in Appendix A and introduce a physical quantity that helps mitigate the complexity of the input–output mapping encountered by deep learning models for Hamiltonian prediction. This quantity is the zeroth-step Hamiltonian $\mathbf{H}^{(0)}$, which is efficiently constructed from the initial electron density $\rho^{(0)}(\mathbf{r})$, given by the sum of the charge densities of isolated atoms, without the requirement of matrix diagonalization. As $\mathbf{H}^{(0)}$ efficiently encodes essential information about the system's electronic structure, we innovatively incorporate it as one of the input features to the neural network. Unlike existing methods that rely on randomly initialized atom and edge embeddings, which lack physical prior knowledge and may suffer from issues of sparsity as analyzed in Appendix M, $\mathbf{H}^{(0)}$ provides richer physical context by embedding the intrinsic characteristics of diverse elements into a unified representation space, thereby enabling robust generalization across chemically complex material systems. Moreover, inspired by the delta-learning paradigm (Bowman et al., 2022), we predict the correction term $\Delta\mathbf{H} = \mathbf{H}^{(T)} - \mathbf{H}^{(0)}$ rather than directly predicting the entire $\mathbf{H}^{(T)}$, reducing both the dimensionality and numerical range of the regression target, and enabling the model to focus on capturing only the essential differences rather than reconstructing the entire Hamiltonian from scratch.

Second, we present a network architecture that strictly adheres to E(3)-symmetry while maintaining high non-linear expressiveness for Hamiltonian prediction by extending the TraceGrad (Yin et al., 2025) method to Transformer framework, thereby providing ample capacity for flexible and accurate modeling of atomic systems for Hamiltonian prediction across a wide range of elements in the periodic table. Furthermore, we introduce model ensemble techniques to enhance the capacity of the framework for handling complex scenarios in Hamiltonian prediction.

Third, we propose a joint optimization framework that simultaneously refines both real-space (R-space) and reciprocal-space (k-space) Hamiltonians. Most existing methods regress only the real-space Hamiltonian, but the large condition number of the overlap matrix can amplify errors in predicted eigenvalues and eigenfunctions, leading to suboptimal physical fidelity. Although recent work (Li et al., 2025) has explored strategies to mitigate this error amplification, their attempts are tailored to finite molecular systems, and also overlook the inherent gauge freedom in Hamiltonian representations (Wang et al., 2024c). In contrast, our method explicitly targets the decoupling of energy subspaces in k-space for infinite periodic systems to eliminate spurious "ghost states" and strictly preserve the band topology. Furthermore, we resolve the gauge ambiguity by analytically determining the optimal gauge parameter within our joint optimization framework, thereby stabilizing the optimization landscape and ensuring unique, physically consistent predictions.

On the dataset side, we curate a diverse-collection large benchmark dataset, **Materials-HAM-SOC**, containing $17,000$ material structures generated using DFT softwares. The dataset spans more than $60$ elements from the first six rows of the periodic table and explicitly incorporates spin–orbit coupling (SOC) effects. To ensure the accuracy of the DFT calculations, we employ high-quality pseudopotentials that include as many valence electrons as possible, enabling our model to handle physically complex and highly challenging systems. We adopt high-quality atomic orbital basis sets, up to 4s2p2d1f orbitals for each element, to ensure fine-grained description of electronic structures. This dataset establishes a challenging yet comprehensive benchmark for evaluating generalization across chemically and structurally diverse systems.

Extensive experiments on the Materials-HAM-SOC dataset demonstrate that NextHAM achieves a prediction error of 1.417 meV across full Hamiltonian matrices in R-space, with spin-off-diagonal (SOC) blocks suppressed to the sub-$\mu$eV scale. Moreover, the band structures derived from k-space Hamiltonian exhibit excellent agreement with first-principles DFT. Furthermore, our method offers a substantial computational advantage over traditional DFT. These results establish a new paradigm for electronic-structure calculations, combining high accuracy, broad generalization capability, and significant computational efficiency. Besides general tasks, our method also achieves state-of-the-art performance in more specialized scenarios from the databases of DeepH Series Li et al. (2022); Gong et al. (2023). These breakthroughs open new avenues for practical applications, including rapid screening of candidate materials, modeling of nano-structures, and simulation of large-scale quantum devices.

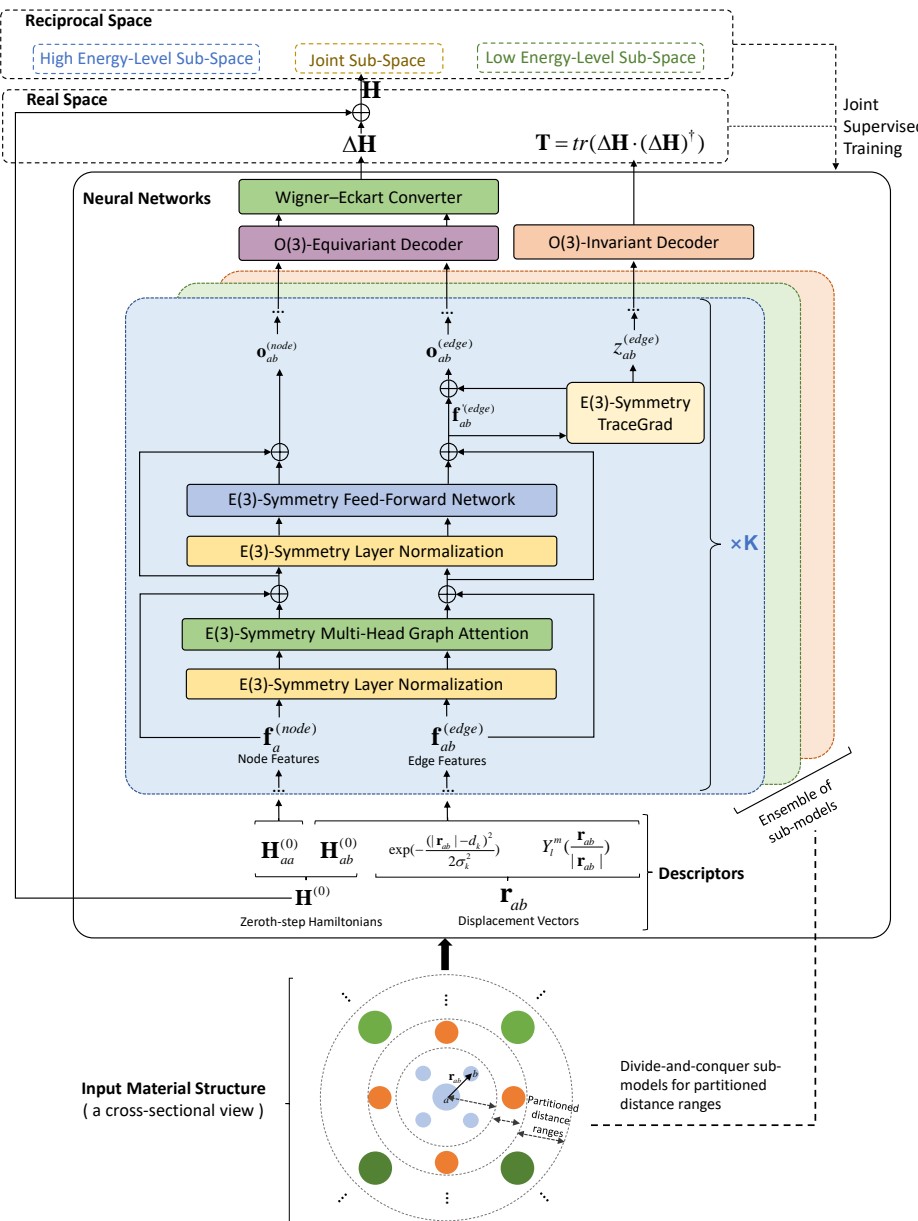

Figure 2: Illustration of the proposed NextHAM framework.

## 2 METHOD

As shown in Fig. 2, to effectively handle wider chemical and structural variability of materials, we develop a unified Hamiltonian prediction framework along three aspects:

### 2.1 INPUT DESCRIPTORS

As shown at the lower part of Fig. 2, we use the displacement vector-based descriptors between atoms that lie within the cutoff distance, together with the zeroth-step Hamiltonian (calculated as detailed in Appendix A), as the input features for the neural network. Introducing the zeroth-step Hamiltonian as the input features for the neural network is one of the core innovations of our framework. The zeroth-step Hamiltonian $\mathbf{H}^{(0)}$, derived from the initial charge density $\rho^{(0)}(\mathbf{r})$, obtained

as the sum of neutral atomic charges, reflects the information of different elements in the system, including the strength of the electron-ion interactions (pseudopotential) and a preliminary estimate of the electron-electron interactions. These components directly influence the system's electronic structure. By embedding $\mathbf{H}^{(0)}$ as the inputs, our method encodes the characteristics of different elements into a unified representation space, bringing a powerful capability to generalize across diverse material systems.

The zeroth-step Hamiltonian $\mathbf{H}^{(0)}$ can be decomposed into on-site sub-matrices, which represent the Hamiltonian blocks corresponding to each atom and its own orbitals, and off-site sub-matrices, which capture the interactions between different atoms. These two types of sub-matrices can naturally serve as the initial descriptors for nodes and edges, respectively, in the graph neural network, as detailed in Appendix E.

What's more, as detailed in Appendix A, computing the zeroth-step Hamiltonian requires no matrix diagonalization, so its cost scales with the number of non-zero matrix elements: it is approximately $\mathcal{O}(N^2)$ for small systems with $N$ atoms and asymptotically approaches $\mathcal{O}(N)$ for sufficiently large systems as the neighbor count saturates for each atom. This matches the scaling behavior of the message passing mechanism of graph neural networks, ensuring that incorporating the zeroth-step Hamiltonian as a new input descriptor does not worsen the $\mathcal{O}$-asymptotics.

## 2.2 Neural Network Architecture

Accurate Hamiltonian prediction requires the neural network to strictly adhere to the symmetries of the E(3) group. While translation symmetry can be easily implemented using relative coordinates, maintaining O(3)-equivariance while also achieving significant expressive power presents a challenging and fundamental problem. To provide the necessary background on the directly relevant basic concepts, we refer the reader to Appendix B-D.

We present a Transformer architecture that not only maintains strict E(3)-symmetry but also achieves strong non-linear expressiveness, as shown in Fig. 2. Our E(3)-symmetry graph attention mechanism is developed from Equiformer (Liao & Smidt, 2023). While Equiformer was designed for regression tasks where the target quantity is essentially a node-level atomic property (e.g., force fields), our Hamiltonian target is fundamentally an edge-level property defined on atomic pairs. This distinction necessitates stronger modeling of edge features and motivates the development of our attention mechanism. First, we explicitly maintain and update edge features across multiple layers, rather than generating them only temporarily from node features on demand (Liao & Smidt, 2023). In this way, the computation of attention weights incorporates both the node features and the persistently maintained edge features. Second, motivated by the decay behavior of Hamiltonian matrix elements with respect to interatomic distance, we explicitly incorporate interatomic distances by introducing distance embeddings as additional signals in the computation of attention weights, enabling the model to better exploit distance information for inference. Third, the attention weights between nodes are directly applied to update edge features via multiplicative operations, and are subsequently refined through equivariant transformations. Together, these developments substantially enhance the capacity of the model to represent edge features, from which the Hamiltonian is regressed.

As analyzed in Section C, the TraceGrad mechanism (Yin et al., 2025) can maintain strong non-linear expressiveness while preserving strict E(3)-symmetry [1]. We extend TraceGrad into the Transformer framework for electronic-structure Hamiltonian prediction. As shown in the middle of Fig. 2, for an atomic pair $(a, b)$, the updated O(3)-equivariant edge feature $\mathbf{f}_{ab}^{'(\text{edge})}$ is further fed into the TraceGrad module to produce the non-linear O(3)-invariant feature $z_{ab}^{(\text{edge})}$, which is subsequently passed to the O(3)-invariant decoder and trained under the supervision of the O(3)-invariant trace quantity $\mathbf{T} = \text{tr}(\Delta \mathbf{H} \cdot \Delta \mathbf{H}^\dagger)$. The learned non-linear expressiveness in $z_{ab}^{(\text{edge})}$ is subsequently delivered into the equivariant feature by $\mathbf{o}_{ab}^{(\text{edge})} = \mathbf{f}_{ab}^{'(\text{edge})} + \frac{\partial z_{ab}^{(\text{edge})}}{\partial \mathbf{f}_{ab}^{'(\text{edge})}}$, where $\mathbf{o}_{ab}^{(\text{edge})}$ represents the non-linearity-enhanced O(3)-equivariant edge feature, which, together with the node feature, are fed into the subsequent encoding modules of the Transformer followed by the $O(3)$-equivariant decoder and a Wigner–Eckart converter (Gong et al., 2023) to regress the correction term $\Delta \mathbf{H} = \mathbf{H}^{(T)} - \mathbf{H}^{(0)}$.

---

[1] Although the original paper of TraceGrad emphasizes its SO(3)-equivariance, it is straightforward to prove that it also preserves E(3)-symmetry, including translation-invariance and O(3)-equivariance in this context.

This residual formulation defines the learning task as a delta-learning problem, reducing both the dimensionality and numerical range of the regression target.

To enhance model capacity and better capture the complex dependence of Hamiltonian matrix elements on diverse inter-atomic distances, we employ an ensemble learning strategy. Specifically, sub-models are trained to predict Hamiltonian sub-matrices corresponding to different distance intervals between atoms. Although each sub-model specializes in a specific range in the output stage, the input to each sub-model is the entire system, including the zeroth-step Hamiltonian and the displacement vectors for all atomic pairs, thereby effectively extracting global information. The final prediction is obtained by aggregating the outputs from all these sub-models.

## 2.3 TRAINING LOSS FUNCTIONS

The objective of training the neural network is to make the predicted Hamiltonian, denoted as $\widehat{\mathbf{H}} = \mathbf{H}^{(0)} + \widehat{\Delta\mathbf{H}}$, approximate the ground truth $\Delta\mathbf{H}^{gt}$ as closely as possible. As illustrated in Fig. 2, to ensure that the predicted Hamiltonian can accurately derive down-stream physical quantities (such as band structures), we design a joint optimization strategy in both real space (R-space) and reciprocal space (k-space) for the neural network. Crucially, as detailed in Appendix G, our entire loss formulation is designed to rigorously resolve the gauge ambiguity Wang et al. (2024c), ensuring the uniqueness and physical consistency of the regression targets.

In R-space, the Hamiltonian and the corresponding trace quantity are jointly supervised. As outlined in Section 2.2, the trace quantity is used to supervise the non-linear O(3)-invariant features, which contribute to constructing the non-linear O(3)-equivariant features required for predicting the Hamiltonian. The R-space training loss function is defined as:

$$\text{loss}(\mathbf{R}) = \mathbb{E}_{\mathbf{R}}\left[\lambda_R\left((1 - \lambda_C) \cdot \text{loss}_H(\mathbf{R}) + \gamma(\text{loss}_H, \text{loss}_T, \lambda_C) \cdot \text{loss}_T(\mathbf{R})\right)\right] \quad (1)$$

where $\mathbb{E}_{\mathbf{R}}[\cdot]$ denotes the empirical expectation, $\lambda_C$, $\lambda_R$ are hyper-parameters; $\mathbf{R}$ denotes the lattice vector connecting the reference unit cell and a neighboring unit cell; $\text{loss}_H(\mathbf{R})$ and $\text{loss}_T(\mathbf{R})$ denote the prediction losses of the Hamiltonian and the trace quantity in $\mathbf{R}$-space, respectively; and $\gamma(\text{loss}_H, \text{loss}_T, \lambda_C)$ is a scaling factor designed to balance their relative contributions for stable training. The detailed forms of these terms are provided in Appendix G.

As analyzed in Appendix F, due to the error amplification mechanism associated with the ill-conditioned overlap matrix, even small numerical errors in $R$-space can be magnified in $k$-space, leading to deviations in downstream physical quantities. To mitigate this, we introduce $k$-space loss functions. Specifically, the spectrum is partitioned into a low-energy subspace $P$, which governs most physical properties, and a high-energy complement $Q$. While downstream phenomena are predominantly determined by $P$, an inaccurately predicted Hamiltonian may introduce spurious couplings between $P$ and $Q$. This can result in unphysical abrupt changes in band structures, which are referred to as "ghost states" (see Fig. 10 in Appendix L). Therefore, it is essential not only to emphasize accuracy in $P$ but also to maintain reasonable fidelity in $Q$ so that the erroneous $PQ$ couplings can be identified and suppressed. To this end, we incorporate differentiated weights for $P$ and $Q$ in the loss design, together with an explicit $PQ$ penalty that eliminates unphysical cross-subspace couplings and suppresses ghost states.

The loss function in reciprocal space is defined as:

$$\text{loss}(\mathbf{k}) = \mathbb{E}_{\mathbf{k}}[\lambda_P \cdot \text{loss}_P(\mathbf{k}) + \lambda_Q \cdot \text{loss}_Q(\mathbf{k}) + \lambda_{PQ} \cdot \text{loss}_{PQ}(\mathbf{k})] \quad (2)$$

where $\lambda_P$, $\lambda_Q$, and $\lambda_{PQ}$ are tunable hyper-parameters that adjust the relative importance of the three loss terms, which respectively measure the errors in the $P$ subspace, the $Q$ subspace, and the combined $PQ$ joint subspace. The detailed formulations of these terms are provided in Appendix G.

The overall loss function combines the losses from both R-space and k-space:

$$\text{loss}_{all} = \text{loss}(\mathbf{R}) + \text{loss}(\mathbf{k}) \quad (3)$$

This consistent treatment of real-space and reciprocal space Hamiltonians provides a robust foundation for high-fidelity band structure predictions and, in particular, effectively eliminates ghost states.

## 3 DATASET

As broad-coverage open-source Hamiltonian datasets that use fine-grained orbital descriptions and include spin-orbit coupling (SOC) effects across a wide range of crystals are still rare, we construct one ourselves and contribute it to the community. Specifically, our dataset, called Materials-HAM-SOC, contains 17,000 material structures sampled from the Materials Project (Jain et al., 2013), with ground-truth Hamiltonians and band structures generated using the DFT software **ABACUS** (Li et al., 2016; Lin et al., 2023) and **PYATB** (Jin et al., 2023). It spans more than 60 distinct elements from the first six rows of the periodic table and explicitly incorporates SOC effects. For these structures, a high-quality atomic orbital basis set (Lin et al., 2021), up to 4s2p2d1f orbitals for each element, is employed, providing a fine-grained representation of their electronic structure. The dataset contains all quantities required by our method, including atomic structures, zeroth-step Hamiltonians, self-consistent Hamiltonians, and overlap matrices. The dataset is partitioned into 12,000 structures for training, 2,000 for validation, and 3,000 for testing. For details of the dataset construction and comprehensive statistical summaries, please refer to Section H.

## 4 EMPIRICAL STUDY

### 4.1 STATISTICAL RESULTS

We perform empirical studies on the Materials-HAM-SOC dataset. The implementation details of the network architecture and training configurations are provided in Appendix I.

First, to evaluate the role of $\mathbf{H}^{(0)}$ as an initial approximation at the output stage, we measure its discrepancy from the ground truth Hamiltonian $\mathbf{H}^{gt} = \mathbf{H}^{(T)}$. This quantifies how much $\mathbf{H}^{(0)}$ reduces the effective size and complexity of the regression target space for subsequent corrections. Second, we examine the final prediction accuracy by comparing $\mathbf{H}^{(0)} + \widehat{\Delta\mathbf{H}}$ with $\mathbf{H}^{(T)}$, thereby measuring the contribution of the learned correction $\widehat{\Delta\mathbf{H}}$ in closing the residual gap between $\mathbf{H}^{(0)}$ and $\mathbf{H}^{(T)}$. These two comparisons together disentangle the effectiveness of the prior $\mathbf{H}^{(0)}$ and the neural correction on achieving high-fidelity Hamiltonian predictions.

While mean absolute error (MAE) is a straightforward error metric, Hamiltonian prediction presents a unique gauge freedom: adding a global shift $\mu\mathbf{S}$, where $\mu$ is an arbitrary scalar and $\mathbf{S}$ is the overlap matrix, leaves all down-stream physical quantities unchanged (Wang et al., 2024c). This necessitates a gauge-invariant error metric for fair evaluation. To remove this gauge freedom, we adopt the Gauge MAE (Wang et al., 2024c) to our context:

$$
\begin{aligned}
\text{Gauge\_MAE}(\mathbf{H}^{(0)}, \mathbf{H}^{(T)}) &= \min_{\mu} \text{MAE}\big(\mathbf{H}^{(0)}, \mathbf{H}^{(T)} + \mu\mathbf{S}\big), \\
\text{Gauge\_MAE}(\mathbf{H}^{(0)} + \widehat{\Delta\mathbf{H}}, \mathbf{H}^{(T)}) &= \min_{\mu} \text{MAE}\big(\mathbf{H}^{(0)} + \widehat{\Delta\mathbf{H}}, \mathbf{H}^{(T)} + \mu\mathbf{S}\big),
\end{aligned}
\tag{4}
$$

where $\mu$ is determined by solving $\mu^* = \arg\min_{\mu} \text{Gauge\_MAE}$.

The experimental results for the above metrics are reported in Table 1. In addition, we also report $\text{Gauge\_MAE}(\mathbf{0}, \mathbf{H}^{(T)})$ for comparison. Comparing between $\text{Gauge\_MAE}(\mathbf{0}, \mathbf{H}^{(T)})$ and $\text{Gauge\_MAE}(\mathbf{H}^{(0)}, \mathbf{H}^{(T)})$ quantifies the actual reduction in the effective output space achieved by introducing $\mathbf{H}^{(0)}$ at the output stage.

As shown in Table 1, the zeroth-step Hamiltonian $\mathbf{H}^{(0)}$ closely matches the self-consistent Hamiltonian $\mathbf{H}^{(T)}$ in the spin-flip submatrices ($\uparrow\downarrow$ and $\downarrow\uparrow$). Similarly, the imaginary parts of the spin-conserving submatrices ($\uparrow\uparrow$ and $\downarrow\downarrow$) also exhibit excellent agreement. In these components, the deviation between $\mathbf{H}^{(0)}$ and $\mathbf{H}^{(T)}$ is negligible, with errors reaching sub-$\mu eV$ level. Achieving SOC predictions with an accuracy of $\mu eV$ level holds significant value, as small differences in SOC energy can greatly impact the electronic structure of materials (Jing et al., 2025).

Furthermore, the $\text{Gauge\_MAE}(\mathbf{H}^{(0)}, \mathbf{H}^{(T)})$ for the real part of the $\uparrow\uparrow$ block is reduced by 96% compared to $\text{Gauge\_MAE}(\mathbf{0}, \mathbf{H}^{(T)})$, yielding a much narrower numerical range for regression. This substantial reduction eases optimization by allowing the network to concentrate on physically meaningful residual corrections rather than reconstructing the entire Hamiltonian, thereby improving prediction accuracy across diverse atomic configurations. In systems with time-reversal symmetry and

Table 1: Comparison of Gauge MAE values computed in real space (R-space) on the testing set of Materials-HAM-SOC. Values are reported for four spin-resolved regions (↑↑, ↑↓, ↓↑, ↓↓) with separate real and imaginary components, and for the entire matrix (Overall), where real and imaginary components are combined into a single metric. Metrics are averaged over non-zero elements only; entries set to zero due to the truncation distance are masked out. All values are in meV.

| Region | Gauge_MAE($\mathbf{0}, \mathbf{H}^{(T)}$) | | Gauge_MAE($\mathbf{H}^{(0)}, \mathbf{H}^{(T)}$) | | Gauge_MAE($\mathbf{H}^{(0)} + \widehat{\mathbf{\Delta H}}, \mathbf{H}^{(T)}$) | |
|---|---|---|---|---|---|---|
| | Real | Imag | Real | Imag | Real | Imag |
| ↑↑ | 149.145 | 0.293 | 5.213 | < 0.001 | 2.834 | < 0.001 |
| ↑↓ | 0.301 | 0.299 | < 0.001 | < 0.001 | < 0.001 | < 0.001 |
| ↓↑ | 0.301 | 0.299 | < 0.001 | < 0.001 | < 0.001 | < 0.001 |
| ↓↓ | 149.145 | 0.293 | 5.213 | < 0.001 | 2.834 | < 0.001 |
| Overall | 74.914 | | 2.606 | | **1.417** | |

real-valued atomic orbitals, which constitute the majority of practical cases, the real parts of the ↑↑ and ↓↓ blocks are identical. This symmetry implies that the correction network only needs to predict the real part of the ↑↑ block in $\mathbf{\Delta H}$, substantially reducing the number of matrix elements to be learned. Finally, with the neural network correction applied, the errors for the ↑↑ and ↓↓ blocks are substantially reduced, achieving a superior prediction accuracy: the overall Gauge MAE is 1.417 meV, closely matching the ground-truth labels obtained from DFT calculations.

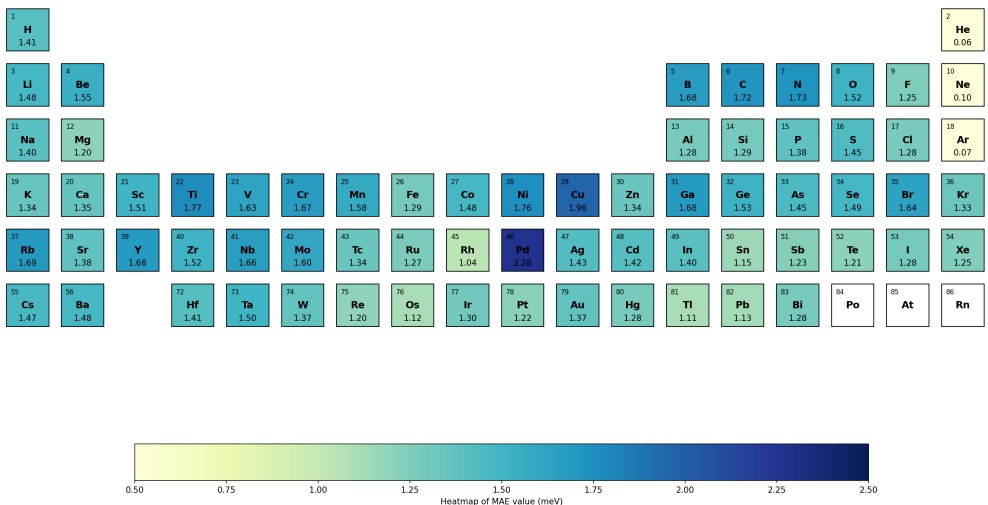

Figure 3: Element-wise analysis of prediction errors. For each chemical element, we collect all of the testing structures containing that element and compute the Gauge MAE values for each subset.

In Fig. 3, we report a fine-grained evaluation of prediction accuracy by partitioning the test set into subsets defined by chemical elements. For each element, we gather all crystal structures that contain it and compute the mean error within this subset. The resulting per-element statistics are visualized on the periodic table, providing a clear view of how the model generalizes across chemically diverse systems. The analysis shows that for most of the elements, the prediction errors are below 1.5 meV, confirming the robustness of our approach across a broad spectrum of the periodic table.

For a more detailed analysis of the contributions of different components in our framework, we conduct fine-grained ablation studies, which are detailed in Appendix L. These studies show that the physics-informed input descriptor $\mathbf{H}^{(0)}$, the correction-based regression target design, the Trace-Grad mechanism, the ensemble strategy, and the joint $R$- and $k$-space training objective each provide significant reductions in errors. The combination of all these components leads to the best overall performance, with notable improvements in both band structure prediction and the suppression of

unphysical artifacts such as ghost states. We also compare our method with DeepH-E3 (Gong et al., 2023) and the original work of TraceGrad (Yin et al., 2025), demonstrating the significant superiority of our method. For details, please refer to Appendix M.

## 4.2 CASE STUDY ON OUT-OF-DISTRIBUTION GENERALIZATION

As shown in Figure 7, our model is trained on a dataset that does not include structures containing the element Neon (Ne). However, in the testing set, a Ne-containing structure is included. Despite Ne being unseen during training, the model is able to generalize well to this new element, with testing error remaining very small, i.e., 0.1 meV for the R-space MAE, as reported in Figure 3. This demonstrates the model's ability to extrapolate knowledge learned from other elements to predict properties for unseen elements. This out-of-distribution generalization capability is a direct result of our approach using the zeroth-step Hamiltonian as a descriptor. Unlike traditional methods that rely on randomly initialized embeddings for each element, which cannot generalize to unseen elements not included in the training set, our model embeds physical information about the system's electronic structure into a unified representation space using the zeroth-step Hamiltonian. This helps the model capture the relationships between different elements' electronic structures, enabling it to generalize more effectively to out-of-distribution elements. This case study demonstrates the theoretical potential of our method's generalizability to unseen elements. In future work, we will conduct further element-level out-of-distribution evaluations to more rigorously quantify this ability.

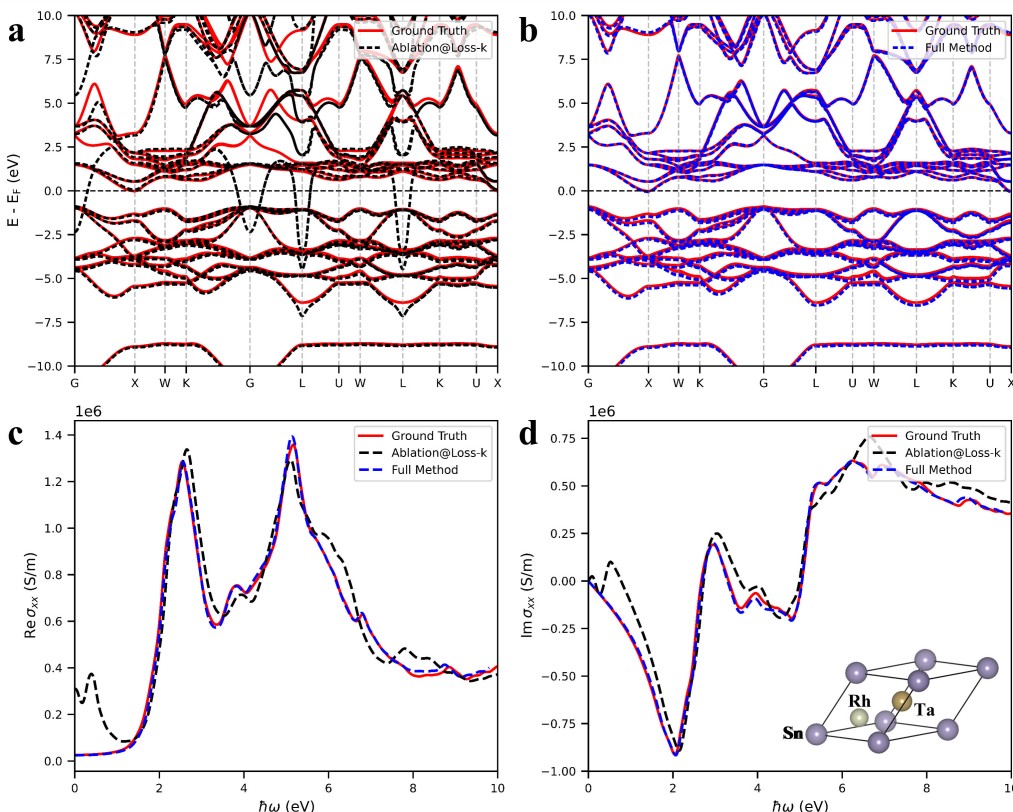

Figure 4: Panels a and b show the band-structure comparison among Ablation@Loss-k, the Full Method, and the Ground Truth DFT results. The setting Ablation@Loss-k drops the k-space loss from the Full Method. Panels c and d display the comparison of the real and imaginary parts of the optical conductivity along the x direction computed from the three Hamiltonians.

### 4.3 Case Study on $K$-space Loss Function

We present a representative case study regarding the effectiveness of the $k$-space loss function in the main text, with more cases available in Figure 10 of Appendix L. As shown in Fig. 4, in panel (a), the model is trained using only the $H(\mathbf{R})$-based loss defined in Eq. (1), ablating the $k$-space loss. In this case, the predicted band structure closely matches the DFT reference near the Fermi level for most $\mathbf{k}$-points. However, several isolated points exhibit abrupt discontinuities and deviate significantly from the ground-truth bands, which is a typical manifestation of ghost states. Notably, the MAE of the $H(\mathbf{R})$ matrix elements is only 0.53 meV. These sharp spectral anomalies show that even small MAE errors in real space ($H(\mathbf{R})$) can trigger ghost states in the resulting band structure.

This issue arises from the intrinsic numerical instability of the generalized eigenvalue problem, where the non-orthogonality of the orbital basis leads to pronounced error amplification. As we analyze in Appendix F, the sensitivity of both eigenvalues and eigenfunctions is enlarged by the factor $\frac{\kappa(S(\mathbf{k}))}{\|S(\mathbf{k})\|_2}$, implying that even small perturbations in the predicted $H(\mathbf{R})$ can cause significant deviations in the computed band energies, manifesting as ghost states in the spectrum.

In Fig. 4(b), we present the result from our Full Method, which includes the $k$-space loss. The predicted bands now align closely with the ground-truth results, and ghost states are nearly eliminated. The MAE of the real-space Hamiltonian $H(\mathbf{R})$ remains at $0.49$ meV, similar to the case where only $\text{loss}(\mathbf{R})$ was used, while the $k$-space loss $\text{loss}(\mathbf{k})$ is reduced by more than $50\%$, substantially enhancing the overall quality of the predicted band structure.

Beyond band energies, the fidelity of the Hamiltonian is also reflected in wave-function-related physical observables. To further assess the influence of different loss functions on the wave-function accuracy, we compute the optical conductivity corresponding to the predictions in Figs. 4(a) and (b). The real and imaginary parts of the conductivity along the x direction, compared with the ground-truth results, are presented in Figs. 4(c) and (d). As demonstrated in these comparisons, the Hamiltonian trained with our Full Method yields substantially improved agreement with the reference conductivity. This result indicates that incorporating $\text{loss}(k)$ not only suppresses ghost states under comparable $H(\mathbf{R})$ MAE, but also enhances the prediction of physical observables that depend sensitively on the wave-function quality.

## 5 Conclusion

We advance universal Hamiltonian deep learning through both a new method and a new dataset. We propose **NextHAM**, a unified deep learning framework designed for accurate and generalizable prediction of electronic-structure Hamiltonians across the periodic table. First, we leverage zeroth-step Hamiltonians constructed from initial charge densities as informative input features, facilitating the model to capture the intrinsic characteristics of electronic structures. Second, we present a Transformer-based neural architecture that enforces strict $E(3)$-equivariance while maintaining high expressive capacity, enabling accurate modeling of spatial symmetries in material systems. Third, we design a novel training objective that jointly optimizes the Hamiltonian prediction in both real space and reciprocal space, ensuring consistency with downstream physical quantities such as band structures. We also release **Materials-HAM-SOC**, a diverse-collection benchmark of 17,000 DFT-calculated material structures spanning six rows of the periodic table, with explicit spin–orbit coupling and high-resolution orbital representations, providing high-quality resources for training and evaluation. Empirically, NextHAM attains DFT-level accuracy for Hamiltonians and band structures while bringing substantial speedups over conventional DFT workflows, providing powerful tools to efficient simulation and design of new materials.

### Acknowledgements

This work was supported by the Advanced Materials–National Science and Technology Major Project (Grant No. 2025ZD0618401), the National Natural Science Foundation of China (Grant No. 12134012, 62506112), the Strategic Priority Research Program of the Chinese Academy of Sciences (Grant No. XDB0500201), and the Innovation Program for Quantum Science and Technology (Grant No. 2021ZD0301200). The numerical calculations were performed on the USTC High-Performance Computing facilities and Hefei advanced computing center.

## REPRODUCIBILITY STATEMENT

The codes of **NextHAM** are available here: `https://github.com/DavidYin94/NextHAM`. The pre-trained weights are available here: `https://dzefile.hpccube.com:65011/efile/s/w/bmV4dGhhbQ==_191f3754a48697e8&`, the extraction code is QoYA. The dataset **Materials-SOC-HAM** is available here: `https://dzefile.hpccube.com:65011/efile/s/w/bmV4dGhhbQ==_c3c50c552df97ace&`, the extraction code is DGEo. Details on loss functions are presented in Appendix G. Information regarding the dataset construction along with comprehensive statistical summaries can be found in Appendix H. The implementation details of the network architecture and the training setup are presented in Appendix I.

## ETHICS STATEMENT

This paper presents work whose goal is to advance the field of deep learning driven by materials research. Although our field has not yet exhibited direct negative social or ethical consequences, we recognize the importance of anticipating broader concerns. A key issue is the limited interpretability of deep learning systems, which obscures the reasoning behind their predictions and constrains their utility for gaining physical understanding. We stress the need for improving model interpretability, particularly regarding how physical knowledge is incorporated and represented, to guarantee both the accuracy of predictions and their applicability to a wide range of scientific problems.

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

## A ELECTRONIC STRUCTURE CALCULATIONS: FROM DENSITY FUNCTIONAL THEORY TO DEEP LEARNING METHODS

Density Functional Theory (DFT) (Hohenberg & Kohn, 1964; Kohn & Sham, 1965) has established itself as a foundational tool in modern electronic structure theory, with wide-ranging applications in condensed matter physics, quantum chemistry, and materials science. First developed in the 1960s by Hohenberg, Kohn, and Sham, DFT reformulates the many-electron problem by replacing the complex many-body wavefunction with the electron density $\rho(\mathbf{r})$ as the central variable. This shift dramatically simplifies the computational treatment of quantum systems while retaining the essential physics, making it feasible to study realistic systems under accepted computational cost. Over the years, DFT has become indispensable for tasks such as computing band structures and orbital energies, performing structural optimizations, and predicting a variety of electronic, magnetic, and optical properties. Its broad applicability and computational efficiency have cemented its role as a key methodology across multiple scientific domains.

At the heart of density functional theory (DFT) lies the Kohn–Sham (KS) equation (Kohn & Sham, 1965), which reformulates the many-body electronic problem into a tractable set of single-particle equations:

$$\hat{H}\psi_i(\mathbf{r}) = \epsilon_i \psi_i(\mathbf{r}), \quad \text{with} \quad \hat{H} = -\frac{\hbar^2}{2m}\nabla^2 + V_{\text{ext}}(\mathbf{r}) + V_{\text{HXC}}[\rho](\mathbf{r}), \tag{5}$$

where $\hat{H}$ is the effective single-particle Hamiltonian. The potential includes the external potential $V_{\text{ext}}(\mathbf{r})$, and the Hartree–exchange–correlation (HXC) potential $V_{\text{HXC}}[\rho](\mathbf{r}) = V_{\text{H}}[\rho](\mathbf{r}) + V_{\text{XC}}[\rho](\mathbf{r})$, which is a functional of the electron density $\rho(\mathbf{r})$. The density itself is obtained from the KS orbitals via:

$$\rho(\mathbf{r}) = \sum_{m=1}^{M} |\psi_m(\mathbf{r})|^2, \tag{6}$$

where $M$ is the number of occupied single-particle states.

To numerically solve Eq. (5), a basis set is introduced. Atomic orbitals (Lin et al., 2023) are a widely adopted choice due to their localized nature and computational efficiency—they typically require fewer basis functions to reach a given level of accuracy compared to plane-wave or other delocalized bases. The atomic basis functions are products of a radial function and a spherical harmonic, that is,

$$\phi_{\kappa\zeta lm}(\mathbf{r}) = f_{\kappa\zeta l}(r)\, \widetilde{Y}_{lm}(\tilde{\mathbf{r}}), \tag{7}$$

where $\kappa$ denotes the element type, $lm$ denotes the angular momentum and the magnetic quantum number. Usually, real spherical harmonic functions are used. The radial functions are typically tabulated numerically on a fine radial mesh, and hence these basis functions are referred to as NAOs. the radial functions $f_{\kappa\zeta l}(\mathbf{r})$ are expanded in terms of spherical Bessel functions and truncated beyond a cutoff distance $r_c$

$$f_{\kappa\zeta l}(\mathbf{r}) = \begin{cases} \sum_q c_{\kappa\zeta lq} j_l(qr), & r < r_c, \\ 0, & r \geq r_c. \end{cases} \tag{8}$$

The KS eigenfunctions are expanded in terms of these atomic orbitals:

$$\psi_{n\mathbf{k}}(\mathbf{r}) = \frac{1}{\sqrt{N_k}} \sum_{\mathbf{R}} \sum_{\mu} C_{n\alpha,\mathbf{k}} e^{i\mathbf{k}\cdot\mathbf{R}} \phi_u(\mathbf{r} - \tau_i - \mathbf{R}), \tag{9}$$

where $\phi_u(\mathbf{r} - \tau_i - \mathbf{R})$ are the $u$th atomic orbitals centered on the $i$th atom in the unit cell $\mathbf{R}$, and $\alpha = \{u, i\}$ is the composite index for the NAOs. $C_{n\alpha,\mathbf{k}}$ are the coefficients of orbitals $\alpha$ of band $n$ at $\mathbf{k}$ point, and $N_k$ is the number of unit cells in the Born–von–Kármán supercell under the periodic boundary conditions, equivalent to the number of $\mathbf{k}$ points in the first Brillouin zone (BZ).

Given the expansion of the KS states in terms of atomic orbitals in Eq. (9), the KS Eq. (5) becomes ageneralized eigenvalue problem,

$$H(\mathbf{k})C_{\mathbf{k}} = E_{\mathbf{k}}S(\mathbf{k})C_{\mathbf{k}}, \tag{10}$$

where $H(\mathbf{k})$, $S(\mathbf{k})$, and $C_{\mathbf{k}}$ are the Hamiltonian matrix, overlap matrix and eigenvectors at a given $\mathbf{k}$ point, respectively. $E_{\mathbf{k}}$ is a diagonal matrix whose entries are the KS eigenenergies, $\epsilon_{n\mathbf{k}}$ denotes the energy eigenvalue of the $n$-th KS eigenstate. To obtain the Hamiltonian matrix $H(\mathbf{k})$, we first calculate the Hamiltonian in real space as

$$H_{\alpha\beta}(\mathbf{R}) = \left\langle \phi_{\alpha 0} \left| -\frac{\hbar^2}{2m}\nabla^2 + V_{\text{ext}} + V_{\text{HXC}}[\rho] \right| \phi_{\beta\mathbf{R}} \right\rangle, \tag{11}$$

where $\alpha, \beta$ are atomic orbital indices within one unit cell, and $\phi_{\alpha 0} \overset{\text{def}}{=} \phi_u(\mathbf{r} - \tau_i)$, $\phi_{\beta\mathbf{R}} \overset{\text{def}}{=} \phi_v(\mathbf{r} - \tau_j - \mathbf{R})$. The Hamiltonian matrix at a given $\mathbf{k}$ point can be obtained via a Fourier transform,

$$H_{\alpha\beta}(\mathbf{k}) = \sum_{\mathbf{R}} e^{i\mathbf{k}\cdot\mathbf{R}} H_{\alpha\beta}(\mathbf{R}). \tag{12}$$

Similarly, the overlap matrix at a given $\mathbf{k}$ point is obtained as

$$S_{\alpha\beta}(\mathbf{k}) = \sum_{\mathbf{R}} e^{i\mathbf{k}\cdot\mathbf{R}} S_{\alpha\beta}(\mathbf{R}), \tag{13}$$

where

$$S_{\alpha\beta}(\mathbf{R}) = \langle \phi_{\alpha 0} | \phi_{\beta\mathbf{R}} \rangle. \tag{14}$$

The overall computational procedure follows an iterative self-consistent (SC) loop:

1. **Initial Guess**: Start with an initial electron density $\rho^{(0)}(\mathbf{r})$. Initialize the number of iterations $t$ to 0.
2. **Potential Construction**: Compute the effective potential $V_{\text{HXC}}^{(t)}[\rho](\mathbf{r})$ by $\rho^{(t)}(\mathbf{r})$.
3. **Hamiltonian Assembly**: Construct the Hamiltonian matrix $\mathbf{H}^{(t)}$ using the current potential using Eq. (11).
4. **Eigenproblem Solution**: Perform a Fourier transformation and solve the generalized eigenvalue problem in Eq. (10) to obtain the KS eigenfunctions $\psi_{n\mathbf{k}}(\mathbf{r})$ and eigenvalues $\epsilon_{n\mathbf{k}}$.
5. **Density Update**: Compute the updated density $\rho^{(t)}(\mathbf{r})$ from the new orbitals using Eq. (6).
6. **Convergence Check**: Update $t \to t+1$, repeat steps 2–6 until the input and output densities agree within a chosen convergence threshold.

This procedure can be summarized schematically as:

$$\rho^{(0)}(\mathbf{r}) \to V_{\text{HXC}}^{(0)}[\rho](\mathbf{r}) \to \mathbf{H}^{(0)} \to \psi_{n\mathbf{k}}^{(0)}(\mathbf{r}) \to \rho^{(1)}(\mathbf{r}) \to \cdots \to \rho^{(T)}(\mathbf{r}) \to V_{\text{HXC}}^{(T)}[\rho](\mathbf{r}) \to \mathbf{H}^{(T)}.$$

Once self-consistency is reached at iteration $T$, the final Hamiltonian matrix $\mathbf{H}^{(T)}$ can be used to compute down-stream physical quantities such as total energy, band structure, orbital energies, and derived electronic, magnetic, or transport properties.

Despite the remarkable success of Kohn–Sham DFT in advancing fields such as materials science, energy, and biomedicine over recent decades (Nagy, 1998; Jones, 2015), it still faces significant computational challenges, especially when applied to large atomic systems under limited computational resources. The primary bottlenecks arise from two aspects. First, the matrix diagonalization in Eq. (10) scales as $\mathcal{O}(N^3)$, where $N$ is the number of atoms in the system. Second, the iterative nature of the SC procedure requires $T$ rounds of self-consistent updates, which further amplifies the overall computational cost. This becomes particularly problematic when a high level of convergence accuracy is needed or when dealing with complex systems, often making it difficult to complete the calculations within reasonable time or resource constraints.

To address this challenge, recent approaches (Schütt et al., 2019; Unke et al., 2021; Li et al., 2022; Gong et al., 2023; Yu et al., 2023b; Li et al., 2025; Luo et al., 2025; Yin et al., 2025) have adopted the deep graph learning paradigm to predict the self-consistent Hamiltonians. These methods bypass the iterative and computationally intensive matrix diagonalization steps in traditional DFT algorithms by directly predicting the final converged Hamiltonian matrix $H_{\alpha\beta}^{(T)}$ in a single forward pass. As

shown in Eq. (11), the Hamiltonian matrix is inherently sparse: only pairs of atoms within a cutoff radius contribute non-zero elements. Therefore, the total number of Hamiltonian matrix elements that need to be computed scales with the number of local atomic pairs in the system, leading to a complexity of $\mathcal{O}(N\overline{E})$, where $N$ is the total number of atoms and $\overline{E}$ denotes the average number of neighboring atoms within the cutoff radius per atom. Since the atomic orbital basis functions have finite spatial support, matrix elements vanish beyond a certain inter-atomic distance. In small systems where all atoms lie within each other's cutoff radius, $\overline{E} \sim N$, and the total number of non-zero elements scales as $\mathcal{O}(N^2)$. However, in sufficiently large systems, $\overline{E}$ saturates to a constant determined by local geometry, making the number of non-zero Hamiltonian elements scale linearly as $\mathcal{O}(N)$. Moreover, since most physical properties, such as transport, optical, and topological properties, depend only on the energy bands near the Fermi level, it is unnecessary to solve for the eigenfunctions of all occupied states once the Hamiltonian is known. Since the Hamiltonian matrix is sparse and only a limited number of bands near the Fermi level are needed, these eigenstates can be efficiently computed using methods like the shift-invert approach available in the ARPACK package (Lehoucq et al., 1998), with a computational complexity of $\mathcal{O}(N)$ for large systems.

These deep-learning approaches have exploited the sparsity of the Hamiltonian, yielding a computational cost that scales approximately linearly with the number of non-zero matrix elements and enabling efficient, scalable prediction of quantum properties in large atomic systems. As a result, they offer a significant efficiency advantage over traditional DFT methods with computational complexity of $\mathcal{O}(TN^3)$. This efficiency makes them particularly promising for predicting electronic structures of complex atomic systems under limited computational resources, potentially accelerating down-stream application areas like materials simulation and design.

## B  OVERVIEW OF BASIC CONCEPTS IN GROUP THEORY

This section reviews several fundamental concepts from group theory that form the basis of the symmetry principles employed in this work. Readers interested in a more comprehensive introduction may consult the monograph of Dresselhaus et al. (2007).

**Definition B.1. Group.** A set $G$ endowed with a binary operation $\cdot$ is called a group if the following axioms hold:

1. **Closure:** For any $f, g \in G$, the product $f \cdot g$ remains in $G$.

2. **Associativity:** For all $f, g, h \in G$, the equality $(f \cdot g) \cdot h = f \cdot (g \cdot h)$ holds.

3. **Identity:** There exists an element $e \in G$ such that $e \cdot f = f \cdot e = f$ for every $f \in G$.

4. **Inverse:** Each $f \in G$ has an inverse $f^{-1} \in G$ satisfying $f \cdot f^{-1} = f^{-1} \cdot f = e$.

**Definition B.2. Group Representation.** A representation of a group $G$ on a tensor space $T(V)$ is a homomorphism

$$\rho : G \to GL(T(V)),$$

mapping each group element to an invertible linear operator acting on $T(V)$. The mapping preserves the group structure, i.e.,

$$\rho(g_1 g_2) = \rho(g_1)\rho(g_2), \quad \rho(e) = I.$$

**Definition B.3. Irreducible Representation.** Let $\rho : G \to GL(V)$ be a representation on a vector space $V$. It is called *irreducible* if no nontrivial subspace $W \subset V$ exists such that $\rho(g)W \subseteq W$ for all $g \in G$. If such a proper invariant subspace exists, the representation is said to be reducible.

**Definition B.4. Equivariant Map.** Let $\rho_V : G \to GL(T(V))$ and $\rho_W : G \to GL(T(W))$ be representations of group $G$. A function $f : T(V) \to T(W)$ is *equivariant* if

$$f(\rho_V(g)\, v) = \rho_W(g)\, f(v), \qquad \forall g \in G, \; v \in T(V).$$

**Definition B.5. Invariant Map.** Given a representation $\rho_V : G \to GL(T(V))$, a function $f : T(V) \to T(W)$ is *invariant* under group $G$ if

$$f(\rho_V(g)\, v) = f(v), \qquad \forall g \in G, \; v \in T(V).$$

**Definition B.6. The Group SO(3).** The special orthogonal group SO(3) consists of all real $3 \times 3$ rotation matrices:

$$\mathrm{SO}(3) = \{\mathbf{R} \in \mathbb{R}^{3\times3} \mid \mathbf{R}^\top \mathbf{R} = I, \ \det(\mathbf{R}) = 1\}.$$

Elements of SO(3) represent rotations in three-dimensional Euclidean space.

**Definition B.7. Representations of SO(3).** A representation of SO(3) is a homomorphism

$$\rho : \mathrm{SO}(3) \to GL(V).$$

The irreducible representations of SO(3) are labeled by a non-negative integer $l$, which corresponds to the angular momentum quantum number in quantum mechanics.

**Definition B.8. Wigner-D Matrices.** A standard family of irreducible representations of SO(3) is provided by the Wigner–D matrices:

$$D^l_{m'm}(\mathbf{R}) = \langle l, m' | \mathbf{R} | l, m \rangle,$$

where $|l, m\rangle$ denotes the eigenstate of angular momentum with quantum number $l$ and magnetic index $m$. These matrices specify how angular momentum states transform under a rotation $\mathbf{R}$.

**Definition B.9. The Group O(3).** The orthogonal group $O(3)$ consists of all real $3 \times 3$ orthogonal matrices, including both proper rotations and reflections:

$$\mathrm{O}(3) = \left\{ \mathbf{R} \in \mathbb{R}^{3\times3} \mid \mathbf{R}^\top \mathbf{R} = I \right\}.$$

Elements of $O(3)$ represent all possible orthogonal transformations in three-dimensional Euclidean space, including both rotations (with determinant 1) and reflections (with determinant -1).

**Definition B.10. Irreducible Representations of the O(3) Group.** The orthogonal group O(3) contains both proper rotations ($\det = +1$), forming the subgroup SO(3), and improper rotations ($\det = -1$), including reflections and spatial inversion. An irreducible representation of O(3) is a homomorphism

$$\Gamma : \mathrm{O}(3) \to GL(V).$$

Irreducible representations of O(3) are obtained by extending the irreducible representations of O(3). For each angular-momentum degree $l$, there exist exactly two inequivalent irreducible representations of O(3):

$$\Gamma^{(l,+)}, \qquad \Gamma^{(l,-)},$$

corresponding respectively to even and odd parity under spatial inversion.

For any group element $\mathbf{R} \in \mathrm{O}(3)$, their actions are defined by

$$\Gamma^{(l,\pm)}(\mathbf{R}) = \pi_\pm(\mathbf{R})\,\mathbf{D}^{(l)}(\mathbf{R}),$$

where $\mathbf{D}^{(l)}(\mathbf{R})$ is the Wigner–D matrix giving the degree-$l$ irreducible representation of O(3), and the parity factor $\pi_\pm(\mathbf{R})$ is

$$\pi_\pm(\mathbf{R}) = \begin{cases} +1, & \det(\mathbf{R}) = +1, \\ +1 & \text{for } \Gamma^{(l,+)}, \ \det(\mathbf{R}) = -1, \\ -1 & \text{for } \Gamma^{(l,-)}, \ \det(\mathbf{R}) = -1. \end{cases}$$

Thus, $\Gamma^{(l,+)}$ is even and $\Gamma^{(l,-)}$ is odd under inversion. These two parity-extended forms exhaust all irreducible representations of O(3).

**Definition B.11. The Euclidean Group** $\mathrm{E}(3)$**.** The Euclidean group $\mathrm{E}(3)$ is the group of all rigid motions in three-dimensional space. It consists of all compositions of a rotation or reflection and a translation:

$$\mathrm{E}(3) = \{(\mathbf{R}, \mathbf{t}) \mid \mathbf{R} \in \mathrm{O}(3), \ \mathbf{t} \in \mathbb{R}^3\}.$$

The action of a group element $(\mathbf{R}, \mathbf{t})$ on a point $\mathbf{x} \in \mathbb{R}^3$ is given by

$$(\mathbf{R}, \mathbf{t}) \cdot \mathbf{x} = \mathbf{R}\mathbf{x} + \mathbf{t}.$$

**Definition B.12. Direct-Product State.** For vector spaces $V_1$ and $V_2$, their tensor product space $V_1 \otimes V_2$ consists of bilinear combinations of vectors from both spaces. A basis of $V_1 \otimes V_2$ can be written as $\{|i\rangle \otimes |j\rangle\}$, and a general element takes the form

$$|v\rangle = \sum_{i,j} c_{ij} |i\rangle \otimes |j\rangle.$$

This construction increases dimensionality multiplicatively:

$$\dim(V_1 \otimes V_2) = \dim(V_1)\,\dim(V_2).$$

A group action on the tensor-product state acts on each factor:

$$g \cdot (|v_1\rangle \otimes |v_2\rangle) = (g \cdot |v_1\rangle) \otimes (g \cdot |v_2\rangle).$$

**Definition B.13. Direct-Sum State.** For vector spaces $V_1$ and $V_2$, the direct-sum space $V_1 \oplus V_2$ consists of ordered pairs $(|v_1\rangle, |v_2\rangle)$. A general vector takes the form

$$|v\rangle = |v_1\rangle \oplus |v_2\rangle,$$

and the dimensionality increases additively:

$$\dim(V_1 \oplus V_2) = \dim(V_1) + \dim(V_2).$$

The group acts independently on each component:

$$g \cdot (|v_1\rangle \oplus |v_2\rangle) = (g \cdot |v_1\rangle) \oplus (g \cdot |v_2\rangle).$$

**Definition B.14. Physical Quantity as a Direct Product of Angular-Momentum Degrees.** Let $l_p$ and $l_q$ be two angular-momentum degrees of freedom, with corresponding O(3) irreducible representations $\Gamma^{(l_p, \pm_p)}$ and $\Gamma^{(l_q, \pm_q)}$. A tensor

$$\mathbf{Q}^{l_p \otimes l_q} \in \mathbb{R}^{(2l_p+1) \times (2l_q+1)}$$

that is formed as the direct-product quantity of these two degrees transforms under any $\mathbf{R} \in$ O(3) as

$$\mathbf{Q}^{l_p \otimes l_q}(\mathbf{R}) = \Gamma^{(l_p, \pm_p)}(\mathbf{R})\, \mathbf{Q}^{l_p \otimes l_q}\, \Gamma^{(l_q, \pm_q)}(\mathbf{R})^{\dagger}.$$

This transformation rule expresses that the tensor carries a product representation of the two angular-momentum degrees, each transforming according to its respective O(3) irrep with the appropriate parity.

**Definition B.15. Clebsch-Gordan Decomposition for O(3).** Given two angular-momentum degrees $l_p$ and $l_q$, the tensor $\mathbf{Q}^{l_p \otimes l_q}$ introduced in Definition B.14 can be decomposed as:

$$CGDecomp(\mathbf{Q}^{l_p \otimes l_q}) = \bigoplus_{l=|l_p - l_q|}^{l_p + l_q} \mathbf{q}^l,$$

with components:

$$q_m^l = \sum_{m_p, m_q} C_{m, m_p, m_q}^{l, l_p, l_q} Q_{m_p, m_q}^{l_p \otimes l_q},$$

where $C_{m, m_p, m_q}^{l, l_p, l_q}$ are the Clebsch-Gordan coefficients.

**Definition B.16. Parametric Clebsch-Gordan Decomposition for O(3).** To introduce learnable parameters while preserving O(3)-equivariance, the above decomposition can be extended to a parametric form. Specifically, the decomposition can be written as:

$$CGDecomp(\mathbf{Q}^{l_p \otimes l_q}; W) = \bigoplus_{l=|l_p - l_q|}^{l_p + l_q} \tilde{\mathbf{q}}^l,$$

where

$$\tilde{q}_m^l = W^l \sum_{m_p, m_q} C_{m, m_p, m_q}^{l, l_p, l_q} Q_{m_p, m_q}^{l_p \otimes l_q},$$

and $W = \{w_l\}_{l=|l_p - l_q|}^{l_p + l_q}$ is a set of scalar (or channel-wise) weights that act on each irreducible component. Since each $w_l$ acts as a scalar on the entire $(2l+1)$-dimensional subspace labeled by $l$ and does not mix the magnetic indices $m$, the map

$$\mathbf{Q}^{l_p \otimes l_q} \mapsto \{\tilde{\mathbf{q}}^l\}_l$$

remains O(3)-equivariant. The non-parametric Clebsch-Gordan decomposition is recovered as the special case where $w_l = 1$ for all $l$.

## C   RELATED WORK

The foundation of deep learning-based electronic-structure Hamiltonian prediction involves constructing neural networks that respect E(3)-symmetry, which inherently includes equivariance to the O(3) group, covering 3D rotations and inversions. O(3)-equivariant graph neural networks typically construct and update features using group-theoretic operators that preserve equivariance, such as linear combinations of tensors, direct sums, tensor products, Clebsch-Gordan decompositions, and tensor contractions (Thomas et al., 2018; Schütt et al., 2018; Gasteiger et al., 2021; Batzner et al., 2022; Batatia et al., 2022; Geiger & Smidt, 2022; Wang et al., 2024a).

However, since traditional non-linear activation functions, when applied directly to O(3)-equivariant features, may break equivariance, a central research topic is to reconcile strong non-linear expressiveness with strict O(3)-equivariance. An early attempt to address this problem was the use of gated activation functions (Weiler et al., 2018), which first apply non-linear activations to O(3)-invariant features and then use them as coefficients to scale the O(3)-equivariant features. Representative works adopting this mechanism include Allegro (Musaelian et al., 2023) for force and energy prediction, as well as DeepH-E3 (Gong et al., 2023) and QHNet (Yu et al., 2023b) for Hamiltonian prediction. To further enhance non-linear expressiveness of O(3)-networks, Zitnick et al. (2022) and Passaro & Zitnick (2023) proposed eSCN (equivariant Spherical Channel Networks), which applies non-linear operations to the coefficients obtained from the spherical decomposition of features. This approach has been widely used in Hamiltonian prediction (Wang et al., 2024b;c) tasks. Nevertheless, eSCN methods project features onto discrete basis functions through inner-product operations, which may degrade strict SO(3)-equivariance to a discrete sub-group. Furthermore, they use SO(2) convolutions in place of SO(3) convolutions, which could result in a loss of strict inversion equivariance. These trade-offs may influence the physical consistency of the results. As introduced in detail in Appendix D, Yin et al. (2025) proposed the TraceGrad method, which effectively unifies strict O(3)-equivariance, with strong non-linear expressiveness for Hamiltonian prediction; however, the backbone network it adopted is a simple graph neural network and has not yet evolved into a non-linear equivariant Transformer framework.

Despite these progresses, deep learning methods for Hamiltonian prediction still face substantial challenges on generalization performance, which can be summarized as follows. First, crystalline materials commonly found in nature can be composed of over $65$ different elements from the first six rows of the periodic table, leading to an exceptionally large and heterogeneous input space for deep neural network models. Existing deep learning methods for Hamiltonian prediction typically employ learnable embeddings to represent nodes (atoms) and edges (atom pairs). These embeddings are randomly initialized and learned directly from the dataset, without incorporating any explicit physical priors. As a result, they struggle to capture the fundamental physical relationships between different atoms and across different material systems, which are crucial for generalization. Second, as illustrated in Figure 1, the regression target, namely the self-consistent electronic-structure Hamiltonian, is inherently high-dimensional and complex, especially when considering SOC effects. For instance, a system containing several tens of atoms may involve nearly several thousands of non-zero Hamiltonian matrix elements that need to be accurately predicted. Most of the existing methods attempt to directly predict the entire self-consistent Hamiltonian matrix, namely $\mathbf{H}^{(T)}$ as formulated in Appendix A, placing a heavy burden on the model due to the vast size of the output space, often resulting in optimization difficulties during training and limited generalization to unseen systems. In addition, most existing methods treat the real-space Hamiltonian as the sole regression target, which can lead to sub-optimal physical fidelity in down-stream applications, particularly in capturing low-energy band structures accurately. Although Li et al. (2025) designed a method for molecular systems to reduce the regression space of Hamiltonians and introduced a basis transformation of the Hamiltonian matrix in the wavefunction loss function to improve the prediction accuracy of downstream physical quantities, their approach is limited to molecular systems and is not applicable to periodic crystalline materials, which have different mathematical formulations and physical properties. For example, in the case of periodic crystalline materials, the predicted electronic-structure Hamiltonian may involve erroneous couplings between high-energy and low-energy subspaces in the $k$-space, which can affect the accuracy of downstream physical quantity predictions. These issues require entirely new considerations. Moreover, their formulation does not explicitly account for the inherent gauge freedom in Hamiltonian representations. Physically, adding a global shift $\mu\mathbf{S}$ to the Hamiltonian matrix, where $\mu$ is an arbitrary scalar and $\mathbf{S}$ is the overlap matrix, leaves down-

stream physical quantities unchanged. This property necessitates a gauge-invariant error metric for rigorous evaluation. However, as their approach currently lacks a mechanism to handle this gauge ambiguity, it may potentially lead to optimization instability and physically inconsistent predictions in crystal systems.

As a result, constructing a unified model that generalizes across diverse crystal prototypes remains challenging, and many existing approaches explicitly constrain their scope. For example, Li et al. (2022), Gong et al. (2023), and Xia et al. (2025) each train and evaluate their methods within a single material system (e.g., $MoS_2$, $Bi_2Se_3$, or a-$HfO_2$), without assessing cross-material generalization. More recently, DeepH-2 (Wang et al., 2024c) broadened coverage to systems involving elements primarily from the first four rows of the periodic table; however, they reduced the orbital basis by omitting $f$-orbitals. While such choices help reduce computational and modeling complexity, they may limit broad applicability to the full diversity of real-world materials. Zhong et al. (2024) developed a Hamiltonian prediction model aimed at a broader range of element types, while also highlighting the challenge of achieving consistently high accuracy across diverse crystal systems. What's more, except for very few exceptions such as DeepH-E3 (Gong et al., 2023), most existing prediction models neglect the spin-orbit coupling (SOC) effect. Furthermore, open-source datasets with a broad and diverse collection of materials dedicated to training and validating universal Hamiltonian models across the periodic table remain scarce. Although the QH9 (Yu et al., 2023a) dataset is a well-known open-source Hamiltonian dataset, it consists of molecular systems rather than periodic material systems, and includes only structures composed of C, H, O, N, and F elements. To solve these challenges, this work presents an advanced unified deep learning framework together with a large benchmark dataset for Hamiltonian prediction, targeting broader generalization across richer classes of materials.

## D  OVERVIEW OF THE TRACEGRAD PARADIGM

The TraceGrad (Yin et al., 2025) mechanism addresses a key challenge in conventional neural architectures, which struggle to preserve O(3)-equivariance when applying non-linear transformations to higher-degree tensor features . It offers a principled solution to achieving strong non-linear expressiveness while strictly maintaining O(3)-equivariance. As our neural network builds upon this foundational idea, we first review the motivation and core mechanism of TraceGrad in this appendix. This background will help readers better understand the extensions and architectural developments presented in Section 2.2, where we adapt and generalize TraceGrad into a high-capacity Transformer framework tailored for Hamiltonian learning.

### D.1  THE EQUIVARIANCE-EXPRESSIVENESS DILEMMA

In O(3)-equivariant neural networks, intermediate features at the $k$-th layer are represented in the direct-sum form of irreducible components, i.e.,

$$\mathbf{f}^{(k)} = \bigoplus_{l \in L^{(k)}} \mathbf{f}^{(k)l}, \qquad \mathbf{f}^{(k)l} \in \mathbb{R}^{2l+1},$$

and they must satisfy the transformation rule:

$$\mathbf{f}^{(k)l}(\mathbf{R}) = \Gamma^{(l,\pm)}(\mathbf{R})\,\mathbf{f}^{(k)l}.$$

The main difficulty is to construct a non-linear operator $g_{\mathrm{nonlin}}(\cdot)$ that provides genuine nonlinear expressive power while still preserving strict O(3)-equivariance. The equivariance condition requires that the output after a non-linear update satisfies:

$$\mathbf{f}^{(k+1)l}(\mathbf{R}) = \Gamma^{(l,\pm)}(\mathbf{R})\,\mathbf{f}^{(k+1)l}, \qquad \mathbf{R} \in O(3),$$

where

$$\mathbf{f}^{(k+1)l} = g_{\mathrm{nonlin}}\left(\mathbf{f}^{(k)l}\right).$$

However, applying standard element-wise nonlinear activation functions such as SiLU or Softmax to $\mathbf{f}^{(k)l}$ with $l \geq 1$ breaks the required equivariance condition. On the other hand, avoiding non-linear operations, significantly limits the expressive power of the model, restricting its fitting and generalization performance. Reconciling these two requirements preserving strict equivariance while

allowing strong nonlinear expressiveness therefore poses a fundamental challenge. This challenge is precisely what the TraceGrad method is designed to overcome.

This challenge is precisely what the TraceGrad method is designed to overcome. The TraceGrad approach addresses this by proposing a unified representation learning framework that integrates O(3)-equivariant and O(3)-invariant physical quantities and neural representations. It constructs O(3)-invariant trace quantities as supervision signals to learn high-quality non-linear O(3)-invariant features. These features are then leveraged to induce non-linear O(3)-equivariant representations through a gradient operator, effectively achieving a strict unification of O(3)-equivariance and non-linear expressiveness.

### D.2   CONSTRUCTING O(3)-INVARIANT TRAINING LABELS

For an Hamiltonian block $\mathbf{H}$, TraceGrad defines an O(3)-invariant **trace** quantity:

$$\mathbf{T} = \mathrm{tr}\big(\mathbf{H} \cdot (\mathbf{H})^{\dagger}\big),$$

which satisfies the equivariance condition:

$$\mathbf{T}(\mathbf{R}) = \mathbf{T}, \quad \forall \mathbf{R} \in \mathrm{O}(3).$$

These invariant quantities are directly derived from the equivariant targets and serve as additional supervision for learning O(3)-invariant neural representations, without the need for extra labeling efforts.

### D.3   BUILDING NON-LINEAR O(3)-INVARIANT FEATURES FROM EQUIVARIANT FEATURES

Given an O(3)-equivariant feature $\mathbf{f}^{(k)} = \bigoplus_{l \in L^{(k)}} \mathbf{f}^{(k)l}$, TraceGrad first forms tensor products $\mathbf{f}^{(k)l_i} \otimes \mathbf{f}^{(k)l_j}$ and applies an extended, parametric Clebsch-Gordan decomposition to extract the degree-0 component:

$$u_c^{(k)} = \mathrm{CGDecomp}_{\mathrm{ext}}\big(\mathbf{f}^{(k)} \otimes \mathbf{f}^{(k)}; W\big)\Big|_0 = \sum_{l_i, l_j \in L^{(k)}, \, l_i = l_j} W_{ij}^c \cdot \mathrm{CGDecomp}\big(\mathbf{f}^{(k)l_i} \otimes \mathbf{f}^{(k)l_j}\big)\Big|_0,$$

where $W = \{W_{ij}^c\}$ are learnable scalar (or channel-wise) weights, and $\Big|_0$ indicates that we only take the scalar ($l = 0$) component of the Clebsch-Gordan decomposition. Collecting all channels gives an invariant vector $\mathbf{u}^{(k)} = [u_1^{(k)}, \ldots, u_C^{(k)}]$. As each degree-0 component is O(3)-invariant, $\mathbf{u}^{(k)}$ is O(3)-invariant.

An arbitrary differentiable non-linear neural network $s_{\mathrm{nonlin}}$ is then applied:

$$\mathbf{z}^{(k)} = s_{\mathrm{nonlin}}\Big(\mathbf{u}^{(k)}\Big),$$

yielding O(3)-invariant features $\mathbf{z}^{(k)}$ with non-linear expressiveness, since invariance is preserved under any non-linear function of invariant inputs.

### D.4   INDUCING NON-LINEAR O(3)-EQUIVARIANT FEATURES VIA GRADIENTS

TraceGrad uses the gradient of the invariant scalar with respect to the equivariant feature to yield an O(3)-equivariant and non-linearly enriched representation. For a single scalar channel $z_c^{(k)}$, the gradient is defined as:

$$\mathbf{v}_c^{(k)} = \frac{\partial z_c^{(k)}}{\partial \mathbf{f}^{(k)}}.$$

Mathematically, $\mathbf{v}_c^{(k)}$ is O(3)-equivariant, meaning that for any rotation $\mathbf{R} \in \mathrm{O}(3)$,

$$\mathbf{v}_c^{(k)}(\mathbf{R}) = \Gamma(\mathbf{R})\,\mathbf{v}_c^{(k)},$$

where $\Gamma(\mathbf{R})$ is the representation matrix. Summing over all channels, we obtain an O(3)-equivariant feature, now enriched with the non-linear expressive power of $\mathbf{z}^{(k)}$:

$$\mathbf{v}^{(k)} = \sum_{c=1}^{C} \mathbf{v}_c^{(k)}.$$

In practice, TraceGrad combines the original equivariant feature and its gradient-induced counterpart in a residual fashion:
$$\mathbf{f}^{(k+1)} = \mathbf{f}^{(k)} + \mathbf{v}^{(k)}.$$

Since $\mathbf{f}^{(k)}$ and $\mathbf{v}^{(k)}$ are O(3)-equivariant, the new feature $\mathbf{f}^{(k+1)}$ also remains O(3)-equivariant:

$$\mathbf{f}^{(k+1)}(\mathbf{R}) = \Gamma(\mathbf{R})\,\mathbf{f}^{(k+1)}.$$

Thus, by applying the gradient operation in this residual fashion, we ensure that both the original and the updated features maintain O(3)-equivariance throughout the layers. The model stacks $K$ such modules to build a deep encoder of O(3)-equivariant non-linear representations.

### D.5 Joint Decoding and Training Objective

TraceGrad uses two decoding branches: (i) an O(3)-equivariant decoder that maps the final equivariant features $\mathbf{f}^{(K)}$ to the target Hamiltonian block $\mathbf{H}$; and (ii) an O(3)-invariant decoder that maps the collection of invariant features $\{\mathbf{z}^{(k)}\}_{k=1}^{K}$ to the trace quantities $\mathbf{T} = \mathrm{tr}(\mathbf{H} \cdot (\mathbf{H})^{\dagger})$.

The two branches are trained jointly with a coupled loss function that balances the error on both the equivariant block predictions and the invariant trace predictions. A adaptive factor adjusts the relative importance of each loss component, ensuring an effective balance during training without backpropagating gradients through the scalar factor.

## E Details on the Construction of Initial Node and Edge Features

### E.1 Initial Node Features

To construct the initial node features, we extract the on-site Hamiltonian block $\mathbf{H}_{aa}^{(0)}$ for each atom $a$ from the full zeroth-step Hamiltonian $\mathbf{H}^{(0)}$. This block encodes the local electronic environment of atom $a$ (where $1 \le a \le N$). To convert this block into a vector-form representation compatible with the input form of equivariant neural networks, we apply the inverse transformation of the Wigner–Eckart layer (Gong et al., 2023), which transforms the SOC Hamiltonian block into a direct sum of vector-form representations aligned with the symmetry of E(3). The transformation is applied to $\mathbf{H}_{aa}^{(0)}$ as follows:

$$\mathbf{f}_a^{(\text{node-init})} = \text{Inv\_Wigner\_Eckart}\left(\mathbf{H}_{aa}^{(0)}\right). \tag{15}$$

### E.2 Initial Edge Features

To construct the initial edge features, we first extract the Hamiltonian block $\mathbf{H}_{ab}^{(0)}$ corresponding to the interaction between atoms $a$ and $b$ from the zeroth-step Hamiltonian $\mathbf{H}^{(0)}$. We then apply the inverse transformation of the Wigner–Eckart layer (Gong et al., 2023) to convert $\mathbf{H}_{ab}^{(0)}$ into direct-sum state:

$$\mathbf{h}_{ab}^{(0)} = \text{Inv\_Wigner\_Eckart}\left(\mathbf{H}_{ab}^{(0)}\right). \tag{16}$$

Next, we apply a spherical harmonics transformation to the normalized displacement vector $\mathbf{r}_{ab} = (\mathbf{r}_b - \mathbf{r}_a)$ to encode the directional information between atoms $a$ and $b$. The spherical harmonics function is defined as $\mathbf{Y}_{ab} = Y_l^m\left(\frac{\mathbf{r}_{ab}}{|\mathbf{r}_{ab}|}\right)$, where $\mathbf{r}_a$ and $\mathbf{r}_b$ represent the position vectors of atoms $a$ and $b$, respectively.

Next, we introduce a parameterized Clebsch-Gordan decomposition (as defined in Definition B.16 of Appendix B), which is applied to the tensor product of the combined edge features and the spherical harmonics functions. The resulting edge feature is computed as:

$$\mathbf{f}_{ab}^{(\text{edge-init})} = \text{CG\_Decomp}\left(\left(\mathbf{f}_a^{(\text{node-init})} \oplus \mathbf{h}_{ab}^{(0)} \oplus \mathbf{f}_b^{(\text{node-init})}\right) \otimes \mathbf{Y}_{ab}; W\right), \tag{17}$$

where $W$ are the weights derived from a Gaussian expansion of the displacement vector magnitude $|\mathbf{r}_{ab}|$. This weights are defined as:

$$W = \text{GaussianBasis}(|\mathbf{r}_{ab}|) = \exp\left(-\frac{(|\mathbf{r}_{ab}| - d_k)^2}{2\sigma_k^2}\right), \quad 1 \le k \le K, \tag{18}$$

where $d_k$ (for $1 \le k \le K$) corresponds to a set of reference distances, and $\sigma_k$ controls the width of the Gaussian function. These weights modulate the Clebsch-Gordan decomposition, enabling the incorporation of distance information between atoms $a$ and $b$.

Eq. (17) integrates the initial node features, the interaction captured by the zeroth-step Hamiltonian, and the directional and distance information from the displacement vector into a unified edge descriptor, i.e., $\mathbf{f}_{ab}^{(\text{edge-init})}$. The use of tensor products and parameterized Clebsch-Gordan decomposition allows us to efficiently combine these various types of information in a form that is compatible with the input form of equivariant neural networks. This enables the model to effectively capture both local atomic environments and interatomic interactions in an expressive, physically informed manner.

## F  INTRODUCTION OF RECIPROCAL SPACE ELECTRONIC-STRUCTURE HAMILTONIANS INTO DEEP LEARNING PARADIGM

The self-consistent Hamiltonian $\mathbf{H}^{(T)}$, obtained through the procedure described in Section A, is inherently defined in **real space**. Its matrix elements $H_{\alpha\beta}^{(T)}$ are constructed over localized atomic orbital basis functions centered at atoms, and are truncated beyond a spatial cutoff. While real-space representations are efficient for representing local interactions, many physical phenomena such as band structures, effective low-energy models, and quasiparticle dynamics are most naturally described in reciprocal space.

To obtain a reciprocal-space Hamiltonian, we perform a Fourier transformation of the real-space matrix elements. For a periodic system with lattice vectors $\{\mathbf{R}\}$, the Bloch Hamiltonian $H(\mathbf{k})$ at wavevector $\mathbf{k}$ is defined as:

$$H_{\alpha\beta}(\mathbf{k}) = \sum_{\mathbf{R}} e^{i\mathbf{k}\cdot\mathbf{R}} H_{\alpha\beta}(\mathbf{R}), \tag{19}$$

where $i$ is the imaginary unit ($i^2 = -1$), and $H_{\alpha\beta}(\mathbf{R})$ denotes the real-space Hamiltonian matrix element between orbital $\alpha$ in a reference unit cell and orbital $\beta$ in a cell displaced by lattice vector $\mathbf{R}$. These elements are directly taken from the converged real-space Hamiltonian $\mathbf{H}^{(T)}$ defined over the localized atomic orbital basis, with each pair of orbitals uniquely associated with a displacement vector $\mathbf{R}$. For simplicity, we omit the superscript $(T)$ in Eq. (19), with the understanding that all real-space matrix elements originate from $\mathbf{H}^{(T)}$.

Diagonalizing $H(\mathbf{k})$ at each wavevector $\mathbf{k}$ in the Brillouin zone yields the system's electronic band structure:

$$H(\mathbf{k})\psi_{n\mathbf{k}} = \varepsilon_{n\mathbf{k}} S(\mathbf{k})\psi_{n\mathbf{k}}, \tag{20}$$

Let $\hat{H}(\mathbf{k}) \in \mathbb{C}^{n \times n}$ denote a Hermitian matrix that approximates $H(\mathbf{k})$. It can also be solved through a generalized eigenvalue equation to obtain the eigenvalues and wave functions.

$$\hat{H}(\mathbf{k})\hat{\psi}_{n\mathbf{k}} = \hat{\varepsilon}_{n\mathbf{k}} S(\mathbf{k})\hat{\psi}_{n\mathbf{k}}, \tag{21}$$

where $\varepsilon_{n\mathbf{k}}$ and $\hat{\varepsilon}_{n\mathbf{k}}$ are diagonal matrices of eigenvalues, and $\psi_{n\mathbf{k}}$ and $\hat{\psi}_{n\mathbf{k}}$ are the corresponding eigenvectors. Define $\Delta H(\mathbf{k}) = H(\mathbf{k}) - \hat{H}(\mathbf{k})$. Assume a spectral gap $\delta$ separates the generalized eigenvalues of $H(\mathbf{k})$ and $\hat{H}(\mathbf{k})$. $\kappa(\cdot)$ denotes the condition number of a given matrix, $\|\cdot\|_2$ represents the spectral norm, $\|\Delta H(\mathbf{k})\|_{1,1} = \sum_{i,j} |\Delta H_{ij}(\mathbf{k})|$. Then, the difference in eigenvalues and the angle $\theta$ between the eigenspace of $H(\mathbf{k})$ and $\hat{H}(\mathbf{k})$ satisfy:

1. **Eigenvalue Differences:**

$$\left|\varepsilon_{n\mathbf{k}} - \hat{\varepsilon}_{n\mathbf{k}}\right| \le \frac{\kappa(S(\mathbf{k}))}{\|S(\mathbf{k})\|_2} \|\Delta H(\mathbf{k})\|_{1,1},$$

2. **Eigenspace Angle:**

$$\sin\theta \leq \frac{\kappa(S(\mathbf{k}))}{\|S(\mathbf{k})\|_2} \frac{\|\Delta H(\mathbf{k})\|_{1,1}}{\delta},$$

where $\theta$ is the angle between the eigenspaces corresponding to $\varepsilon_{n\mathbf{k}}$ and $\hat{\varepsilon}_{n\mathbf{k}}$. The theorem (Golub & Van Loan, 2013) highlights that due to the non-orthogonality of the orbital basis set, the errors in band energies and wave functions can be amplified by the condition number factor $\frac{\kappa(S(\mathbf{k}))}{\|S(\mathbf{k})\|_2}$. As a result, even a small error may cause the band eigenvalues and wave functions to deviate significantly from the true results, manifesting as the appearance of ghost states in the band structure.

To mitigate the amplification of perturbations in the predicted results caused by the condition number, a feasible approach is to perform a basis transformation for the Hamiltonian matrix $H(\mathbf{k})$ by introducing projection operators $\mathcal{U}(\mathbf{k})$ formed from the complete set of eigenstates $\psi_{n\mathbf{k}}$, thereby transforming $H(\mathbf{k})$ into a diagonal representation. From a physical perspective, the low-energy subspace near the Fermi level governs essential material properties such as optical, thermal and transport behaviors. Accordingly, the projected Hamiltonian $H(\mathbf{k})$ can be decomposed into three parts of the projection space:

- **Low-energy subspace** $\widetilde{\mathbf{H}}_{PP}(\mathbf{k})$: $\mathcal{P}(\mathbf{k})$ are spanned by $N_P$ eigenvectors $\{\psi_{n\mathbf{k}}\}$ with energies below the cutoff energy, the $H(\mathbf{k})$ are projected into $\mathcal{P}(\mathbf{k})$ space.

- **High-energy subspace** $\widetilde{\mathbf{H}}_{QQ}(\mathbf{k})$: $\mathcal{Q}(\mathbf{k})$ are spanned by the remaining $N_Q$ eigenvectors above the cutoff energy, the $H(\mathbf{k})$ are projected into $\mathcal{Q}(\mathbf{k})$ space.

- **Coupling subspace** $\widetilde{\mathbf{H}}_{PQ}(\mathbf{k})$: the off-diagonal coupling between $P$ and $Q$, encoded in the cross blocks of the full Hamiltonian.

Let $\mathcal{P}(\mathbf{k}) \in \mathbb{C}^{N \times N_P}$ and $\mathcal{Q}(\mathbf{k}) \in \mathbb{C}^{N \times N_Q}$ be the matrices whose columns are orthonormal eigenvectors spanning the low- and high-energy subspaces, stacking the bases as

$$\mathcal{U}(\mathbf{k}) = \begin{bmatrix} \mathcal{P}(\mathbf{k}) \ \mathcal{Q}(\mathbf{k}) \end{bmatrix} \in \mathbb{C}^{N \times (N_P + N_Q)},$$

and assuming $N_P + N_Q = N$ ($\mathcal{U}(\mathbf{k})^\dagger S(\mathbf{k})\mathcal{U}(\mathbf{k}) = \mathbf{1}$), the Hamiltonian in the $(\mathcal{P}, \mathcal{Q})$ basis is obtained by a single similarity transform:

$$\widetilde{\mathbf{H}}(\mathbf{k}) = \mathcal{U}(\mathbf{k})^\dagger \, \mathbf{H}(\mathbf{k}) \mathcal{U}(\mathbf{k}) = \begin{bmatrix} \widetilde{\mathbf{H}}_{PP}(\mathbf{k}) & \widetilde{\mathbf{H}}_{PQ}(\mathbf{k}) \\ \widetilde{\mathbf{H}}_{QP}(\mathbf{k}) & \widetilde{\mathbf{H}}_{QQ}(\mathbf{k}) \end{bmatrix}.$$

For the ground-truth Hamiltonian, when transformed by its own eigenbasis $\mathcal{U}(\mathbf{k})$, the cross block vanishes, i.e., $\widetilde{\mathbf{H}}_{PQ}(\mathbf{k}) = \mathbf{0}$. In contrast, when a predicted Hamiltonian is projected onto the subspaces defined by the ground-truth eigenbasis, the mismatch between the predicted and exact eigenvectors may produce spurious non-zero entries, $\widetilde{\mathbf{H}}_{PQ}(\mathbf{k}) \neq 0$. These unphysical couplings manifest as artifacts such as ghost states, and thus provide a meaningful signal for penalization during training.

Because the eigenvalues of $\mathbf{H}(\mathbf{k})$ directly define the band structure, reciprocal-space supervision provides a natural training signal. We therefore assign distinct loss terms to the three components. The low-energy block $\widetilde{\mathbf{H}}_{PP}(\mathbf{k})$ governs the states near the Fermi level and thus dominates observable physics; accurate supervision on this block is crucial. The high-energy block $\widetilde{\mathbf{H}}_{QQ}(\mathbf{k})$ does not directly determine low-energy phenomena, but maintaining its fidelity is important: otherwise errors in $Q$ may propagate indirectly through erroneous $PQ$ couplings. Finally, the cross block $\widetilde{\mathbf{H}}_{PQ}(\mathbf{k})$ should ideally vanish; we enforce this by adding an explicit penalty on $\|\widetilde{\mathbf{H}}_{PQ}(\mathbf{k})\|$, which suppresses unphysical couplings between $P$ and $Q$, thereby eliminating ghost states and restoring the intended decoupling of subspaces.

## G   DETAILS ON TRAINING LOSS FUNCTIONS

We elaborate on the details of Eq. (1) in the following equation:

$$\text{loss}(\mathbf{R}) = \mathbb{E}_{\mathbf{R}}[\lambda_R\Big((1 - \lambda_C) \cdot \text{loss}_H(\mathbf{R}) + \gamma(\text{loss}_H, \text{loss}_T, \lambda_C) \cdot \text{loss}_T(\mathbf{R})\Big)],$$

$$\text{loss}_H(\mathbf{R}) = MSE\big(\widehat{\mathbf{H}}(\mathbf{R}), \mathbf{H}^{gt}(\mathbf{R}, \mu)\big),$$

$$\text{loss}_T(\mathbf{R}) = MAE\big(\widehat{\mathbf{T}}(\mathbf{R}), \mathbf{T}^{gt}(\mathbf{R}, \mu)\big),$$ 

$$\gamma(\text{loss}_H, \text{loss}_T, \lambda_C) = \lambda_C \cdot \text{No\_Grad}\left(\frac{\text{loss}_H(\mathbf{R})}{\text{loss}_T(\mathbf{R})}\right).$$

$$(22)$$

where $\lambda_R$ is a hyper-parameter, $\mathbf{R}$ denotes the lattice vector connecting the reference unit cell and a neighboring unit cell. $\widehat{\mathbf{H}}(\mathbf{R})$ and $\widehat{\mathbf{T}}(\mathbf{R})$ denote the predicted Hamiltonian and its corresponding trace quantity in real space, respectively. Here, we compute $\widehat{\mathbf{H}}(\mathbf{R})$ as:

$$\widehat{\mathbf{H}}(\mathbf{R}) = \mathbf{H}^{(0)}(\mathbf{R}) + \widehat{\Delta\mathbf{H}}(\mathbf{R}),$$

where $\widehat{\Delta\mathbf{H}}(\mathbf{R})$ is the predicted correction term of the Hamiltonian.

The ground truth Hamiltonians are denoted as $\mathbf{H}^{gt}(\mathbf{R}) = \mathbf{H}^{(T)}(\mathbf{R})$. However, rather than directly using these ground truth values to supervise $\widehat{\mathbf{H}}(\mathbf{R})$, we construct augmented supervision targets by introducing an additional term:

$$\mathbf{H}^{gt}(\mathbf{R}, \mu) = \mathbf{H}^{gt}(\mathbf{R}) + \mu \cdot \mathbf{S}(\mathbf{R}), \tag{23}$$

where $\mu$ is a scalar coefficient, $\mathbf{S}(\mathbf{R})$ denotes the real-space overlap matrix, and $\mathbf{H}^{(0)}(\mathbf{R})$ denotes the real-space zeroth-step Hamiltonian matrix. Following the gauge-error formulation of Wang et al. (2024c), adding a shift term $\mu \cdot \mathbf{S}(\mathbf{R})$ to the Hamiltonian leaves all down-stream physical observables unchanged. In practice, $\mu$ is chosen as the solution that minimizes the overall loss, as established in Wang et al. (2024c). This removes the gauge freedom inherent in the Hamiltonian representation, facilitating more stable and efficient convergence of the neural network.

The corresponding trace quantity used as the supervision signal is computed as:

$$\begin{aligned}\mathbf{T}^{gt}(\mathbf{R}, \mu) &= \text{tr}\left(\Delta\mathbf{H}^{gt}(\mathbf{R}, \mu) \cdot \Delta\mathbf{H}^{gt}(\mathbf{R}, \mu)^{\dagger}\right) \\ &= \text{tr}\left((\mathbf{H}^{gt}(\mathbf{R}, \mu) - \mathbf{H}^{(0)}(\mathbf{R})) \cdot (\Delta\mathbf{H}^{gt}(\mathbf{R}, \mu) - \mathbf{H}^{(0)}(\mathbf{R}))^{\dagger}\right),\end{aligned} \tag{24}$$

where $\mathbf{H}^{(0)}(\mathbf{R})$ denotes the real-space zeroth-step Hamiltonian matrix.

Inspired by Yin et al. (2025), the scaling factor $\gamma(\text{loss}_H, \text{loss}_T, \lambda_C)$ in Eq. (22) is designed to harmonize the contributions from the two loss terms, ensuring stable optimization. Here, $\lambda_C$ is a hyper-parameter that controls the overall strength of the balancing mechanism. The term $\text{No\_Grad}(\cdot)$ ensures that gradients are dropped during the computation of this coefficient, preventing interference with the back-propagation of $\text{loss}_{T(R)}$. By applying this balancing strategy, better numerical stability and balanced learning performance across both the Hamiltonian and trace quantity supervision branches can be achieved.

We elaborate on the details of Eq. (2) as follows. Let $\widehat{\mathbf{H}}(\mathbf{k})$ denote the predicted full Hamiltonian in reciprocal space, obtained from the Fourier transform of $\widehat{\mathbf{H}}(\mathbf{R})$ using Eq. (19). Similarly, let $\mathbf{H}^{gt}(\mathbf{k}, \mu)$ denote the ground-truth Hamiltonian in reciprocal space, obtained from the Fourier transform of $\mathbf{H}^{gt}(\mathbf{R}, \mu)$. Both Hamiltonians are projected by the ground-truth eigenbasis $\mathcal{U}(\mathbf{k}) = [\mathcal{P}(\mathbf{k}), \mathcal{Q}(\mathbf{k})]$, yielding block-partitioned forms:

$$\widetilde{\mathbf{H}}(\mathbf{k}) = \mathcal{U}(\mathbf{k})^{\dagger}\,\widehat{\mathbf{H}}(\mathbf{k})\,\mathcal{U}(\mathbf{k}) = \begin{bmatrix} \widetilde{\mathbf{H}}_{PP}(\mathbf{k}) & \widetilde{\mathbf{H}}_{PQ}(\mathbf{k}) \\ \widetilde{\mathbf{H}}_{QP}(\mathbf{k}) & \widetilde{\mathbf{H}}_{QQ}(\mathbf{k}) \end{bmatrix},$$

$$\widetilde{\mathbf{H}}^{gt}(\mathbf{k}, \mu) = \mathcal{U}(\mathbf{k})^{\dagger}\,\mathbf{H}^{gt}(\mathbf{k}, \mu)\,\mathcal{U}(\mathbf{k}) = \begin{bmatrix} \widetilde{\mathbf{H}}_{PP}^{gt}(\mathbf{k}, \mu) & \widetilde{\mathbf{H}}_{PQ}^{gt}(\mathbf{k}, \mu) \\ \widetilde{\mathbf{H}}_{QP}^{gt}(\mathbf{k}, \mu) & \widetilde{\mathbf{H}}_{QQ}^{gt}(\mathbf{k}, \mu) \end{bmatrix}.$$

For the exact Hamiltonian, the off-diagonal block ideally vanishes, i.e., $\widetilde{\mathbf{H}}_{PQ}^{gt}(\mathbf{k}, \mu) = \mathbf{0}$, whereas for the predicted Hamiltonian, spurious non-zero entries generally appear in $\widetilde{\mathbf{H}}_{PQ}(\mathbf{k})$, manifesting as unphysical ghost states.

The loss is then defined block-wise:

$$
\begin{aligned}
\text{loss}(\mathbf{k}) = \mathbb{E}_{\mathbf{k}}\Big[ &\lambda_{\mathcal{P}} \cdot \text{MSE}\big(\widetilde{\mathbf{H}}_{PP}(\mathbf{k}), \widetilde{\mathbf{H}}_{PP}^{gt}(\mathbf{k}, \mu)\big) \\
&+ \lambda_{\mathcal{Q}} \cdot \text{MSE}\big(\widetilde{\mathbf{H}}_{QQ}(\mathbf{k}), \widetilde{\mathbf{H}}_{QQ}^{gt}(\mathbf{k}, \mu)\big) \\
&+ \lambda_{PQ} \cdot \text{MSE}\big(\widetilde{\mathbf{H}}_{PQ}(\mathbf{k}), \widetilde{\mathbf{H}}_{PQ}^{gt}(\mathbf{k}, \mu)\big)\Big],
\end{aligned}
\tag{25}
$$

where $\lambda_{\mathcal{P}}, \lambda_{\mathcal{Q}}, \lambda_{PQ}$ are tunable hyper-parameters controlling the relative importance of the three terms.

The overall loss function combines the losses from both R-space and k-space:

$$
\begin{aligned}
\text{loss}_{all} =& \text{loss}(\mathbf{R}) + \text{loss}(\mathbf{k}) \\
=& \mathbb{E}_{\mathbf{R}}[\lambda_R\big((1-\lambda_C)\cdot\text{loss}_H(\mathbf{R}) + \gamma(\text{loss}_H, \text{loss}_T, \lambda_C)\cdot\text{loss}_T(\mathbf{R})\big)] \\
&+ \mathbb{E}_{\mathbf{k}}[\lambda_{\mathcal{P}}\cdot\text{loss}_{\mathcal{P}}(\mathbf{k}) + \lambda_{\mathcal{Q}}\cdot\text{loss}_{\mathcal{Q}}(\mathbf{k}) + \lambda_{PQ}\cdot\text{loss}_{PQ}(\mathbf{k})]
\end{aligned}
\tag{26}
$$

where the value of $\mu$ is determined by $\frac{\partial \text{loss}_{all}}{\partial \mu} = 0$. It can be solved analytically by:

$$
\begin{aligned}
\partial\Big(\frac{\lambda_R}{N_R}\sum_{\mathbf{R},\alpha\beta}\Big[\Big|\big(\widehat{\mathbf{H}}(\mathbf{R}) - \mathbf{H}^{gt}(\mathbf{R})\big)_{\alpha\beta}\Big|^2 &+ \mu^2\Big|\mathbf{S}(\mathbf{R})_{\alpha\beta}\Big|^2 \\
&- 2\mu\text{Re}\Big([\widehat{\mathbf{H}}(\mathbf{R}) - \mathbf{H}^{gt}(\mathbf{R})]_{\alpha\beta}^*\mathbf{S}(\mathbf{R})_{\alpha\beta}\Big)\Big] \\
+ \frac{\lambda_{\mathcal{P}}}{N_{\mathcal{P}}}\sum_{\mathbf{k},\alpha\beta}\Big[\Big|\big(\widetilde{\mathbf{H}}_{PP}(\mathbf{k}) - \widetilde{\mathbf{H}}_{PP}^{gt}(\mathbf{k})\big)_{\alpha\beta}\Big|^2 &+ \mu^2\delta_{\alpha\beta} \\
&- 2\mu\text{Re}\Big([\widetilde{\mathbf{H}}_{PP}(\mathbf{k}) - \widetilde{\mathbf{H}}_{PP}^{gt}(\mathbf{k})]_{\alpha\beta}^*\delta_{\alpha\beta}\Big)\Big] \\
+ \frac{\lambda_{\mathcal{Q}}}{N_{\mathcal{Q}}}\sum_{\mathbf{k},\alpha\beta}\Big[\Big|\big(\widetilde{\mathbf{H}}_{QQ}(\mathbf{k}) - \widetilde{\mathbf{H}}_{QQ}^{gt}(\mathbf{k})\big)_{\alpha\beta}\Big|^2 &+ \mu^2\delta_{\alpha\beta} \\
&- 2\mu\text{Re}\Big([\widetilde{\mathbf{H}}_{QQ}(\mathbf{k}) - \widetilde{\mathbf{H}}_{QQ}^{gt}(\mathbf{k})]_{\alpha\beta}^*\delta_{\alpha\beta}\Big)\Big] \\
+ \frac{\lambda_{\mathcal{P}\mathcal{Q}}}{N_{\mathcal{P}\mathcal{Q}}}\sum_{\mathbf{k},\alpha\beta}\Big[\Big|\big(\widetilde{\mathbf{H}}_{PQ}(\mathbf{k}) - \widetilde{\mathbf{H}}_{PQ}^{gt}(\mathbf{k})\big)_{\alpha\beta}\Big|^2\Big]&\Big)/\big(\partial\mu\big) = 0
\end{aligned}
\tag{27}
$$

which obtains:

$$
\begin{aligned}
\mu =& \frac{\Delta_1}{\Delta_2}, \\
\Delta_1 =& \frac{\lambda_R}{N_R}\sum_{\mathbf{R},\alpha\beta}\text{Re}\Big([\widehat{\mathbf{H}}(\mathbf{R}) - \mathbf{H}^{gt}(\mathbf{R})]_{\alpha\beta}^*\mathbf{S}(\mathbf{R})_{\alpha\beta}\Big) \\
&+ \frac{\lambda_{\mathcal{P}}}{N_{\mathcal{P}}}\sum_{\mathbf{k},\alpha}[\widetilde{\mathbf{H}}_{PP}(\mathbf{k}) - \widetilde{\mathbf{H}}_{PP}^{gt}(\mathbf{k})]_{\alpha\alpha} \\
&+ \frac{\lambda_{\mathcal{Q}}}{N_{\mathcal{Q}}}\sum_{\mathbf{k},\alpha}[\widetilde{\mathbf{H}}_{QQ}(\mathbf{k}) - \widetilde{\mathbf{H}}_{QQ}^{gt}(\mathbf{k})]_{\alpha\alpha} \\
\Delta_2 =& \frac{\lambda_R}{N_R}\sum_{\mathbf{R},\alpha\beta}\mathbf{S}(\mathbf{R})_{\alpha\beta}^*\mathbf{S}(\mathbf{R})_{\alpha\beta} + \sum_{\mathbf{k},\alpha}\frac{\lambda_{\mathcal{P}}}{N_{\mathcal{P}}} + \sum_{\mathbf{k},\alpha}\frac{\lambda_{\mathcal{Q}}}{N_{\mathcal{Q}}}
\end{aligned}
\tag{28}
$$

where $*$ denotes the complex conjugate operation, $N_R, N_{\mathcal{P}}, N_{\mathcal{Q}}$, and $N_{\mathcal{P}\mathcal{Q}}$ denote the total number of Hamiltonian matrix elements corresponding to real space, $\mathcal{P}$ space, $\mathcal{Q}$ space, and their coupling space respectively. $\widetilde{\mathbf{H}}_{PP}^{gt}(\mathbf{k})$ and $\widetilde{\mathbf{H}}_{QQ}^{gt}(\mathbf{k})$ are computed from the ground-truth Hamiltonian $\mathbf{H}^{gt}(\mathbf{k})$ by:

$$
\mathcal{U}(\mathbf{k})^{\dagger}\,\mathbf{H}^{gt}(\mathbf{k})\,\mathcal{U}(\mathbf{k}) = \begin{bmatrix} \widetilde{\mathbf{H}}_{PP}^{gt}(\mathbf{k}) & \mathbf{0} \\ \mathbf{0} & \widetilde{\mathbf{H}}_{QQ}^{gt}(\mathbf{k}) \end{bmatrix}.
$$

It is important to clarify that, in the analytical derivation of $\mu$ in Eq.(27), the real-space contribution can be written in simplified form as:

$$\lambda_R \cdot \text{loss}_H(\mathbf{R}) = \frac{\lambda_R}{N_R} \sum_{\mathbf{R},\alpha\beta} \left[ \left| \left( \widehat{\mathbf{H}}(\mathbf{R}) - \mathbf{H}^{gt}(\mathbf{R}) \right)_{\alpha\beta} \right|^2 + \mu^2 \left| \mathbf{S}(\mathbf{R})_{\alpha\beta} \right|^2 - 2\mu \text{Re}\left( [\widehat{\mathbf{H}}(\mathbf{R}) - \mathbf{H}^{gt}(\mathbf{R})]^*_{\alpha\beta} \mathbf{S}(\mathbf{R})_{\alpha\beta} \right) \right]$$

rather than explicitly retaining the trace supervision term $\text{loss}_T(\mathbf{R})$ like:

$$\lambda_R \cdot \left( (1 - \lambda_C) \cdot \text{loss}_H(\mathbf{R}) + \gamma(\text{loss}_H, \text{loss}_T, \lambda_C) \cdot \text{loss}_T(\mathbf{R}) \right).$$

This simplification is purely at the *algebraic and notational level* and does not imply that $\text{loss}_T(\mathbf{R})$ is omitted. In fact, the balancing factor $\gamma(\text{loss}_H, \text{loss}_T, \lambda_C) = \lambda_C \cdot \text{No\_Grad}\left( \frac{\text{loss}_H(\mathbf{R})}{\text{loss}_T(\mathbf{R})} \right)$ guarantees that this weighted combination of $\text{loss}_H(\mathbf{R})$ and $\text{loss}_T(\mathbf{R})$ is numerically equivalent to $\text{loss}_H(\mathbf{R})$, in which a fixed fraction of the contribution has been substituted by $\text{loss}_T(\mathbf{R})$ in a stable and adaptive manner. In other words, $\text{loss}_T$ serves as a surrogate for a controlled fraction of $\text{loss}_H$, while after normalization the effective value of the entire term remains consistent with $\text{loss}_H(\mathbf{R})$. Therefore, in the derivation of $\mu$, it is sufficient and mathematically consistent to retain only $\text{loss}_H(\mathbf{R})$, while the beneficial regularization effect of $\text{loss}_T$ is still fully incorporated through the design of $\gamma(\cdot)$.

## H    DATASET DETAILS

To construct Materials-HAM-SOC, the first-principles calculations are performed using the Atomic-Orbital Based Ab-initio Computation at USTC (ABACUS)(Li et al., 2016; Lin et al., 2023) package. The Perdew–Burke–Ernzerhof (PBE) exchange-correlation functional (Perdew et al., 1996) and the optimized norm-conserving Vanderbilt (ONCV) fully relativistic pseudopotentials (Hamann, 2013) from the PseudoDojo library (van Setten et al., 2018) are used. Table 2 summarizes the valence electron configurations used in the pseudopotentials and the corresponding numerical atomic orbital (NAO) basis for each element. In self-consistent calculations, the energy cutoff for wave functions is set to 120 Ry and the charge density was converged to a threshold of $1 \times 10^{-6}$. The $\Gamma$-centered Monkhorst-Pack $6 \times 6 \times 6$ $k$-point mesh is employed for self-consistent calculations.

The crystal structures were obtained from the Materials Project database, from which a total of approximately 17,000 nonmagnetic compounds were randomly selected. Among them, 12,000 structures were used for training, 2,000 for validation, and 3,000 for testing. The statistical distributions of atomic species and atomic counts in the training, validation, and test sets are illustrated in Figures 5 and 6. Furthermore, the occurrence frequencies of different elements across the three subsets are presented in Figures 7a–7c.

We visualize representative crystal structures in Fig. 8, highlighting the diversity and broad coverage of our curated dataset **Materials-HAM-SOC**. These samples span a wide range of chemistries, crystal symmetries, and atomic complexities, illustrating the richness of the dataset and its suitability for training universal Hamiltonian prediction models.

Table 2: The valence electron configurations for pseudopotentials and corresponding NAOs of the elements used in this study.

| Element Number | Element Name | Valence Electrons | NAOs | Cutoff Radius |
|---|---|---|---|---|
| 001 | H | $1s^1$ | 2s1p | 7 a.u. |
| 002 | He | $1s^2$ | 2s1p | 7 a.u. |
| 003 | Li | $1s^2 2s^1$ | 4s1p | 7 a.u. |
| 004 | Be | $1s^2 2s^2$ | 4s1p | 7 a.u. |
| 005 | B | $2s^2 2p^1$ | 2s2p1d | 7 a.u. |
| 006 | C | $2s^2 2p^2$ | 2s2p1d | 7 a.u. |
| 007 | N | $2s^2 2p^3$ | 2s2p1d | 7 a.u. |
| 008 | O | $2s^2 2p^4$ | 2s2p1d | 7 a.u. |
| 009 | F | $2s^2 2p^5$ | 2s2p1d | 7 a.u. |
| 010 | Ne | $2s^2 2p^6$ | 2s2p1d | 7 a.u. |
| 011 | Na | $2s^2 2p^6 3s^1$ | 4s2p1d | 7 a.u. |

| Element Number | Element Name | Valence Electrons | NAOs | Cutoff Radius |
|---|---|---|---|---|
| 012 | Mg | $2s^2 2p^6 3s^2$ | 4s2p1d | 7 a.u. |
| 013 | Al | $3s^2 3p^1$ | 2s2p1d | 7 a.u. |
| 014 | Si | $3s^2 3p^2$ | 2s2p1d | 7 a.u. |
| 015 | P | $3s^2 3p^3$ | 2s2p1d | 7 a.u. |
| 016 | S | $3s^2 3p^4$ | 2s2p1d | 7 a.u. |
| 017 | Cl | $3s^2 3p^5$ | 2s2p1d | 7 a.u. |
| 018 | Ar | $3s^2 3p^6$ | 2s2p1d | 7 a.u. |
| 019 | K | $3s^2 3p^6 4s^1$ | 4s2p1d | 7 a.u. |
| 020 | Ca | $3s^2 3p^6 4s^2$ | 4s2p1d | 7 a.u. |
| 021 | Sc | $3s^2 3p^6 4s^2 3d^1$ | 4s2p2d1f | 7 a.u. |
| 022 | Ti | $3s^2 3p^6 4s^2 3d^2$ | 4s2p2d1f | 7 a.u. |
| 023 | V | $3s^2 3p^6 4s^2 3d^3$ | 4s2p2d1f | 7 a.u. |
| 024 | Cr | $3s^2 3p^6 4s^2 3d^4$ | 4s2p2d1f | 7 a.u. |
| 025 | Mn | $3s^2 3p^6 4s^2 3d^5$ | 4s2p2d1f | 7 a.u. |
| 026 | Fe | $3s^2 3p^6 4s^2 3d^6$ | 4s2p2d1f | 7 a.u. |
| 027 | Co | $3s^2 3p^6 4s^2 3d^7$ | 4s2p2d1f | 7 a.u. |
| 028 | Ni | $3s^2 3p^6 4s^2 3d^8$ | 4s2p2d1f | 7 a.u. |
| 029 | Cu | $3s^2 3p^6 4s^2 3d^9$ | 4s2p2d1f | 7 a.u. |
| 030 | Zn | $3s^2 3p^6 4s^2 3d^{10}$ | 4s2p2d1f | 7 a.u. |
| 031 | Ga | $3d^{10} 4s^2 4p^1$ | 2s2p2d1f | 8 a.u. |
| 032 | Ge | $3d^{10} 4s^2 4p^2$ | 2s2p2d1f | 8 a.u. |
| 033 | As | $4s^2 4p^3$ | 2s2p1d | 7 a.u. |
| 034 | Se | $4s^2 4p^4$ | 2s2p1d | 7 a.u. |
| 035 | Br | $4s^2 4p^5$ | 2s2p1d | 8 a.u. |
| 036 | Kr | $4s^2 4p^6$ | 2s2p1d | 8 a.u. |
| 037 | Rb | $4s^2 4p^6 5s^1$ | 4s2p1d | 9 a.u. |
| 038 | Sr | $4s^2 4p^6 5s^2$ | 4s2p1d | 8 a.u. |
| 039 | Y | $4s^2 4p^6 5s^2 4d^1$ | 4s2p2d1f | 8 a.u. |
| 040 | Zr | $4s^2 4p^6 5s^2 4d^2$ | 4s2p2d1f | 7 a.u. |
| 041 | Nb | $4s^2 4p^6 5s^2 4d^3$ | 4s2p2d1f | 7 a.u. |
| 042 | Mo | $4s^2 4p^6 5s^2 4d^4$ | 4s2p2d1f | 7 a.u. |
| 043 | Tc | $4s^2 4p^6 5s^2 4d^5$ | 4s2p2d1f | 7 a.u. |
| 044 | Ru | $4s^2 4p^6 5s^2 4d^6$ | 4s2p2d1f | 7 a.u. |
| 045 | Rh | $4s^2 4p^6 5s^2 4d^7$ | 4s2p2d1f | 7 a.u. |
| 046 | Pd | $4s^2 4p^6 4d^{10}$ | 2s2p2d1f | 7 a.u. |
| 047 | Ag | $4s^2 4p^6 5s^2 4d^9$ | 4s2p2d1f | 7 a.u. |
| 048 | Cd | $4s^2 4p^6 5s^2 4d^{10}$ | 4s2p2d1f | 7 a.u. |
| 049 | In | $4d^{10} 5s^2 5p^1$ | 2s2p2d1f | 7 a.u. |
| 050 | Sn | $4d^{10} 5s^2 5p^2$ | 2s2p2d1f | 7 a.u. |
| 051 | Sb | $4d^{10} 5s^2 5p^3$ | 2s2p2d1f | 7 a.u. |
| 052 | Te | $4d^{10} 5s^2 5p^4$ | 2s2p2d1f | 7 a.u. |
| 053 | I | $5s^2 5p^5$ | 2s2p1d | 7 a.u. |
| 054 | Xe | $5s^2 5p^6$ | 2s2p1d | 7 a.u. |
| 055 | Cs | $5s^2 5p^6 6s^1$ | 4s2p1d | 8 a.u. |
| 056 | Ba | $5s^2 5p^6 5d^1 6s^1$ | 4s2p2d1f | 8 a.u. |
| 072 | Hf | $5s^2 5p^6 6s^2 5d^2$ | 4s2p2d2f | 7 a.u. |
| 073 | Ta | $5s^2 5p^6 6s^2 5d^3$ | 4s2p2d2f | 7 a.u. |
| 074 | W | $5s^2 5p^6 6s^2 5d^4$ | 4s2p2d2f | 7 a.u. |
| 075 | Re | $5s^2 5p^6 6s^2 5d^5$ | 4s2p2d1f | 7 a.u. |
| 076 | Os | $5s^2 5p^6 6s^2 5d^6$ | 4s2p2d1f | 7 a.u. |
| 077 | Ir | $5s^2 5p^6 6s^2 5d^7$ | 4s2p2d1f | 7 a.u. |
| 078 | Pt | $5s^2 5p^6 6s^2 5d^8$ | 4s2p2d1f | 7 a.u. |
| 079 | Au | $5s^2 5p^6 6s^2 5d^9$ | 4s2p2d1f | 7 a.u. |

| Element Number | Element Name | Valence Electrons | NAOs | Cutoff Radius |
|----------------|--------------|-------------------|------|---------------|
| 080 | Hg | $5s^2 5p^6 6s^2 5d^{10}$ | 4s2p2d1f | 7 a.u. |
| 081 | Tl | $5d^{10} 6s^2 6p^1$ | 2s2p2d1f | 7 a.u. |
| 082 | Pb | $5d^{10} 6s^2 6p^2$ | 2s2p2d1f | 7 a.u. |
| 083 | Bi | $5d^{10} 6s^2 6p^3$ | 2s2p2d1f | 7 a.u. |

## I  IMPLEMENTATION DETAILS

Our NextHAM framework is implemented based on PyTorch 2.2.0, E3NN 0.5.6, and CUDA 12.1. The training was conducted on a GPU cluster equipped with NVIDIA A800 GPUs, each with 80 GiB memory.

For the input of the neural network, we adopt a distance of $8.0\,\text{Å}$ to define the neighboring range in the atomic graph. The angular relations between atoms are represented using spherical harmonics with degrees $0 \leq l \leq 5$, while the interatomic distances are encoded through a Gaussian basis expansion (Gong et al., 2023) with a preset base number of 64. The Transformer network consists of 4 stacked basic blocks. Each block contains an E(3)-symmetry layer normalization module, an E(3)-symmetry feed-forward module, an E(3)-symmetry multi-head graph attention module, and a TraceGrad module. For the first three blocks, the internal node features $\mathbf{f}_a^{(\text{node})}$ and edge features ($\mathbf{f}_{ab}^{(\text{edge})}$, $\mathbf{f}_{ab}'^{(\text{edge})}$, and $\mathbf{o}_{ab}^{(\text{edge})}$) are represented in a direct-sum state with a total of 392 channels. In the final block, we apply tensor product and decomposition to $\mathbf{f}_{ab}'^{(\text{edge})}$ and $\mathbf{o}_{ab}^{(\text{edge})}$ to lift the representation to higher angular momentum degrees, constructing tensor representations that correspond to the atomic orbital basis sets up to $4s2p2d1f$. For the TraceGrad module, the constructed O(3)-invariant feature $z_{ab}^{(\text{edge})}$ has a dimension of 256. On the output side, to map the network outputs from the direct-sum E(3)-symmetric tensors into Hamiltonian matrices, we employ the conversion modules provided by Gong et al. (2023), thereby ensuring the exact symmetry of the predicted results with SU(2) symmetry. We employ an ensemble of four sub-models to predict the electronic-structure Hamiltonian. The first sub-model is responsible for predicting the Hamiltonian submatrices formed by atomic pairs with interatomic distances in the range $[0, 1.0\,\text{Å})$, where the case of distance equal to zero corresponds to the on-site Hamiltonian (i.e., the Hamiltonian formed by an atom with itself). The second sub-model handles atomic pairs with distances in the range $[1.0\,\text{Å}, 2.0\,\text{Å})$, the third sub-model covers the range $[2.0\,\text{Å}, 4.0\,\text{Å})$, and the fourth sub-model addresses the range $[4.0\,\text{Å}, 6.0\,\text{Å})$. For atomic pairs with distances greater than $6.0$ Å, we found that their self-consistent Hamiltonian is almost identical to the zeroth-step Hamiltonian numerically. Therefore, for these distant atoms, we bypass the neural network correction step and use the zeroth-step Hamiltonian as the final result.

In the training stage, each card is assigned to one of the sub-models. In our training strategy, electronic states $\leq 10$ eV above the Fermi level are included in the low-energy subspace $\mathcal{P}$, while the remaining states are divided to the high-energy subspace $\mathcal{Q}$. We train the model for a total of 100 epochs on the training set and evaluate the checkpoint that achieves the best performance on the validation set for testing. The hyper-parameters for loss functions are set as $\lambda_C = 0.2$, $\lambda_R = 0.99955$, $\lambda_{\mathcal{P}} = 0.0002$, $\lambda_{\mathcal{Q}} = 0.0001$, and $\lambda_{PQ} = 0.00015$, determined according to the performance on the validation set. We adopt the Adam optimizer with an initial learning rate of $5 \times 10^{-4}$. A warm-up phase of 5 epochs linearly increases the learning rate from $1 \times 10^{-6}$ to the base value, followed by cosine decay to a minimum learning rate of $1 \times 10^{-5}$ by the end of training. To mitigate stochastic variations, we fix the random seed to 1 throughout model training and inference.

## J  BAND STRUCTURE RESULTS

We examine the accuracy and physical reliability of the band structures predicted by our method, by comparing the results obtained from three different Hamiltonians on representative testing samples spanning diverse elements and structures, as illustrated in Fig. 9. The red curves correspond to the ground-truth bands derived from the self-consistent Hamiltonian $\mathbf{H}^{gt} = \mathbf{H}^{(T)}$; the blue curves correspond to the bands obtained from the zeroth-step Hamiltonian $\mathbf{H}^{(0)}$; and the orange curves

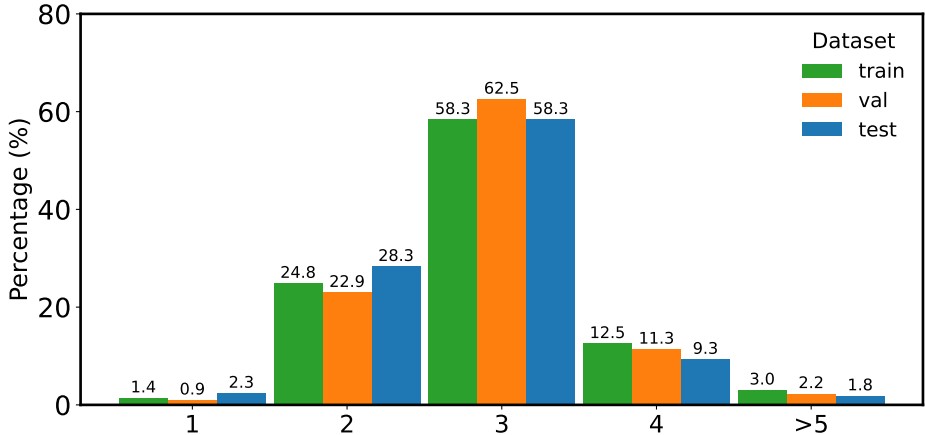

Figure 5: Bar charts of elemental species distributions in the training, validation, and test sets.

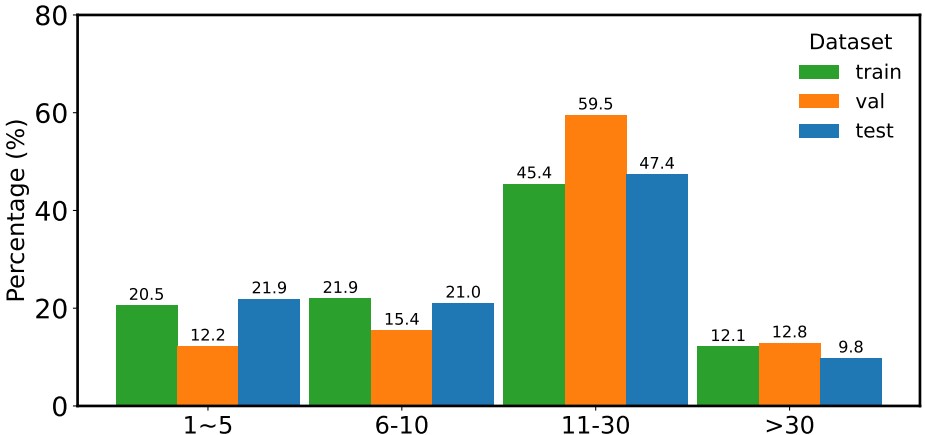

Figure 6: Bar charts of atomic count distributions in the training, validation, and test sets.

represent the bands obtained from the predicted Hamiltonian of our full method, $\widehat{\mathbf{H}} = \mathbf{H}^{(0)} + \widehat{\Delta\mathbf{H}}$. The results show that the zeroth-step Hamiltonian $\mathbf{H}^{(0)}$ provides only a rough sketch of the band structure: it approximately captures the overall positions and qualitative trends of the bands, but suffers from noticeable deviations in curvature and energy levels. In contrast, after applying neural corrections, the predicted Hamiltonian $\widehat{\mathbf{H}}$ yields band structures that align almost perfectly with the DFT ground truth, showing no significant deviations. This striking agreement demonstrates the practical value of our method for materials science and technology, where obtaining accurate band structures is a central problem.

## K    EFFICIENCY COMPARISON BETWEEN OUR METHOD AND DFT

We evaluate the efficiency of **NextHAM** against the conventional DFT workflow on the same Linux server equipped with **Intel(R) Xeon(R) Silver 4114 CPUs@2.20 GHz** and **NVIDIA A800 (80 GiB) GPUs**. All DFT computations are executed on the CPU, while the neural inference of **NextHAM** is evaluated on both CPU and GPU. On the CPU, both DFT and **NextHAM** are run with four CPU cores in parallel. On the GPU, we use four A800 cards: each card executes one neural-network sub-model, and the outputs are aggregated on a single card. The testing batch size is fixed to 1 (no batching). We report the minimum, mean, and maximum wall-clock times across all testing samples.

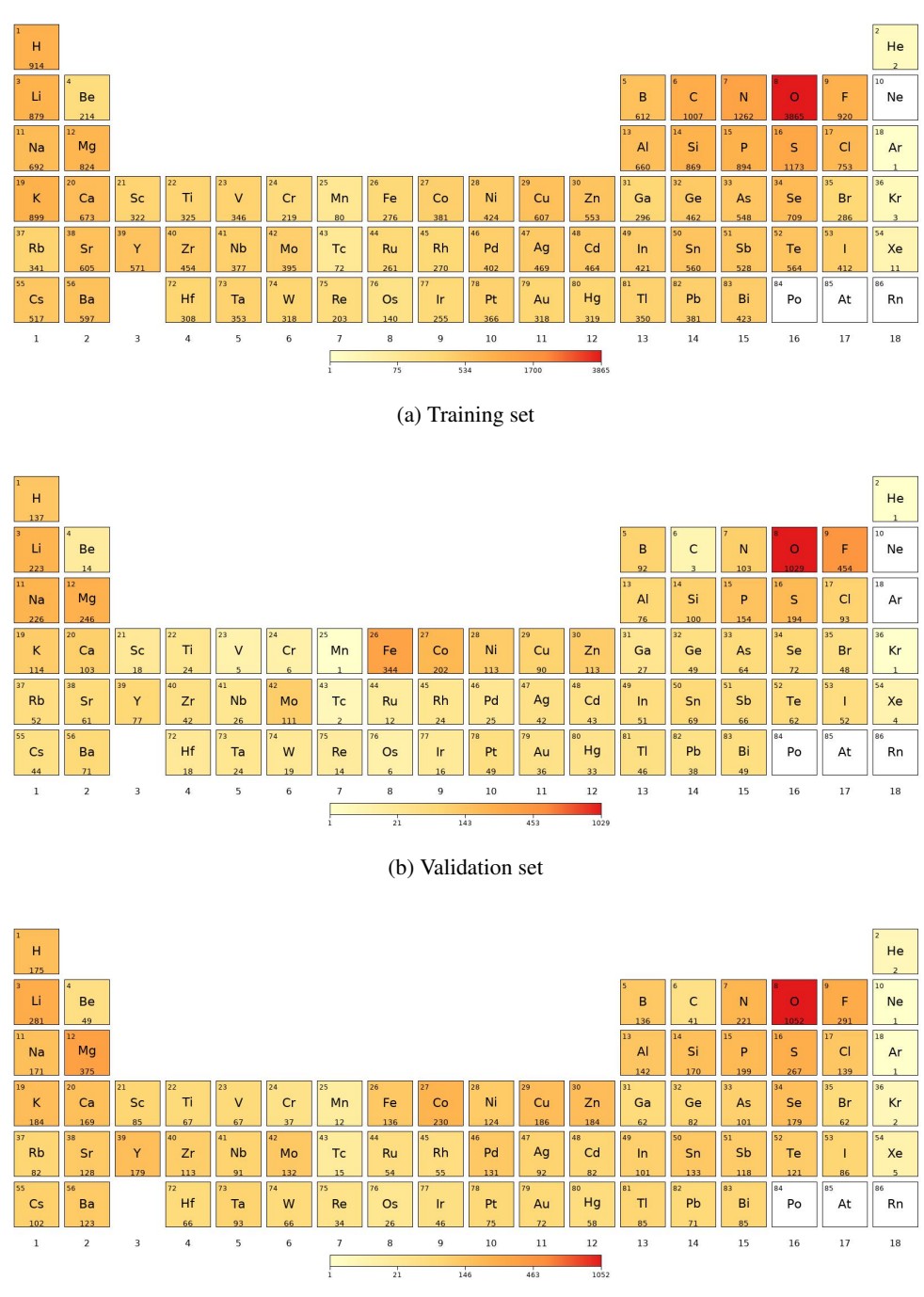

Figure 7: Statistical charts of element occurrence frequencies in the training, validation, and test sets.

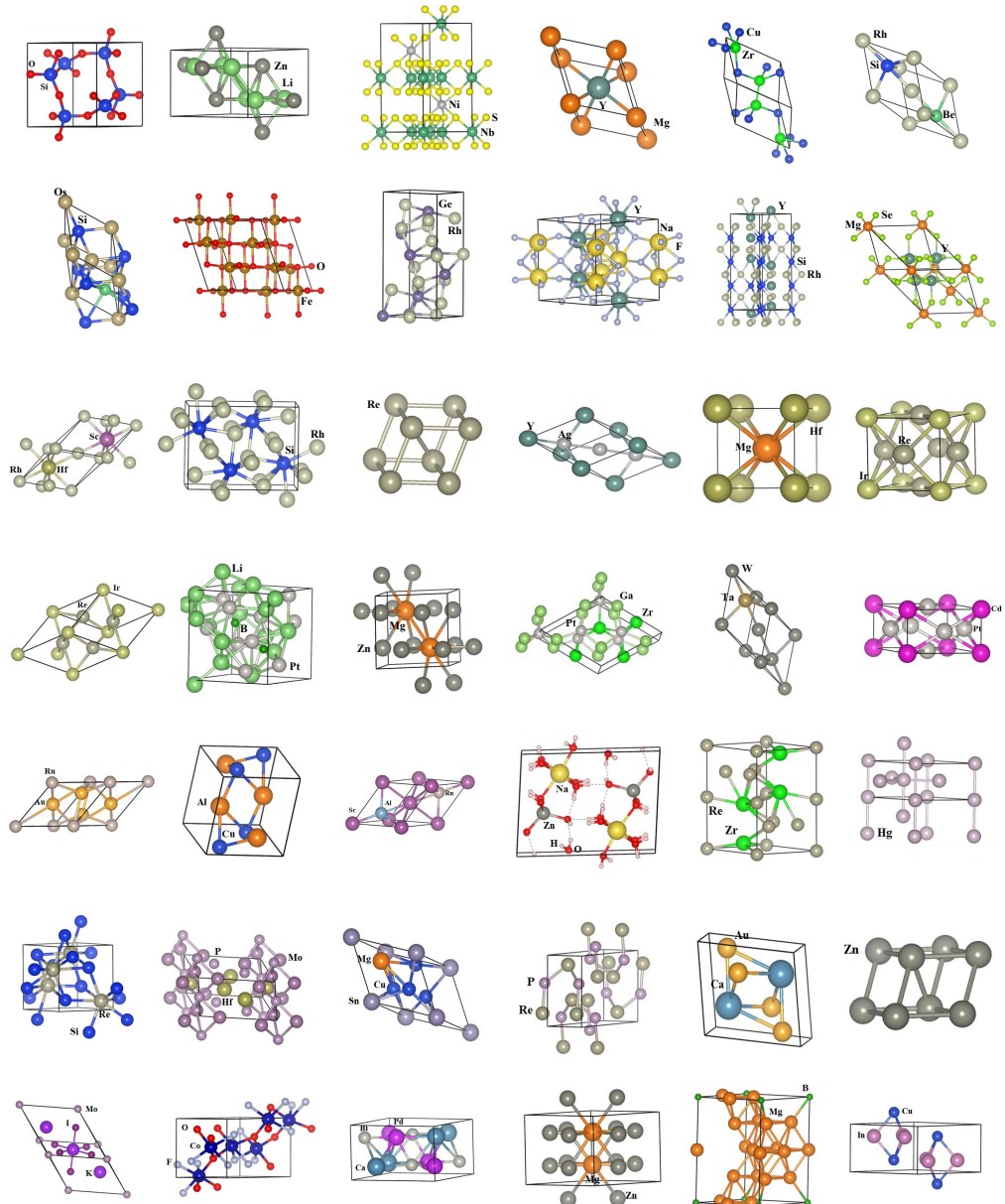

Figure 8: Representative crystal structures sampled from the **Materials-HAM-SOC** dataset. The examples cover diverse chemical compositions, structural patterns, and atomic configurations, demonstrating the dataset's broad coverage across the periodic table. Such diversity ensures that the benchmark provides a comprehensive foundation for training and evaluating universal Hamiltonian prediction models.

Table 3 summarizes the runtime results:

For **DFT**, the entry $\mathbf{H}^{(0)}$**@CPU** (stage 1) includes reading the structural inputs from disk and constructing the zeroth-step Hamiltonian $\mathbf{H}^{(0)}$ from scratch. This stage performs no diagonalization. The entry **SC@CPU** (stage 2) measures the self-consistent (SC) loop that starts from $\mathbf{H}^{(0)}$ and iterates to the converged $\mathbf{H}^{(T)}$ with repeated matrix diagonalizations. Writing the final results to disk is also included in this stage. The entry **Total: $\mathbf{H}^{(0)}$@CPU + SC@CPU** is the total runtime for the **DFT** workflow, i.e., the sum of the two stages.

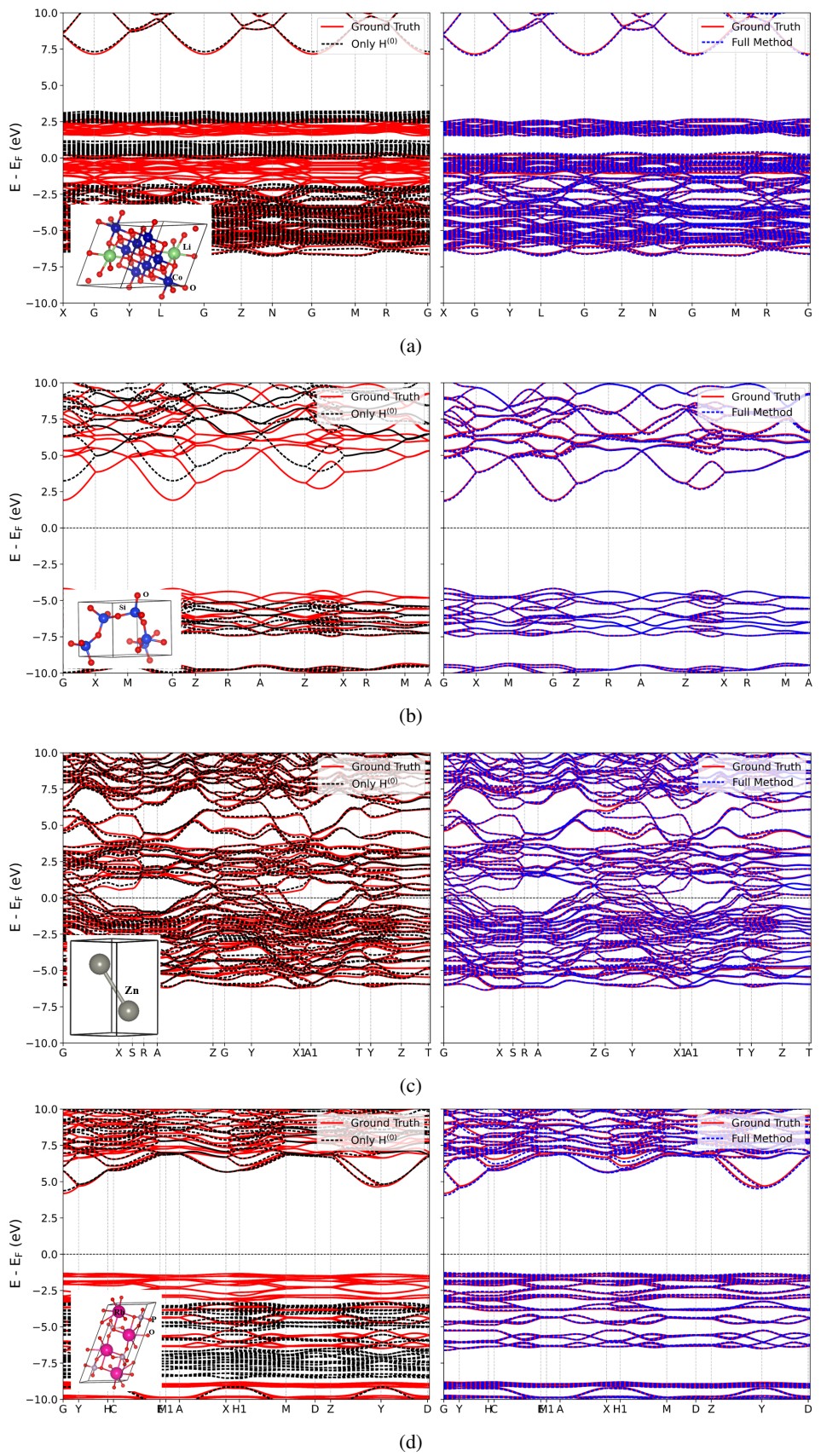

(a)

(b)

(c)

(d)

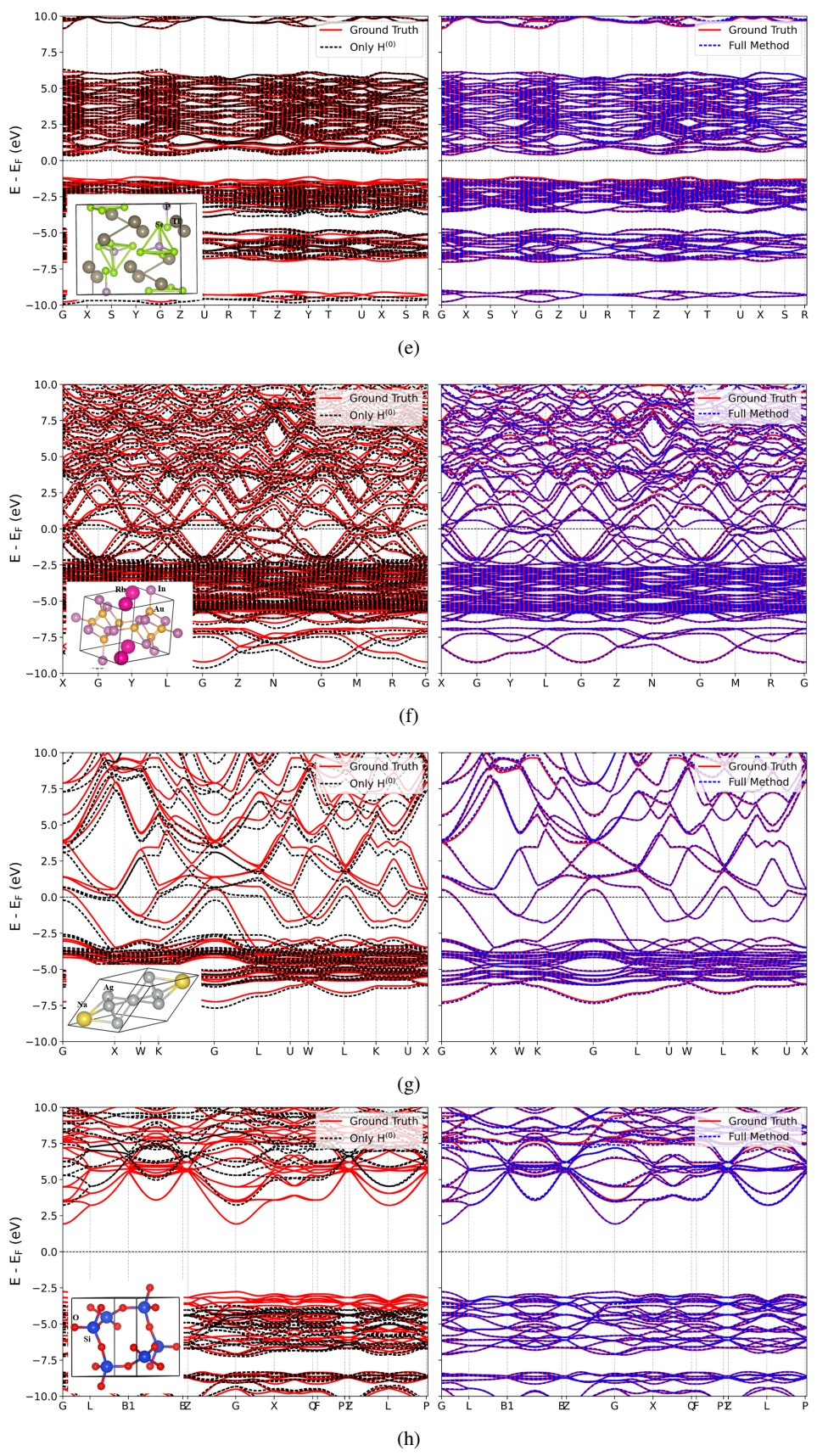

(e)

(f)

(g)

(h)

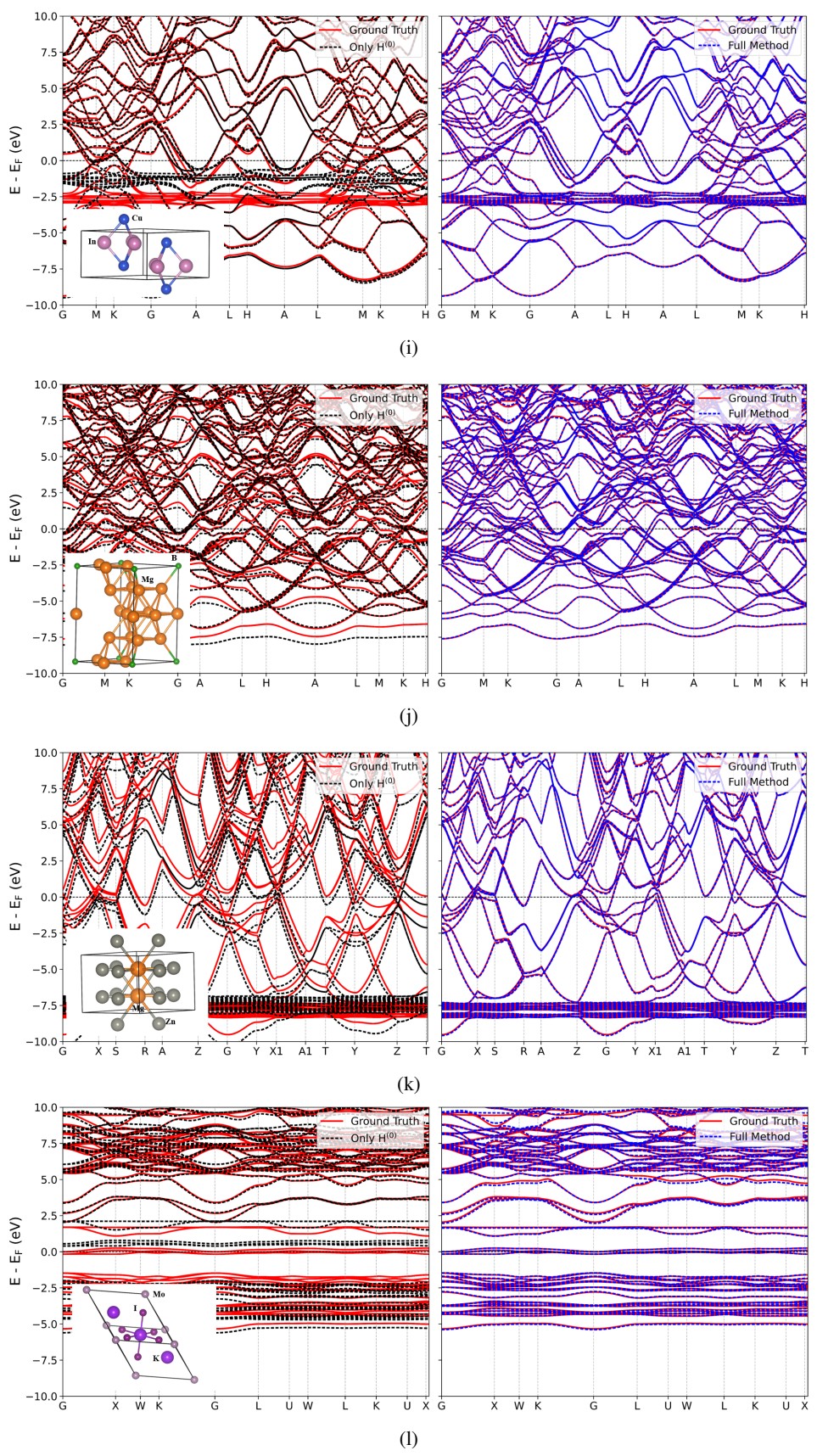

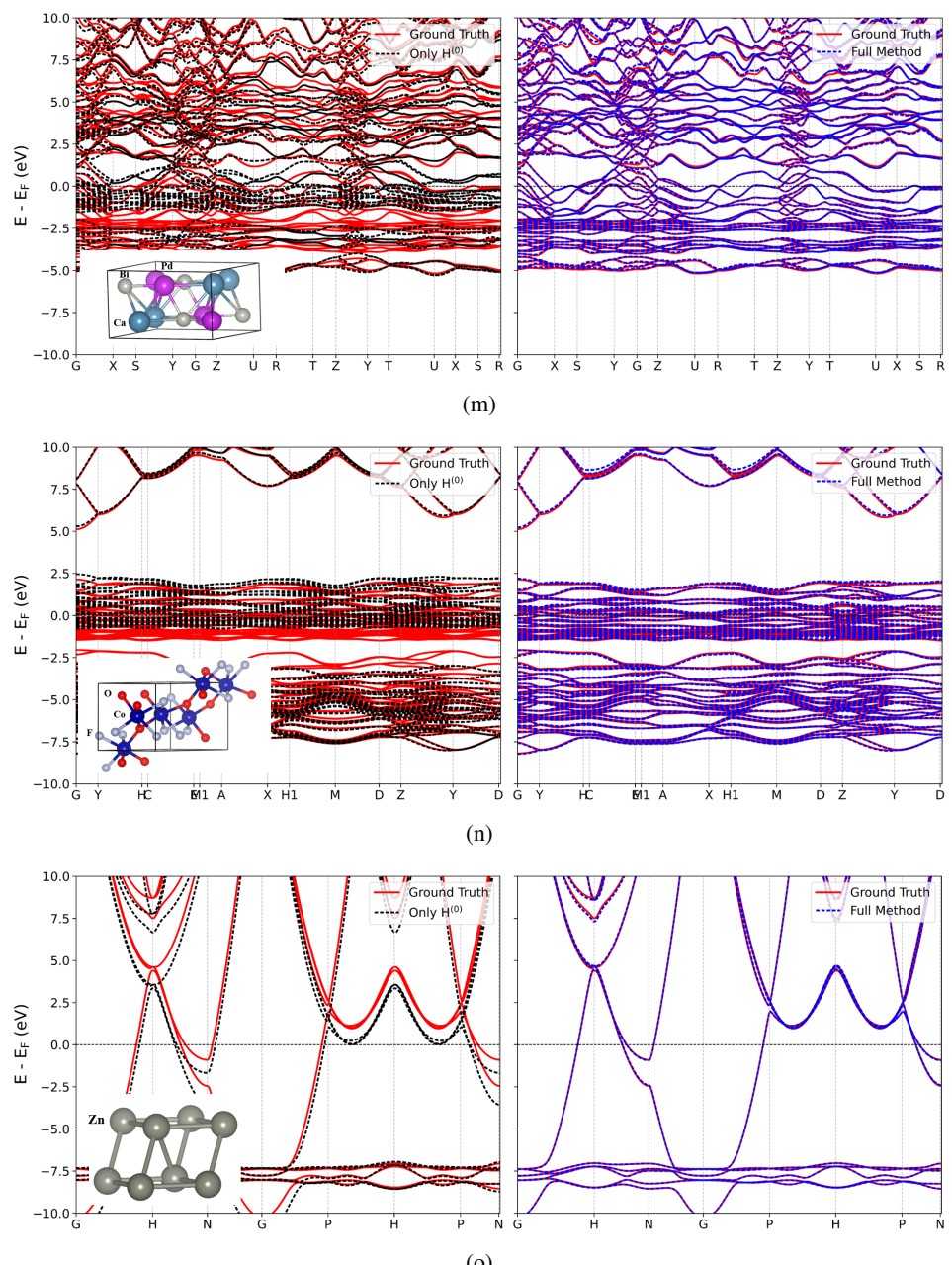

Figure 9: Comparison of band structures obtained from Hamiltonians of representative testing samples. For each subfigure, the left and right panels show different comparisons. In both panels, the red solid curves correspond to the ground-truth bands derived from the self-consistent Hamiltonian $\mathbf{H}^{gt} = \mathbf{H}^{(T)}$. In the left panel, the black dashed curves represent the bands from the zeroth-step Hamiltonian $\mathbf{H}^{(0)}$. In the right panel, the blue dashed curves represent the bands from the predicted Hamiltonian of our full method, $\widehat{\mathbf{H}} = \mathbf{H}^{(0)} + \widehat{\Delta\mathbf{H}}$.

For **NextHAM**, the stage 1, i.e., $\mathbf{H}^{(0)}$@**CPU** has the same meaning as in DFT: the cost of constructing $\mathbf{H}^{(0)}$ from the initial electron density. The stage 2, i.e., **NN@CPU** or **NN@GPU**, covers the full inference workflow after $\mathbf{H}^{(0)}$ is available: loading $\mathbf{H}^{(0)}$ into the model's input tensors, running the neural-network forward pass to predict $\Delta\mathbf{H}$, post-processing the outputs into a DFT-compatible Hamiltonian format, and writing the results to disk. The rows **Total: $\mathbf{H}^{(0)}$@CPU + NN@CPU** and

Table 3: Runtime on the testing set of Materials-HAM-SOC (min/max/mean seconds per sample). All stage timings include the data I/O associated with that stage. Note that the total times are computed per sample as the sum of the corresponding stages; therefore their min/max *need not* equal the sum of the per-stage minima/maxima.

| Method | Stage | Min (s) | Max (s) | Mean (s) |
|---|---|---|---|---|
| **DFT** | $\mathbf{H}^{(0)}$**@CPU** | 3.14 | 742.43 | 55.46 |
| | **SC@CPU** | 16.01 | 28397.45 | 2251.64 |
| | **Total: $\mathbf{H}^{(0)}$@CPU + SC@CPU** | 21.86 | 28617.18 | 2307.11 |
| **NextHAM** | $\mathbf{H}^{(0)}$**@CPU** | 3.14 | 742.43 | 55.46 |
| | **NN@CPU** | 5.15 | 26.92 | 12.62 |
| | **NN@GPU** | 1.16 | 8.95 | 3.01 |
| | **Total: $\mathbf{H}^{(0)}$@CPU + NN@CPU** | 12.69 | 755.84 | 68.08 |
| | **Total: $\mathbf{H}^{(0)}$@CPU + NN@GPU** | 4.84 | 744.66 | 58.47 |

**Total: $\mathbf{H}^{(0)}$@CPU + NN@GPU** report end-to-end runtimes of **NextHAM** with neural inference on CPU or GPU, respectively.

From Table 3, we could observe that, **NextHAM** is substantially faster than the conventional DFT pipeline. Using GPU inference, the mean wall-clock time drops from 2307.11 s (DFT total) to 58.47 s (**97.4**% time reduction). Even with CPU inference, the mean time is 68.08 s (**97.0**% time reduction). In the worst case, the total runtime decreases from 28617.18 s to 744.66 s with GPU inference (**97.3**% time reduction), and to 755.84 s with CPU inference (**97.3**% speedup).

Within the DFT workflow, the self-consistent (SC) stage constitutes the overwhelming majority of the runtime, accounting for **97.6**% of the mean total (2251.64 s out of 2307.11 s) and **99.2**% of the observed maximum (28397.45 s out of 28617.18 s). This observation is consistent with its algorithmic structure: each SC iteration entails dense matrix diagonalizations with computational complexity $\mathcal{O}(N^3)$, leading to an overall cost of $\mathcal{O}(TN^3)$, where $N$ denotes the atom number in a cell and $T$ denotes the number of SC iterations. Since $T$ may be very large and is strongly problem dependent, with no reliable *a priori* bound on convergence, wall-clock times are both substantial and difficult to predict, and the worst-case runtime can be prohibitive. By contrast, **NextHAM** avoids the iterative SC loop entirely. As discussed in previous sections, constructing $\mathbf{H}^{(0)}$ scales with the number of non-zero Hamiltonian elements and is $\mathcal{O}(N^2)$ for small systems, crossing over toward $\mathcal{O}(N)$ for sufficiently large ones; the neural inference follows the same scaling and produces a result in a single forward pass. This one-shot computation makes the runtime more predictable and markedly lower in both mean and worst-case scenarios. Moreover, neural inference benefits strongly from hardware parallelism: switching from CPU to GPU significantly reduces the mean inference time.

It is worth noting that our testing set does not contain very large systems, and the number of non-zero Hamiltonian entries typically scales as $N^2$ (many atoms fall within each other's cutoff). Even in this less favorable sparsity regime, **NextHAM** already delivers the large speedups reported in Table 3. For substantially larger systems, the neighbor count of each atom saturates and the total number of non-zero elements grows only as $\mathcal{O}(N)$, so both $\mathbf{H}^{(0)}$ construction and neural inference become near-linear, while DFT remains $\mathcal{O}(TN^3)$ with an a priori unknown iteration count $T$. Hence the efficiency advantage of **NextHAM** over DFT should increase further at scale as the system becomes larger. We point out that in our current CPU implementation the construction of $\mathbf{H}^{(0)}$ accounts for a large portion of the runtime. Fortunately, this step requires no matrix diagonalization and can be carried out in a highly parallel fashion. In future work, we plan to exploit GPU-based parallel algorithms for $\mathbf{H}^{(0)}$ preparation, which is expected to dramatically reduce this overhead and further amplify the efficiency advantage of **NextHAM**. We leave these works as future work plans.

Overall, the combination of favorable scaling, single-pass prediction (no SC iterations), and efficient GPU parallelization enables **NextHAM** to deliver large speedups across the board, opening a practical path to high-throughput materials simulations.

Table 4: Comparison of Gauge MAE computed in real space (R-space) for different ablation terms and the full method on the testing set of Materials-HAM-SOC. Metrics are averaged over non-zero elements only; entries set to zero due to the truncation distance are masked out. All values are in meV.

| Method | Gauge MAE (meV) |
|---|---|
| Ablation@Input | 1.720 |
| Ablation@Output | 2.974 |
| Ablation@TraceGrad | 1.789 |
| Ablation@Ensemble | 1.862 |
| Ablation@Loss-k | 1.615 |
| Ablation@Loss-PQ | 1.496 |
| **Full Method** | **1.417** |

## L  ABLATION STUDIES

We conduct fine-grained ablation studies for our framework by comparing the following settings. All ablation variants are implemented by removing a single component from the **Full Method** of NextHAM, while keeping all other settings identical, so as to validate the effect of each component:

• **Ablation@Input:** In this ablation term, we replace the zeroth-step Hamiltonians in our input descriptors with conventional atom (node) and atomic-pair (edge) embeddings. Specifically, for an atom $a$ of chemical element $Z_a$, we maintain a learnable 32-dimensional embedding vector $\mathbf{e}_a = \mathbf{e}_{Z_a} \in \mathbb{R}^{32}$, randomly initialized and updated during network training. The embedding of an atomic pair $(a, b)$ is the concatenation of the two element embeddings, $\mathbf{e}_{ab} = [\mathbf{e}_{Z_a}; \mathbf{e}_{Z_b}] \in \mathbb{R}^{64}$.

• **Ablation@Output:** In this ablation term, the residual learning scheme, in which the network predicts the correction term $\Delta \mathbf{H} = \mathbf{H}^{(T)} - \mathbf{H}^{(0)}$, is removed. Instead, the neural network is trained to directly regress the full self-consistent Hamiltonian $\mathbf{H}^{(T)}$, following the setting commonly adopted in existing deep learning approaches for Hamiltonian prediction. This ablation allows us to examine the effectiveness of using $\Delta \mathbf{H}$ as the output target in reducing the complexity of the regression space and improving generalization.

• **Ablation@TraceGrad:** In this ablation term, we remove the TraceGrad mechanism. Concretely, the supervision from the trace quantity is omitted in the loss function, and the gradient-based mechanism that delivers non-linearity from O(3)-invariant features $z_{ab}^{(\text{edge})}$ to induce O(3)-equivariant features via $\frac{\partial z_{ab}^{(\text{edge})}}{\partial \mathbf{f}_{ab}^{\prime(\text{edge})}}$ is also discarded.

• **Ablation@Ensemble:** In this ablation term, we remove the ensemble mechanism based on distance ranges. Instead of training multiple sub-models specialized for different interatomic distance intervals and aggregating their outputs, a single neural network is used to predict all Hamiltonian correction terms across all distance ranges.

• **Ablation@Loss-k:** In this ablation term, we remove the $k$-space loss terms and train the neural network using only the real-space loss, as is commonly used in most of the existing deep learning approaches for Hamiltonian prediction. This setting allows us to assess the contribution of the $k$-space supervision in improving the physical fidelity of the predicted Hamiltonians and the resulting band structures, particularly in eliminating ghost states.

• **Ablation@Loss-PQ:** This variant retains the $k$-space supervision on the intra-subspace blocks ($\mathcal{P}$ and $\mathcal{Q}$) but removes the cross-subspace coupling penalty, i.e., we set $\lambda_{PQ} = 0$. This ablation isolates the role of the $PQ$ term.

• **Full Method** of NextHAM.

We train all of the ablation terms under the same number of epochs, optimizer, and scheduler as the full method (see Appendix I), then evaluate them on the testing set. The $R$-space errors are summarized in Table 4. Beyond $R$-space, because $k$-space is directly tied to downstream quan-

tities (e.g., band structures), we visualize band predictions for **Ablation@Loss-k** (R-space only), **Ablation@Loss-PQ** (setting $\lambda_{PQ} = 0$) versus the **Full Method** in Fig. 10.

From Table 4, the **Full Method** achieves the lowest Gauge MAE. The **Full Method** reduces the error by 17.6%, 52.3%, 20.7%, 23.8%, 12.2%, and 5.2% compared with **Ablation@Input**, **Ablation@Output**, **Ablation@TraceGrad**, **Ablation@Ensemble**, **Ablation@Loss-k**, and **Ablation@Loss-PQ**, respectively. As shown in Fig. 10, **Ablation@Loss-k**, which removes the $k$-space supervision and relies solely on real-space loss, produces band structures with frequent ghost states: in many cases, while most $k$-points are predicted reasonably well, some $k$-points exhibit abrupt and severe deviations from the ground truth—hallmarks of non-physical artifacts. This phenomenon mainly arises from the error amplification effect analyzed in Appendix F, where the large condition number of the overlap matrix can magnify small real-space errors into significant $k$-space deviations. Importantly, such sparse but catastrophic failures cannot be effectively captured by real-space loss alone. **Ablation@Loss-PQ**, which augments the training with $k$-space supervision on the intra-subspace blocks ($P$ and $Q$), demonstrates better performance than **Ablation@Loss-k**, but still fails to completely suppress ghost states. The reason is that unphysical couplings between the low-energy subspace $P$ and the high-energy subspace $Q$ remain unpenalized, and these couplings are precisely what give rise to unphysical artifacts in the band structures. In contrast, the **Full Method** introduces an important penalty on the $PQ$ cross block, which has clear physical significance: for the exact Hamiltonian, $P$ and $Q$ are strictly decoupled, and any spurious $PQ$ couplings in the predicted Hamiltonian are the direct source of ghost states. By explicitly enforcing this decoupling, the $PQ$ loss term addresses the root cause of the artifacts. As a result, the full method produces band structures in excellent agreement with first-principles DFT and free of ghost states. This comparison clearly demonstrates the necessity of our $k$-space loss design, in particular the $PQ$ penalty, for ensuring the physical reliability of predicted band structures.

These results collectively indicate that injecting the physically informed zeroth-step Hamiltonian as an input prior improves generalization, and predicting $\Delta \mathbf{H} = \mathbf{H}^{(T)} - \mathbf{H}^{(0)}$ reduces the effective regression space and eases optimization. They further confirm the effectiveness of the Trace-Grad mechanism: supervising with the trace quantity and propagating non-linearity from invariant to equivariant features enhances representation quality. Notably, this observation aligns with the findings of Yin et al. (2025) on simpler GNN backbones, and our results demonstrate that Trace-Grad remains effective within a Transformer-based framework. Moreover, the ensemble strategy, which partitions the regression space by interatomic distance and aggregates multiple specialized sub-models, yields measurable capacity gains over a single monolithic predictor, highlighting the benefit of distance-dependent specialization. In addition, $k$-space supervision provides complementary guidance that enhances physical fidelity, while explicitly penalizing the cross-subspace coupling ($PQ$) significantly suppresses band structure errors and eliminates unphysical artifacts. In summary, all validated components contribute both individually and synergistically to the overall performance of our method.

## M  COMPARISON WITH RELATED WORK

In the field of electronic-structure Hamiltonian prediction for periodic material systems, one of the most representative methods is DeepH-E3 (Gong et al., 2023). As a pioneering effort, DeepH-E3 has played a crucial role in introducing equivariant networks to the calculation of electronic structures of materials. We aim to perform a clear comparison with it to demonstrate the innovations and advantages of our approach compared with DeepH-E3.

First, DeepH-E3 uses randomly initialized element (node) and element pair (edge) embeddings trained from scratch as input descriptors. However, the sparse nature of these embeddings poses a problem. Since elements are not uniformly distributed in nature, many element pairs appear infrequently, resulting in poorly trained embeddings that struggle with generalization. What's more, this leads to the need to maintain an embedding cost of $\mathcal{O}(M)$ and $\mathcal{O}(M^2)$ for nodes and edges, respectively, where $M$ is the number of elements included in the dataset. As the number of elements increases, this cost grows rapidly, leading to significant memory overhead.

In contrast, our approach addresses these issues by replacing the node and edge embeddings with the zeroth-step Hamiltonian $\mathbf{H}^{(0)}$, which eliminates the space complexity and sparsity problems. $\mathbf{H}^{(0)}$ captures crucial information about the system's electronic structure, embedding the funda-

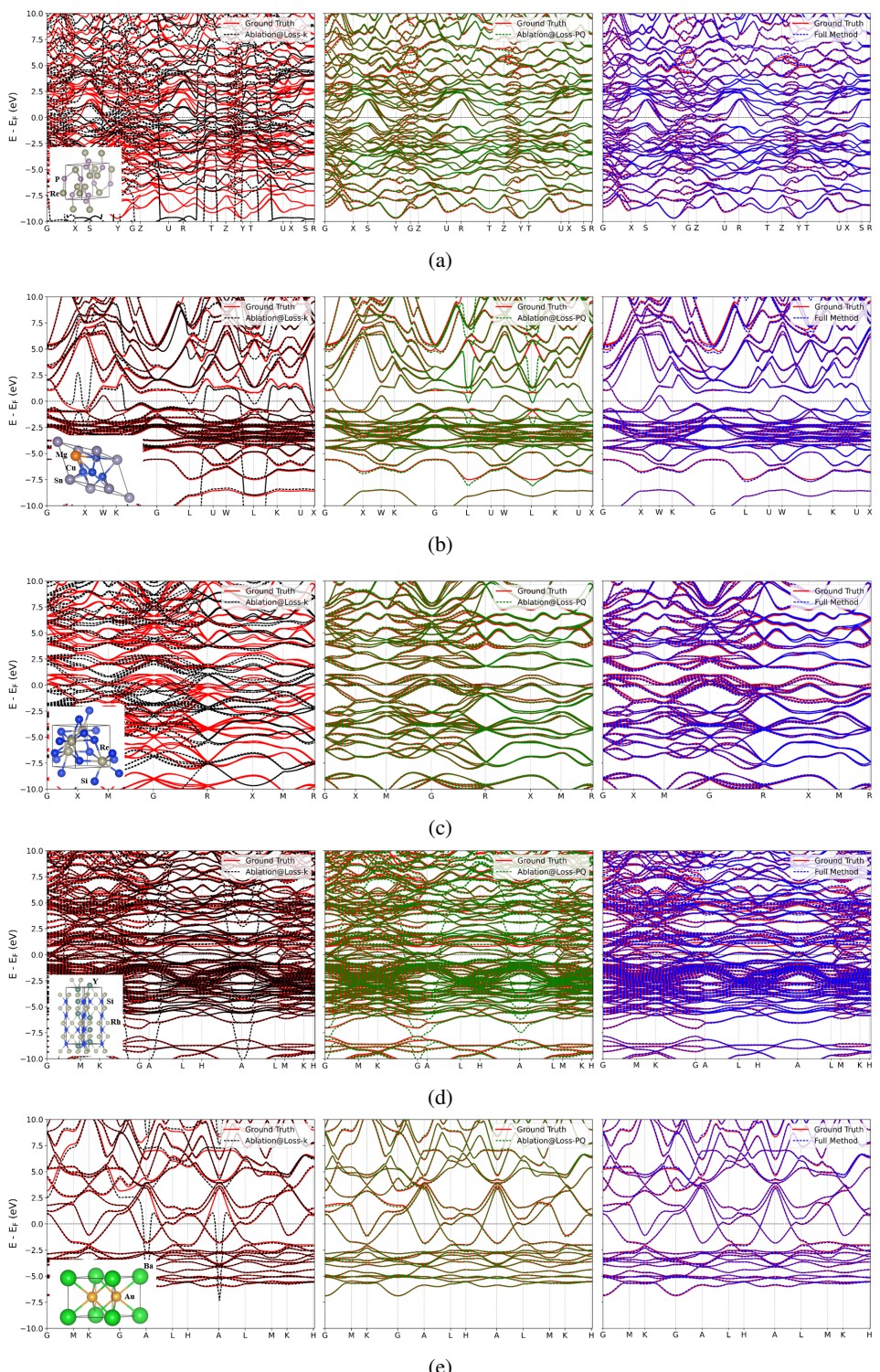

mental characteristics of different elements into a unified representation space, thereby offering a richer physical context. This enables robust generalization across a wide range of material systems, making our approach better suited for constructing a truly universal model. What's more, the space complexity of our input descriptor is independent of the number of distinct elements in the dataset, thus avoiding the exponential cost of embeddings. This use of an easily accessible DFT

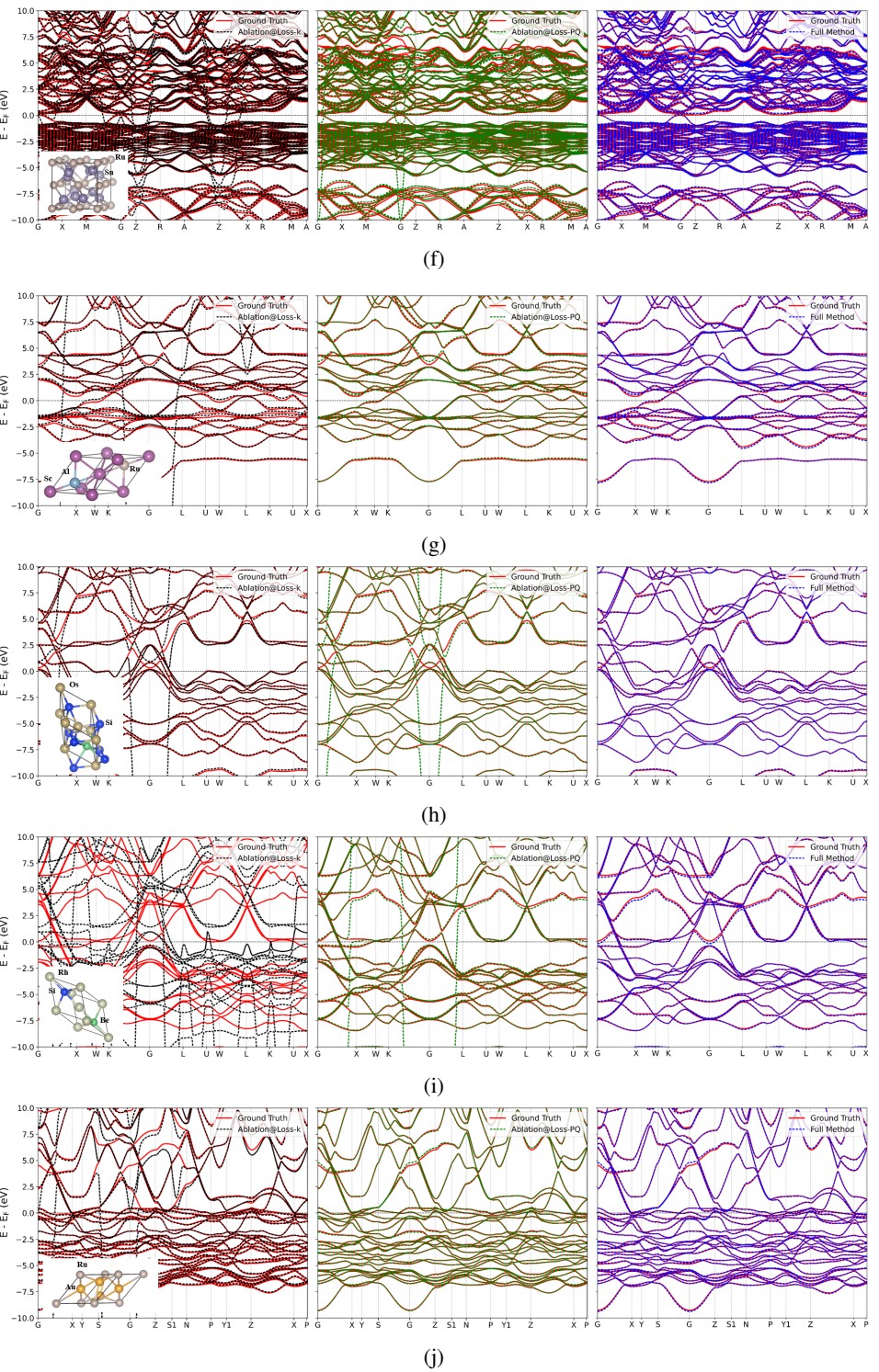

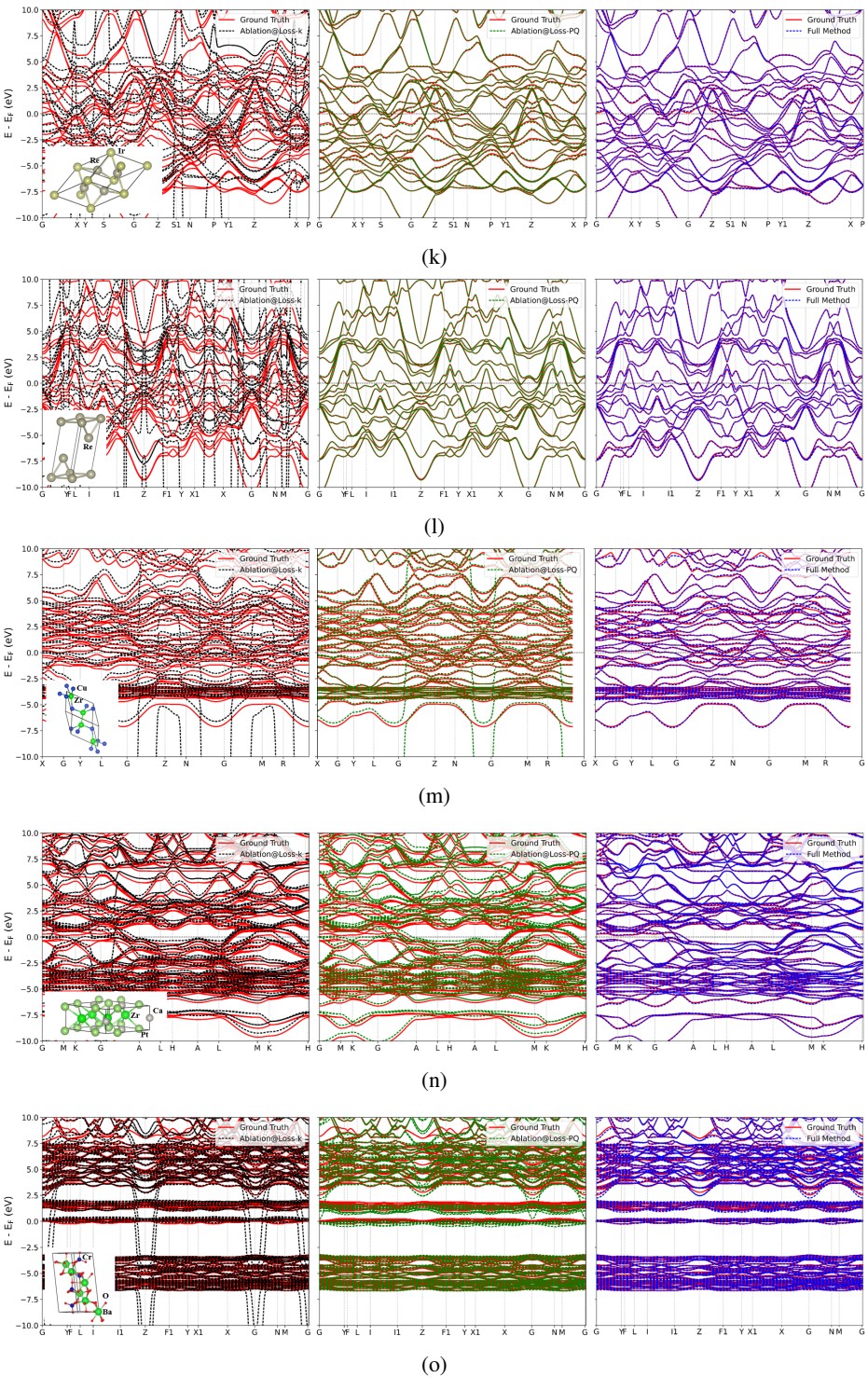

Figure 10: Comparison of band structure performance on representative testing samples. For each subfigure: in all panels, the red solid curves correspond to the ground-truth bands derived from the ground-truth self-consistent Hamiltonians. In the left panel, the black dashed curves represent the band structure results of the ablation term **Ablation@Loss-k**, which exhibit ghost states, i.e., abrupt and severe deviations from the ground truth at certain $k$-points. In the middle panel, the green dashed curves correspond to the results of **Ablation@Loss-PQ**, where such artifacts are mitigated but not fully removed. In the right panel, the blue dashed curves denote the predictions of our **full method**, which successfully eliminates ghost states and achieves excellent agreement with the ground truth.

Table 5: Comparison of R-space errors for DeepH-E3 and our method on the testing set of Materials-HAM-SOC. All values are in meV.

| Method | Gauge MAE |
|---|---|
| **DeepH-E3** | 12.605 |
| **Our Method** | **1.417** |

initial state tensor as a descriptor highlights an emerging direction in AI for Science: embedding physically grounded information directly into machine learning models, which enhances predictive power and provides a general principle that can be broadly applied across scientific ML tasks beyond electronic-structure prediction.

Second, the architecture of DeepH-E3 is relatively simple and lacks sufficient expressive power for building a universal model, which is reflected in two aspects. First, it uses a non-attentive structure, with feature fusion performed through simple average pooling. This may not be well-suited to handle highly complex, diverse structures. Second, it only employs the gating mechanism as the non-linear activation function, without fully exploring the non-linear expressiveness. In contrast, our approach builds upon the TraceGrad paradigm with advanced framework to effectively induce non-linearity from invariant quantities and representations, and develops an E(3)-symmetry Transformer architecture with high non-linear expressiveness. This architecture is capable of dynamically weighting features to adapt to different elements and geometric configurations, making it more suitable for building universal large-scale models.

In addition, our network adopts a delta-learning approach, predicting $\Delta H = \mathbf{H}^{(T)} - \mathbf{H}^{(0)}$ instead of the Hamiltonian itself. This significantly reduces the complexity of the regression target, making it easier for the network to finely fit and generalize complex systems, especially those with heavy atoms and a large number of orbitals. In contrast, DeepH-E3 directly predicts the entire Hamiltonian, which becomes challenging when dealing with more complex scenarios.

Third, DeepH-E3 only uses the R-space loss function for training, while we propose a joint loss function combining both R-space and k-space. As shown in Section 4.3 and Appendix L, the inclusion of k-space loss improves the physical reliability of downstream quantities, such as band structure, and prevents ghost states, which appear as abrupt discontinuities at isolated $\mathbf{k}$-points, caused by the amplification of small R-space errors in k-space (Appendix F). By addressing these issues, our method is better suited for handling a wide range of materials and elements, making it more adaptable and reliable across various scientific applications.

We apply DeepH-E3 to our Materials-HAM-SOC dataset, and follow the same training-validation-testing pipeline on our dataset as described in Appendix H. In the feature layers near the output layer and the output itself, we construct tensor representations that correspond to our dataset's atomic orbital basis sets, which extend to 4s2p2d1f, while leaving the rest of the DeepH-E3 unchanged. These modifications make the comparison more relevant and fair. Unfortunately, even with these adjustments, DeepH-E3 achieves a high R-space error on our testing set, failing to converge to a reasonable error range, as shown in Table 5. In contrast, our method achieves a low error value. This result confirms that DeepH-E3 is not suited for the task of universal electronic structure Hamiltonian prediction, as we have previously analyzed. The research in the DeepH-E3 paper focuses on specialized scenarios, where the training and testing sets consist of perturbations from molecular dynamics simulations of the same material. In contrast, our task involves predicting the Hamiltonian across a broad range of materials, which requires a more generalizable approach that DeepH-E3's specialized design cannot handle effectively.

Furthermore, to compare the performance in specialized scenarios, we conduct experiments on the Monolayer Graphene (MG) and Monolayer MoS$_2$ (MM) datasets, which are released by the DeepH series (Li et al., 2022; Gong et al., 2023), as introduced in Table 6.

For both datasets, we use the same training, validation, and testing sets as those used in DeepH-E3. To ensure fairness, we retrain our method under identical conditions, without pre-training, and measure errors using the classic MAE (Mean Absolute Error) metric, in alignment with DeepH-E3. In addition to DeepH-E3, we also compare our method with the original TraceGrad work (Yin

Table 6: Overview of the Monolayer Graphene (MG) and Monolayer $MoS_2$ (MM) datasets. $m$: number of samples in the current dataset; $a$: number of atoms per unit cell in the current dataset.

| Statistic Types | | MG | MM |
|---|---|---|---|
| Training | $m$ | 270 | 300 |
| | $a$ | 72 | 75 |
| Validation | $m$ | 90 | 100 |
| | $a$ | 72 | 75 |
| Testing | $m$ | 90 | 100 |
| | $a$ | 72 | 75 |

Table 7: Comparison of MAE values among DeepH-E3, the original TraceGrad work, and NextHAM$^{cut-down}$. All values are in meV.

| Dataset | MAE | | |
|---|---|---|---|
| | DeepH-E3 | Original TraceGrad | NextHAM$^{cut-down}$ |
| MG | 0.251 | 0.175 | **0.102** |
| MM | 0.406 | 0.285 | **0.163** |

et al., 2025), which extends the DeepH-E3 backbone network by constructing non-linear equivariant representations from invariant ones.

The version of our method used in these experiments is a cut-down version, denoted as NextHAM$^{cut-down}$. Since the datasets do not provide the $\mathbf{H}^{(0)}$ (zeroth-step Hamiltonian) label, we modify NextHAM by removing the components related to $\mathbf{H}^{(0)}$. Specifically, we do not use $\mathbf{H}^{(0)}$ as an input descriptor. Instead, we use randomly initialized node and edge embeddings, which are trained jointly with the network, similar to those used in DeepH-E3, to represent the elements and their pairwise interactions. The output directly predicts $\mathbf{H}^{(T)}$ rather than $\Delta H = \mathbf{H}^{(T)} - \mathbf{H}^{(0)}$, and because the datasets do not include wavefunction-related data, we retain only the R-space loss function during training, omitting the k-space loss function. What remains is the E(3)-equivariant graph Transformer network architecture developed upon the TraceGrad paradigm. The comparisons are presented in Table 7, from which the results of DeepH-E3 and the original TraceGrad work are from Yin et al. (2025).

These results show that our method, even after cutting out some modules, still dramatically outperforms both DeepH-E3 and the TraceGrad work in prediction accuracy for these material systems. While the original TraceGrad work explores effective methods for constructing non-linear equivariant features, it remains a simple graph network based on average pooling. In contrast, our approach extends the TraceGrad paradigm as a more advanced E(3)-symmetry Transformer architecture that can dynamically weight and adapt non-linear equivariant features, which is more expressive than the simple graph network. In addition to demonstrating strong generalization across a variety of materials in previous sections, these results show that our method excels in specialized systems as well, making it a powerful tool for both broad and focused applications in computational materials science.

## STATEMENTS REGARDING THE USAGE OF LLMS

No large language models were used for this work.

