# OpenReview forum: "Advancing Universal Deep Learning for Electronic-Structure Hamiltonian Prediction of Materials"
_ICLR.cc/2026/Conference — ICLR 2026 Poster_

### Official Review · Reviewer_Yiie · 2025-10-18

**Soundness:** 2
**Presentation:** 2
**Contribution:** 2
**Rating:** 4
**Confidence:** 5

**Summary:**

This paper presents NextHAM, a deep learning framework for predicting electronic structure Hamiltonians. Leveraging zeroth-step Hamiltonians as informative priors, NextHAM employs an E(3)-equivariant transformer for expressive modeling and incorporates a dual-space loss function to ensure both stability and accuracy. Additionally, the paper introduces a new dataset, Materials-HAM-SOC, which includes spin–orbit coupling effects, to facilitate benchmarking.

**Strengths:**

* The paper is well-structured and clearly written, offering an accessible and concise overview of both the problem and the proposed solution.
* The appendix includes a thorough introduction to the relevant DFT background, making the work particularly approachable for readers less familiar with the field.
* The authors open source their code. Moreover, the authors provide a new dataset of material Hamiltonians, which represents a valuable contribution to the open-source community in this domain.

**Weaknesses:**

* The authors propose using the zeroth-step Hamiltonian as input and training the model to predict the correction term—a strategy that essentially follows the well-established delta-learning paradigm in machine learning. This approach has been previously employed in the context of machine-learned density functional theory (DFT), for instance in [1]. While the ablation study demonstrates that this technique effectively reduces prediction errors, it does not introduce significant novelty.
* Although the related work section discusses equivariant approaches, it omits several key references. While this paper focuses on Hamiltonian prediction, the earliest and most influential applications of equivariance in materials science were in machine-learned force fields—foundational works that paved the way for modern equivariant networks. These should be cited; for example, [2-10]. In addition, several important studies [11, 12] on using machine learning to predict Hamiltonians are also missing.
* The evaluation results are primarily demonstrated on the authors’ newly constructed dataset, Materials-HAM-SOC. However, to provide a more comprehensive and convincing assessment of the model’s true performance, comparisons with existing methods on established public benchmarks are essential.
* I greatly appreciate the authors’ inclusion of a detailed introduction to DFT-related background in the appendix, which enhances accessibility. That said, since the proposed model heavily relies on TraceGrad, the current manuscript provides surprisingly little explanation of it. For the sake of completeness and self-containment, I recommend adding a more thorough description of TraceGrad in the appendix, so readers do not need to consult external sources to understand key technical details.

[1] Chen Y, Zhang L, Wang H, et al. DeePKS: A comprehensive data-driven approach toward chemically accurate density functional theory[J]. Journal of Chemical Theory and Computation, 2020, 17(1): 170-181.

[2] Schütt K T, Sauceda H E, Kindermans P J, et al. Schnet–a deep learning architecture for molecules and materials[J]. The Journal of chemical physics, 2018, 148(24).

[3] Batzner S, Musaelian A, Sun L, et al. E (3)-equivariant graph neural networks for data-efficient and accurate interatomic potentials[J]. Nature communications, 2022, 13(1): 2453.

[4] Thomas N, Smidt T, Kearnes S, et al. Tensor field networks: Rotation-and translation-equivariant neural networks for 3d point clouds[J]. arXiv preprint arXiv:1802.08219, 2018.

[5] Musaelian A, Batzner S, Johansson A, et al. Learning local equivariant representations for large-scale atomistic dynamics[J]. Nature Communications, 2023, 14(1): 579.

[6] Gasteiger J, Groß J, Günnemann S. Directional message passing for molecular graphs[J]. arXiv preprint arXiv:2003.03123, 2020.

[7] Gasteiger J, Becker F, Günnemann S. Gemnet: Universal directional graph neural networks for molecules[J]. Advances in Neural Information Processing Systems, 2021, 34: 6790-6802.

[8] Wang Y, Wang T, Li S, et al. Enhancing geometric representations for molecules with equivariant vector-scalar interactive message passing[J]. Nature Communications, 2024, 15(1): 313.

[9] Wang Z, Liu G, Zhou Y, et al. Efficiently incorporating quintuple interactions into geometric deep learning force fields[J]. Advances in Neural Information Processing Systems, 2023, 36: 77043-77055.

[10] Batatia I, Kovacs D P, Simm G, et al. MACE: Higher order equivariant message passing neural networks for fast and accurate force fields[J]. Advances in neural information processing systems, 2022, 35: 11423-11436.

[11] Schütt K T, Gastegger M, Tkatchenko A, et al. Unifying machine learning and quantum chemistry with a deep neural network for molecular wavefunctions[J]. Nature communications, 2019, 10(1): 5024.

[12] Unke O, Bogojeski M, Gastegger M, et al. SE (3)-equivariant prediction of molecular wavefunctions and electronic densities[J]. Advances in Neural Information Processing Systems, 2021, 34: 14434-14447.

**Questions:**

* I greatly appreciate the authors’ effort in performing extensive DFT calculations to provide the open-source community with a new dataset of material Hamiltonians. The paper mentions that “17,000 material structures [were] sampled from the Materials Project.” I am curious about the selection criteria: many of these materials exhibit weak SOC effects. What is the rationale for performing the more computationally expensive SOC-inclusive calculations on such systems? What benefits does this bring, especially when SOC contributions are negligible?
* NextHAM uses the zeroth-order Hamiltonian $H_0$ as model input. The paper states: “These two types of sub-matrices can naturally serve as the initial descriptors for nodes and edges, respectively, in the graph neural network.” While I agree this is a natural choice, I would like to understand more concretely how these equivariant matrices are transformed into node and edge features. Moreover, have the authors considered alternative approaches to constructing $H_0$-based features, such as those proposed in [1] and [2]?
* In the introduction section, the authors note that prior works often “neglect spin–orbit coupling (SOC) effects.” However, to the best of my knowledge, DeepH-E3 [3] already incorporates a method to handle SOC. This makes the introductory claim somewhat inaccurate. More importantly, does NextHAM introduce any specific architectural or algorithmic modifications to better capture SOC effects?
* The paper frames the use of $H_0$ as a way to inject prior knowledge into the model. In the Hamiltonian prediction literature, other forms of prior knowledge have also been explored—for instance, enforcing self-consistency constraints [2,4,5]. It would be valuable to include a discussion comparing different strategies for incorporating physical priors: what are their respective strengths, limitations, and trade-offs in terms of accuracy, computational cost, and generalizability?

[1] Qiao Z, Christensen A S, Welborn M, et al. Informing geometric deep learning with electronic interactions to accelerate quantum chemistry[J]. Proceedings of the National Academy of Sciences, 2022, 119(31): e2205221119.

[2] Wang Z, Liu C, Zou N, et al. Infusing self-consistency into density functional theory hamiltonian prediction via deep equilibrium models[J]. Advances in Neural Information Processing Systems, 2024, 37: 89652-89681.

[3] Gong X, Li H, Zou N, et al. General framework for E (3)-equivariant neural network representation of density functional theory Hamiltonian[J]. Nature Communications, 2023, 14(1): 2848.

[4] Zhang H, Liu C, Wang Z, et al. Self-consistency training for density-functional-theory hamiltonian prediction[J]. arXiv preprint arXiv:2403.09560, 2024.

[5] Li Y, Tang Z, Chen Z, et al. Neural-network density functional theory based on variational energy minimization[J]. Physical Review Letters, 2024, 133(7): 076401.

---

> ### Author Response · Authors · 2025-11-25
> **Response to Reviewer Yiie (Part1)**
>
> We would like to sincerely thank the reviewer for their valuable time and effort in reviewing our manuscript. Their thoughtful comments and constructive feedback are greatly appreciated and have helped us improve our presentation.
>
> 1. The reviewer mentioned that delta-learning is an established paradigm in the deep learning field, which we agree with. In the revised paper, we explicitly highlight that our use of $ \mathbf{H}^{(0)} $ as the output starting point to predict the correction term is inspired by existing works. However, unlike these existing works that use the initial guess of DFT solely as a starting point for the output, our approach introduces a significant innovation by using the zeroth-step Hamiltonian not only as an output starting point but also as an input descriptor. Unlike existing methods that rely on randomly initialized atom and edge embeddings lacking physical prior knowledge, $ \mathbf{H}^{(0)} $ efficiently encodes essential information about the system's electronic structure, providing richer physical context by embedding the intrinsic characteristics of diverse elements into a unified representation space, thereby enabling robust generalization across chemically complex material systems. Its effectiveness has been fully validated in our paper, including significant reductions in test errors and the ability to generalize to out-of-distribution elements. This is demonstrated in Section 4.2 and Appendix K.
>
>     Additionally, most of the existing electronic structure prediction works using delta-learning focus on molecular systems without considering the SOC (Spin-Orbit Coupling) effect. In contrast, our work is dedicated to the electronic structure prediction for periodic material systems with explicit modeling of the SOC effect. In this context, we make an important discovery: directly using the matrix elements of $ \mathbf{H}^{(0)} $ achieves sub-µeV level accuracy for spin-off-diagonal blocks. **This discovery, which is a very interesting finding, has not been reported or observed in prior studies. We believe that achieving SOC predictions with an accuracy of μeV level holds significant value.** Small differences in SOC energy can greatly impact the electronic structures, topological properties, and quantum behaviors of materials. For instance, in recent papers [1], it is noted that in bilayer graphene (BLG), SOC opens a small gap of tens of μeV in the low-energy bands, which plays a crucial role in spin relaxation and decoherence. This example demonstrates that even small SOC energy differences can have notable effects, making μeV precision essential for accurate predictions in these fields. Moreover, our method achieves this high precision while maintaining computational efficiency, representing a significant improvement over traditional approaches. In conclusion, μeV SOC precision is valuable for material design, quantum computing, and topological material research, offering important insights for both theory and experiments.
>
>     [1] Electric-field-independent spin-orbit-coupling gap in h-BN-encapsulated bilayer graphene. Physical Review Applied, 2025.
>
> 2. In accordance with the reviewers’ suggestions, we have incorporated the additional references.
>
> 3. We appreciate the request for comparisons on established public benchmarks. However, the current open datasets (e.g., those used by DeepH-E3) target narrowly scoped, system-specific settings (such as MoS2 or Bi2Se3 under simple deformations) rather than the universal scenario we pursue. Our objective is a universal Hamiltonian predictor that generalizes across diverse materials. In line with trends seen across all of the AI fields, universal models are valued for broader applicability and better transfer: they cover a wide range of scenarios without retraining from scratch, and when needed, small-data few-shot fine-tuning usually suffices, crucial for rare elements or sturctures where large datasets are infeasible.
>
>     **We also point out a limitation that the known effort toward universal Hamiltonian predictor of materials, such as DeepH-2 [2], has not released code or data, preventing strict same-condition reproduction. To overcome this barrier and enable rigorous, fair benchmarking of universal approaches, we release our data, code, and pretrained models.** This creates a reproducible, comparable benchmark for the community and lowers the entry cost for future universal models to be evaluated and improved. Our goal is that, for most materials, the universal model can be used directly; for cases requiring extreme accuracy, small-scale fine-tuning on top of our pretrained model suffices. This path minimizes usage cost, scales to rare elements, and provides a practical foundation for broad deployment.
>
>     [2] Universal materials model of deep-learning density functional theory Hamiltonian. Science Bulletin 2024.

---

> ### Author Response · Authors · 2025-11-25
> **Response to Reviewer Yiie (Part 2)**
>
> 4. Following the reviewer’s suggestion, we have added a thorough description of TraceGrad in Appendix D of the revised paper. To further aid understanding, we also added Appendix B, which provides a self-contained primer on the group-theoretic concepts essential for TraceGrad and for our overall method.
>
> 5. We thank the reviewer for the positive comments on our open-source dataset. Regarding the concern about including SOC for materials with intrinsically weak SOC effects, we note that our sampling strategy indeed prioritizes structures containing heavy elements, where SOC plays a significant role. However, SOC is a fundamental intrinsic interaction that, in principle, exists in all materials, and including it ensures a consistent and unified representation of Hamiltonians across the entire dataset. Moreover, by employing the zeroth-step Hamiltonian $ \mathbf{H}^{(0)} $  scheme, the computational cost of SOC-inclusive calculations is comparable to that of non-SOC calculations in our workflow. Therefore, adding SOC does not introduce additional computational burden. At the same time, it enhances the model’s ability to learn Hamiltonians across both strong-SOC and weak-SOC regimes, improving generality and applicability.
>
> 6. We have added details on how the sub-matrices of the zeroth-step Hamiltonian are transformed into initial node and edge descriptors in Section 2.1.1 and 2.1.2 of the revised paper.
>
> 7. The reviewer is correct that most prior works neglect SOC, and DeepH-E3 is a rare exception. We have now explicitly acknowledged this in Appendix C of the revised paper. However, DeepH-E3’s treatment of SOC and the scope of its applicability differ fundamentally from ours. DeepH-E3 directly regresses the SOC blocks. In contrast, NextHAM adopts a delta-learning strategy.  In this context, we have made a significant new discovery not reported by existing works. Specifically, $ \mathbf{H}^{(0)} $ and $ \mathbf{H}^{(T)} $ are remarkably close in the spin-off-diagonal block of SOC Hamiltonian blocks of materials. For this part of the block, directly using the matrix elements of $ \mathbf{H}^{(0)} $ achieves accuracy at the sub-µeV level.
>
>     Moreover, DeepH-E3 is designed for narrow, system-specific scenarios (e.g., MoS2, Bi2Se3 under simple deformations). In contrast, our work targets universal electronic-structure Hamiltonian prediction across diverse periodic materials with explicit SOC, spanning more than 60 elements. **Through carefully designed descriptors, network structures, and loss functions, we achieve high accuracy under such a broad and heterogeneous material space.** This achievement is substantially more challenging and cannot be compared directly with DeepH-E3’s restricted setting.
>
> 8. The reviewer asks what unique advantages using $ \mathbf{H}^{(0)} $ as an input provides over other method leveraging physics-informed priors. A key benefit is, $ \mathbf{H}^{(0)} $ empowers the model to generalize to elements that are not seen during training, introducing a novel out-of-distribution generalization capability. This is because $ \mathbf{H}^{(0)} $ is a physically grounded operator that encodes cross-element regularities of electronic structure, rather than relying on randomly initialized per-element embeddings. As shown in **Figure 7** of the paper, our model is trained on a training set that does not include structures containing the element Neon (Ne). However, in the test set, a Ne-containing structure is included. Despite Ne being unseen during training, the model is able to generalize well to this new element, with testing error remaining very small, i.e., 0.1 meV for the R-space MAE, as reported in **Figure 4**. This demonstrates the model's ability to extrapolate the knowledge it learned from seen elements to predict properties for unseen elements, showcasing its ability to generalize beyond the training data.. In contrast, traditional methods that rely on randomly initialized embeddings for each element cannot generalize to unseen elements that are not included in the training set.

---

> > ### Comment · Reviewer_Yiie · 2025-11-25
> >
> > Thanks for the authors' response. Most of my concerns are addressed, so I will increase my score.

---

> > > ### Author Response · Authors · 2025-11-29
> > >
> > > I sincerely appreciate your positive feedback and your decision to raise the rating to 6.

---

### Official Review · Reviewer_hRHV · 2025-10-18

**Soundness:** 3
**Presentation:** 2
**Contribution:** 2
**Rating:** 4
**Confidence:** 4

**Summary:**

This paper focuses on predicting the hamiltonian matrix of materials. There are four core contributions that I understood:

- Using H0 as an initial guess and predicting the difference
- Incorporating the TraceGrad architecture (although I'm not sure if this is intended to be a novelty, since it seems to be already published).
- Incorporating the coupling between occupied and unoccupied states of the reciprocal Hamiltonian into the loss
- A new dataset of material hamiltonians

**Strengths:**

I think that predicting the difference between H0 and the final Hamiltonian is an interesting approach. However, as pointed out by the authors in the appendix, it comes at a formidable computational cost (55s on average for structures which seem to have between 5 and 20 atoms, going by Fig. 7).

I also appreciate that this network accounts for SOC (the only other one which does so to my knowledge is DeepHE3)

**Weaknesses:**

My primary concerns are (1) I think there have been simpler ways to do what the authors are doing with an otherwise complex architecture and loss function, (2) without either performance comparisons or applications (eg, scf iteration reduction), there is no way to judge the utility of their model, and (3) I question the universality and utility of this dataset of 12,000 structures, seemingly containing fewer than 10 instances of several atomic elements. I elaborate on these points below:

(1):

A lot of the components of this model (use of the trace as a supervision signal, initializing the features with H0, predicting the refinement of H) seem to be ultimately reducing the numerical range of the data (as mentioned in the paper). Other works have tackled the same problem by simply scaling the trace out of the data [1, 2], similar to what is commonly done in MLIPs. I'm not yet convinced that the architectural complications introduced has a benefit over simply scaling the data and using a regular equivariant network like DeepHE3 (which also treats SOC) or QHNet.

Other overly complex sections of the architecture include the sub-models for various radii. I think this kind of hard-coding defeats the purpose of either learning the correct attention mechanism, or simply learning the coulomb decay from data. Does this improve generalizability, compared to a model with the same number of parameters?

Loss function: I understand that the near-ill-conditioning of the overlap matrix can amplify small errors in matrix elements. However, I believe this is effectively treated by Ref [3] in the wavefunction alignment loss. Can the authors discuss the computational cost and scaling of their loss function, and compare to using the wavefunction alignment loss? Could the effective separation between occupied and unoccupied subspaces be recovered by using a simpler subspace loss?

(2):

I'm not sure how to evaluate the performance of this model. Matrix-element MAE errors are meaningless when introducing a new dataset, especially one which covers multiple atomic elements since heavier elements will have larger-valued elements. The authors mentioned that off-diagonal matrix element errors are "sub-meV" - but these matrix elements are typically an order of magnitude lower than diagonal matrix elements already, so this is expected. MAE matrix element errors are only useful for comparing on benchmarks with other models.

Examining the provided bandstructures in the appendix, and comparing them to those computed by methods like Wannierization, I think the errors seem to be fairly high. In many cases, a rigid shift from the bandstructures computed from H0 looks like it could recover the learned solution.

(3):

Several key details about the dataset are missing, such as the basis set used, and the distribution of the number of atoms per structures (the authors mention it does not contain large systems, but this is not quantified). My current understanding based on Fig 7 is that there are 5-20 atoms per structures, and the corresponding Hamiltonian matrices have sizes on the order of 100s.

In general, Hamiltonian datasets can contain fewer structures due to the increased amount of information-per-structure. However, I am questioning the utility of 12,000 training structures distributed across 68 elements. For example, looking at Fig 6a in the appendix, there seems to be on the order of 1 atom of several noble gases in the training data. I don't know if this can really be considered a representative dataset of "68 atomic elements". I think it would ultimately be more useful to focus on a smaller range of elements with better sampling across them.

[1] Equivariant Electronic Hamiltonian Prediction with Many-Body Message Passing, https://arxiv.org/html/2508.15108v1
[2] GEARS H: Accurate machine-learned Hamiltonians for next-generation device-scale modeling, https://arxiv.org/abs/2506.10298
[3] Enhancing the Scalability and Applicability of Kohn-Sham Hamiltonians for Molecular Systems, https://arxiv.org/abs/2502.19227

**Questions:**

What is the computational complexity of this loss function?

---

> ### Author Response · Authors · 2025-11-25
> **Response to Reviewer hRHV (Part 1)**
>
> Thanks for the time and effort spent on evaluating our work. However, **we find there have been many misunderstandings about our approach**, and we would like to provide clarifications.
>
> 1. The reviewer mentioned that introducing $ \mathbf{H}^{(0)} $ might slow down the computation. However, the construction of $ \mathbf{H}^{(0)} $ does not affect the asymptotic complexity of the neural network. Specifically, the complexity is $\mathcal{O}(N^2)$ for small systems with $N$ atoms, and it asymptotically approaches $\mathcal{O}(N)$ for sufficiently large systems as the neighbor count saturates for each atom. This scaling behavior is consistent with the message passing mechanism of graph neural networks, ensuring that incorporating the zeroth-step Hamiltonian as an additional input descriptor does not worsen the asymptotic complexity.
>
>     In contrast, DFT methods have a computational complexity of  $\mathcal{O}(T N^3)$. Therefore, as the system size increases, the time cost for constructing $ \mathbf{H}^{(0)} $ becomes negligible compared to the total cost saved from avoiding DFT calculations, making it practically insignificant for large systems.
>
> 2. First, the reviewer has mentioned that some of the components of our method might not be as effective as simpler frameworks. However, the effect of each component in our method has been comprehensively demonstrated in this paper, as detailed in Section 4 and Appendix K. Here, we would like to highlight several significant achievements and discoveries:
>    - The introduction of $ \mathbf{H}^{(0)} $ as a descriptor enables out-of-distribution generalization capabilities (Section 4.2).
>    - $ \mathbf{H}^{(0)} $ at the output stage can achieve sub-µeV accuracy in the spin-off-diagonal block under SOC (Spin-Orbit Coupling) conditions.
>    - The introduction of k-space loss effectively eliminates ghost states, leading to a fundamental breakthrough in the accurate and reliable prediction of band structures and downstream physical quantities, such as optical conductivity (Section 4.3, Appendix K). **These achievements showcase the effectiveness of our proposed method in practical applications. This addresses the reviewer's concerns regarding the value and applicability of our method.**
>
>     The reviewer has made comments regarding the comparison with existing works, including papers [1-5]. However, we believe there may be some misunderstandings. **Paper [1]**, which was submitted to arXiv in August 2025, was released nearly simultaneously with our work and should not be considered in the comparison. Additionally, while Paper [1] focuses on incorporating complex many‑body expansions and high angular-momentum orbital blocks for specific systems like twisted bilayers materials, it lacks the broader focus on generalizing across diverse material systems, a key feature of our approach. **Paper [2]** focuses on a specific system, WSe2 interfaced with Ni-doped amorphous HfO2, which differs from our approach that generalizes across diverse material systems. Similarly, **DeepH-E3** [3] primarily addresses specific materials, such as MoS2 and Bi2Se3 under simple deformations, and lacks generalization across various systems, making direct comparison difficult. **QHNet** [4], focused on molecular systems with a limited set of elements (C, H, O, N, F). In summary, these approaches typically focus on a narrow range of materials or molecular systems. They develop specialized models, whereas we are developing a universal model, which represents a different technical approach.
>
>     **To the best of our knowledge, there are only very few methods in the literature that target on universal electronic-structure Hamiltonian prediction for material systems. Although the DeepH-2 method [5] is a representative effort for this task, it is not open-source, making it difficult to replicate or compare directly. In contrast, we have made our data, code, and pretrained models publicly available, creating a reproducible benchmark for future work. This openness will enable the broader research community to validate and compare with our work, fostering further advancements in the field. Given the lack of open-source, generalizable models in this area, we believe our contribution significantly advances the field, offering a valuable resource for continued research.**
>
>     [1] Equivariant Electronic Hamiltonian Prediction with Many-Body Message Passing.
>
>     [2] GEARS H: Accurate machine-learned Hamiltonians for next-generation device-scale modeling.
>
>     [3] General framework for E(3)-equivariant neural network representation of density functional theory Hamiltonian. Nature Communications 2023.
>
>     [4] Efficient and Equivariant Graph Networks for Predicting Quantum Hamiltonian. ICML 2023.
>
>     [5] Universal materials model of deep-learning density functional theory Hamiltonian. Science Bulletin 2024.

---

> ### Author Response · Authors · 2025-11-25
> **Response to Reviewer hRHV (Part 2)**
>
> 3. The reviewer mentions that **Paper [6]** introduces a wavefunction-based loss to improve the prediction of downstream physical quantities. However, this paper targets molecular systems, whereas our work focuses on periodic materials with spin-orbit coupling (SOC) effects. As a result, **the mathematical formulation is also different.** Compared with **Paper [6]**, which employs a wavefunction-based loss as a constraint, we extend this idea to periodic solid-state systems with SOC. Additionally, we incorporate the coupling interaction between the low-energy and high-energy subspaces, $\widetilde{\mathbf{H}}^{\mathrm{gt}}_{PQ}(\mathbf{k})$, which we found plays an important role during training and in eliminating ghost states in band structures and the resulting downstream quantities. Additionally, we obtain the analytical solution to address the gauge freedom problem in our loss function, as detailed in Appendix F.
>
>     [6] Enhancing the Scalability and Applicability of Kohn-Sham Hamiltonians for Molecular Systems. ICML 2025.
>
> 4. The reviewer questioned the effectiveness of our distance-partitioned sub-model ensemble method and asked whether simply increasing the model size would achieve similar performance. To address this concern, we conducted controlled comparison experiments on our **Materials-HAM-SOC** dataset under *exactly the same total parameter budget*.
>
>     Specifically, we compared:
>     - **(a)** Enlarging a single model by increasing depth  ($4×$depth) (no ensemble);
>     - **(b)** Using a **4-model ensemble** based on atomic-distance partitioning (no increase in depth per model). **This corresponds to our method.**
>
>     The resulting R-space MAEs are:
>     - **1.730 meV** — Single model with 4× depth
>     - **1.417 meV** — 4-model distance-based ensemble **(This corresponds to our method).**
>
>     This clearly shows that, under equal parameter budgets, the distance-partitioned ensemble significantly outperforms simply scaling up a single model. The underlying reason is that Hamiltonian matrix elements vary dramatically across different interatomic distances, often spanning multiple orders of magnitude. A single model struggles to simultaneously fit matrix elements across such a wide dynamic range, whereas the distance-based divide-and-conquer ensemble effectively decomposes the learning task into more homogeneous subproblems. This leads to substantially improved accuracy and stability.

---

> ### Author Response · Authors · 2025-11-25
> **Response to Reviewer hRHV (Part 3)**
>
> 5. The reviewer raised concerns regarding the evaluation of our model's performance, specifically questioning the accuracy of the band structure predictions. **However, we believe there has been some misunderstandings**. In addition to the R-space MAE, we have provided extensive band structure predictions in Figures 9 and 10, as well as optical conductivity results in the revised paper (Section 4.3). These results clearly demonstrate the effectiveness and reliability of our method. **The reviewer also mentioned that our band structure prediction errors seem "fairly high," but this is not substantiated by experimental evidence.** The band structure predictions from the Full Method, shown in the right-most column of Figures 9 and 10, exactly match the ground-truth DFT results. The larger errors observed in the figures are from the baseline methods or ablation experiments in the left sub-figures, which are results from removing some components of our approach. The Full Method outperforms these ablation variants, which directly supports the effectiveness of the new components we introduced.
>
>     The reviewer also suggested that "a rigid shift from the band structures computed from $ \mathbf{H}^{(0)}$ could recover the learned solution", but this is a misunderstanding. If globally shifting the bands of $ \mathbf{H}^{(0)}$ could solve all the issues, then why have we spent decades developing self-consistent iterative algorithms for DFT? Upon closer inspection of Figure 9, it is clear that the band structures computed with just $ \mathbf{H}^{(0)}$ do not match the true DFT bands in terms of the curvature at several $ \mathbf{k} $-points. These discrepancies are not due to simple offsets and cannot be corrected by a global shift.
>     Moreover, as shown in Figure 9 (i), the band structures computed solely using $ \mathbf{H}^{(0)}$ exhibit significant shifts for some bands, while they match well for others. Such random deviations cannot be resolved by a global shift. The shape differences and the random offsets in certain bands indicate that the errors are not trivial and require the full regression method to resolve.
>
>     **Additionally, the reviewer mentioned that the off-diagonal matrix element errors are "sub-meV", which is another misunderstanding**. In fact, the MAE for these matrix element errors is at the sub-µeV level, as shown in Table 1. **We believe that achieving SOC predictions with an accuracy of  μeV level holds significant value.** Small differences in SOC energy can greatly impact the electronic structures, topological properties, and quantum behaviors of materials. For instance, in recent papers [7], it is noted that in bilayer graphene (BLG), SOC opens a small gap of tens of μeV in the low-energy bands, which plays a crucial role in spin relaxation and decoherence. This example demonstrates that even small SOC energy differences can have notable effects, making μeV precision essential for accurate predictions in these fields. Moreover, our method achieves this high precision while maintaining computational efficiency, representing a significant improvement over traditional approaches. In conclusion, μeV SOC precision is valuable for material design, quantum computing, and topological material research, offering important insights for both theory and experiments.
>
>     [7] Electric-field-independent spin-orbit-coupling gap in h-BN-encapsulated bilayer graphene. Physical Review Applied, 2025.
>
>     The reviewer suggested validating the effect of our method on reducing SCF iteration. However, this is not the primary goal of our approach. Our aim is to completely bypass the SCF iteration process to truly reduce the computational complexity of electronic structure Hamiltonian calculations. In fact, as we have already demonstrated DFT-level accuracy on the majority of our testing samples, these samples no longer require SCF iterations.
>
> 6. **The reviewer mentioned the dataset and the range of elements, with some misunderstandings.** First, our dataset contains more than just structures with 5–20 atoms. As shown in Figure 6, the training, validation, and test sets also include many structures with more than 20 atoms, with approximately 10% of the structures in each set containing over 30 atoms. Second, the reviewer mentioned that some elements have fewer samples, particularly rare gases. This is actually due to the natural rarity of these elements in materials, not an issue with our dataset. Nevertheless, to address this concern, we have rephrased the description of our dataset in the paper from "68 atomic elements" to "more than 60 elements," as this more accurately reflects the dataset.

---

> ### Author Response · Authors · 2025-11-25
> **Response to Reviewer hRHV (Part 4)**
>
> 7. **The reviewer suggested that it would be more useful to focus on a smaller range of elements. We respectfully disagree.**
> In most of the AI fields, such as computer vision, natural language processing, and AI for Science, there are both specialized models and universal models. Both types of models are encouraged in the research community, with an increasing trend toward universal models [8,9] due to their wider applicability and better generalization performance. Universal models offer several advantages: they cover a broader range of objects and scenarios, enabling more flexibility, and do not require re-training from scratch for each specific problem, which can be cost-prohibitive. In most cases, using a universal model directly is sufficient, while for other problems, fine-tuning with small data is typically enough, unlike specialized models which need to be trained from scratch every time.
>
>     For example, in cases where certain elements are rare in nature, it would be hard to train specialized models for those elements due to the lack of data. This is where universal models shine, as they are well-versed in a wide variety of input systems and can make analogies across different problems, leveraging the few-shot or zero-shot learning capabilities to generalize to unseen elements. As shown in **Figure 7** of the paper, our model is trained on a training set that does not include structures containing the element Neon (Ne). However, in the testing set, a Ne-containing structure is included. Despite Ne being unseen during training, the model is able to generalize well to this new element, with testing error remaining very small, i.e., 0.1 meV for the R-space MAE, as reported in **Figure 4**. This demonstrates the model's ability to extrapolate the knowledge it learned from seen elements to predict properties for unseen elements, showcasing its ability to generalize beyond the training data.. **This is the type of transfer learning capability that only a universal model, with a broad exposure to various objects, can possess.  Our universal model, with its extensive exposure to different elements, combined with advanced descriptors and network structures, equips it with this capability.**
>
>     **Our target is that for most materials, our universal model can be directly applied to predict Hamiltonian without further training, minimizing both usage costs and computational inefficiency. For those materials with more extreme accuracy requirements, we can fine-tune the pre-trained universal model on a small dataset, thus avoiding the need to build extensive datasets for every new material. This approach is both efficient and scalable, especially when dealing with materials that have limited known structures. Ultimately, a reliable universal model serves as the foundation for this streamlined and adaptable approach.**
>
>     [8] Segment Anything. ICCV 2023.
>
>     [9] Language Models are Few-Shot Learners. NeurIPS 2020.
>
> 8. The computational complexity of this loss function is $ O(N^3) $, where $ N $ is the number of atoms in the system. However, **during the inference phase, there is no need to construct the loss function anymore. At this stage, the asymptotic computational complexity of our model is the same as that of existing deep learning Hamiltonian prediction models: $ O(N^2) $ for small systems and $ O(N) $ for large systems.**

---

> > ### Comment · Reviewer_hRHV · 2025-11-25
> >
> > 1. My question was more straightforward: what if you simply scaled the data? I cited similar papers that did this successfully, and it’s true that they are very recent. But that does not change that, if it works, many of the architectural choices made here could be unnecessary.
> >
> > 2. It is not clear (or ablated) what specific architectural innovations allow NextHAM to be 'universal', which could not be achieved with DeephE3.
> >
> > 3. The stated number of elements in the dataset has been reduced from 68 to ‘more than 60’. I did not find an explanation of which elements are included in this 60, or numerical details behind the sampling over remaining elements. Universal datasets are not unimportant - but for a dataset to meet the high standard of ‘universal’, it should properly sample over elements and contain sufficient amounts of data. If the size of the dataset is limited, it may be better to restrict to a smaller set of elements where sufficient sampling can be achieved. 'Generalizability to unseen elements' is also quite a strong claim, and needs more thorough experiments to quantify, beyond a single example number.
> >
> > 4. I acknowledge the authors’ point about complex features in the band structures. However, there still remains no comparison with other models or use of previous benchmarks to understand the model's performance in context. I understand this paper is in the applications track, but it remains an ML conference.
> >
> > Despite the strong claims made in this work and the author’s response, I hesitate to say that it meets the standards of ICLR. I would maintain my score.

---

> > > ### Author Response · Authors · 2025-11-30
> > > **1. A Deep Discussion on Data Scaling**
> > >
> > > The reviewer mentioned that data scaling could be used to remove the trace, and that such normalization might help improve the network's generalization performance. We would like to provide a detailed analysis of this argument.
> > >
> > > In AI regression tasks, scaling is a common choice. Scaling the features to make their distribution approach a standard normal distribution is widely recognized as a good practice for training, convergence, and generalization. We have implemented this through **E(3)-symmetry Layer normalization** at the feature layer, as we descriped in Section 2.2 of our paper.
> > >
> > > The key question raised by the reviewer is whether to apply scaling to the regression target at the network's output layer. Scaling the output could reduce the numerical range of the network's predictions, which the reviewer suggests would lower the regression complexity and potentially enhance fitting and generalization. Reference [1] cited by the reviewer provides a formula (Eq. 16) for scaling the regression target using its standard deviation .
> > >
> > > However, we still prefer our approach, which applies scaling operations at each internal feature layer within the network, rather than scaling values at the final output layer. There are two reasons for this:
> > >
> > > First, scaling the regression target at the output layer may lead to instability in the prediction error. Scaling the regression target essentially involves dividing the target (e.g., Hamiltonian $\mathbf{H}^{(T)}$) by a factor such as the standard deviation $\sigma$, resulting in a scaled version $H' = \frac{\mathbf{H}^{(T)}}{\sigma}$. The advantage of this approach is that it reduces the numerical range, which may make it easier for the network to learn. However, a key issue arises when transforming the predictions back to the original space. Any small error in predicting $H'$ would be amplified when multiplied by $\sigma$ to reverse the scaling, leading to significant errors in the original Hamiltonian space. This amplification of error can compromise the accuracy of the final output. We have observed in our experiments that introducing such scaling factors at the output layer result in a **12\% decrease in R-space prediction accuracy** on our dataset.
> > >
> > > In contrast, no scaling at the output layer has some advantages. Larger values in Hamiltonians are often critical for downstream tasks, such as predicting band structures or other material properties. Being aware of their magnitude and producing the corresponding supervisory signal allows the network to focus on the most important matrix elements. Moreover, the trace quantities $\mathbf{T} = \mathrm{tr}( \mathbf{H} \cdot \ \mathbf{H}^\dagger)$ or $\mathrm{tr}( \Delta \mathbf{H} \cdot \Delta \mathbf{H}^\dagger)$ constructed from the Hamiltonian quantify the total coupling strength encoded in the Hamiltonian matrix, representing the overall amplitude of electronic interactions in the system. Trace variables are a highly effective supervisory signal, as they can supervise high-quality non-linear invariant features, which in turn induce non-linear equivariant features, increasing the non-linear expressiveness of the regression network for predicting the system's Hamiltonian, as demonstrated by the TraceGrad paper [2].  When using trace-removal methods to eliminate the magnitudes or traces of the Hamiltonian, this important feature may be lost, potentially harming the model's performance.
> > >
> > > Furthermore, our approach leverages a more advanced **delta-learning** paradigm, where we predict $\Delta H = \mathbf{H}^{(T)} - \mathbf{H}^{(0)}$ rather than directly predicting $\mathbf{H}^{(T)}$. This dramatically reduces the regression target space without introducing scaling factors, allowing the model to focus on the essential differences between the zeroth-step Hamiltonian and the self-consistent Hamiltonian. This method greatly diminishes the complexity of the regression task while avoiding the potential pitfalls of scaling, and ensures that the model maintains high accuracy while preserving key physical information.
> > >
> > >     [1] Equivariant Electronic Hamiltonian Prediction with Many-Body Message Passing, https://arxiv.org/html/2508.15108v1.
> > >     [2] Tracegrad: a Framework Learning Expressive SO(3)-equivariant Non-linear Representations for Electronic-Structure Hamiltonian Prediction. ICML 2025

---

> ### Author Response · Authors · 2025-11-30
> **2. Why is DeepH-E3 unsuitable for universal Hamiltonian prediction scenarios, while our method excels?**
>
> First, **DeepH-E3** uses randomly initialized element (node) and element pair (edge) embeddings trained from scratch as input descriptors However, the sparse nature of these embeddings poses a problem. Since elements are not uniformly distributed in nature, many element pairs appear infrequently, resulting in poorly trained embeddings that struggle with generalization. What's more, this leads to the need to maintain an embedding cost of $\mathcal{O}(M)$ and $\mathcal{O}(M^2)$ for nodes and edges, respectively, where $\mathcal{O}(M)$ is the number of elements included in the dataset. As the number of elements increases, this cost grows rapidly, leading to significant memory overhead. In fact, in the official code of DeepH-E3, they explicitly point out that these input descriptors "might not be good if the dataset has many elements" (https://github.com/Xiaoxun-Gong/DeepH-E3/blob/main/deephe3/model.py, line 396).
>
> In contrast, our approach addresses these issues by replacing the node and edge embeddings with the zeroth-step Hamiltonian **$\mathbf{H}^{(0)}$**, which eliminates the space complexity and sparsity problems.  **$\mathbf{H}^{(0)}$** captures crucial information about the system's electronic structure, embedding the fundamental characteristics of different elements into a unified representation space, thereby offering a richer physical context. This enables robust generalization across chemically complex material systems, making our approach better suited for constructing a truly universal model, capable of handling a wide range of materials and elements. What's more, the space complexity of our input descriptor is independent of the number of distinct elements in the dataset, thus avoiding the exponential cost of embeddings.  **This use of an easily accessible DFT initial state tensor as a descriptor highlights an emerging direction in AI for Science: embedding physically grounded information directly into machine learning models, which enhances predictive power and provides a general principle that can be broadly applied across scientific ML tasks beyond electronic-structure prediction.**
>
> Second, the architecture of DeepH-E3 is relatively simple and lacks sufficient expressive power for building au universal model, which is reflected in two aspects. First, it uses a non-attentive structure, with feature fusion performed through simple average pooling (https://github.com/Xiaoxun-Gong/DeepH-E3/blob/main/deephe3/model.py, line 202-204). This may not be well-suited to handle highly complex, diverse structures. Second, as analyzed in paper [3], it employs a gating mechanism as the non-linear activation function, which has limited non-linear expressiveness. In contrast, our approach builds upon the TraceGrad paradigm with advanced framework to effectively induce non-linearity from invariant quantities and representations, and develops an E(3)-symmetry Transformer architecture with high non-linear expressiveness. This architecture is capable of dynamically weighting features to adapt to different elements and geometric configurations, making it more suitable for building universal large-scale models.
>
> In addition, our network adopts a delta-learning approach, predicting $\Delta H = \mathbf{H}^{(T)} - \mathbf{H}^{(0)}$ instead of the Hamiltonian itself. This significantly reduces the complexity of the regression target, making it easier for the network to finely fit and generalize complex systems, especially those with heavy atoms and a large number of orbitals. In contrast, DeepH-E3 directly predicts the entire Hamiltonian, which becomes challenging when dealing with more complex scenarios.
>
>
> Third, DeepH-E3 only uses the R-space loss function for training, whilewe propose a joint loss function combining both R-space and k-space. As shown in Section 4.3 and Appendix K, the inclusion of k-space loss improves the physical reliability of downstream quantities, such as band structure, and prevents ghost states, which appear as abrupt discontinuities at isolated $\mathbf{k}$-points, caused by the amplification of small R-space errors in k-space (Appendix E). By addressing these issues, our method is better suited for handling a wide range of materials and elements, making it more adaptable and reliable across various scientific applications.
>
> We hope that the expanded explanation, together with the quantitative evidence presented in Appendix L, clarifies why NextHAM requires its specific design and why simpler architectures cannot achieve comparable universality.

---

> ### Author Response · Authors · 2025-11-30
> **3. Responses to Comments on the Constructed Dataset**
>
> We thank the reviewer for the constructive comments.  Regarding element coverage, our sampling was based on the open Materials Project database, with the following filtering criteria: we exclude lanthanides and elements with atomic number \(Z>83\), as these species typically involve strongly correlated valence electrons, appear rarely in standard DFT calculations, and require specialized pseudopotentials, making large-scale SOC-inclusive DFT calculations substantially more complex; (ii) for computational feasibility, we also exclude structures with more than 80 atoms. After this filtering, we retain essentially all elements from the first six rows of the periodic table that commonly appear in solid-state DFT studies. From the remaining pool, we randomly sample roughly 20\% of structures for training, validation, and testing. As quantified in Appendix G (Dataset Details), the distributions of atomic numbers, stoichiometries, atom counts, and element frequencies closely follow those of the full Materials Project, ensuring an approximately independent-and-identically-distributed subsample.
> Detailed information on pseudopotentials, orbital configurations, and the full element statistics of the splits is provided there.
>
> Concerning dataset scale, our dataset is already significantly larger than existing open-source Hamiltonian-learning datasets (such as those from the DeepH series), especially with SOC included. Expanding to an even broader set of elements would require a substantial increase in DFT computations, which is beyond the scope of this initial release. We also note that the current set of 60+ elements already covers a large portion of the chemistries commonly explored in solid-state DFT studies, and thus provides a reasonable foundation for training a broadly applicable Hamiltonian model. Nevertheless, we agree that further extension is valuable, and we plan to enlarge both the element coverage and the number of structures in future versions.
>
> Finally, the reviewer raised the point that " 'Generalizability to unseen elements' is a strong claim". In response, in Section 4.2 of our paper, we emphasize that this is a case study demonstrating the theoretical potential of our method's generalizability to unseen elements. In future work, we will conduct further element-level out-of-distribution evaluations to more rigorously quantify this ability.

---

> ### Author Response · Authors · 2025-11-30
> **4. Quantitive Comparison with Related Work (Part1)**
>
> The reviewer suggests that we perform a quantitative comparison with DeepH-E3. In fact, we have already conducted experiments with DeepH-E3. Yet, due to the simplicity of the DeepH-E3 model, it is clear that it is not suitable for universal Hamiltonian prediction tasks. This limitation is apparent to us, and since our paper is already of considerable length, we initially chose not to include these results and instead made brief qualitative discussions.
>
> In response to the reviewer’s request, we present the quantitative results below and have added this comparison to Appendix L of the paper.
>
> First, we apply DeepH-E3 to our Materials-HAM-SOC dataset, and follow the same training-validation-testing pipeline on our dataset as described in Appendix H. In the features near the output layer and the output itself, we construct tensor representations that correspond to our dataset's atomic orbital basis sets, which extend to 4s2p2d1f, while leaving the rest of the DeepH-E3 unchanged. These modifications make the comparison more relevant and fair. Unfortunately, **even with these adjustments, DeepH-E3 achieves a high R-space error of $12.605 meV$ on our testing set, failing to converge to a reasonable error range. In contrast, our method achieves a low error of $1.417 meV$**. This result confirms that **DeepH-E3** is not suited for the task of universal electronic structure Hamiltonian prediction, as we have previously analyzed. The research in the DeepH-E3 paper  focuses on specialized scenarios, where the training and testing sets consist of perturbations from molecular dynamics simulations of the same material. In contrast, our task involves predicting the Hamiltonian across a broad range of materials, which requires a more generalizable approach that DeepH-E3's specialized design cannot handle effectively.

---

> ### Author Response · Authors · 2025-11-30
> **4. Quantitive Comparison with Related Work (Part2)**
>
> Second, we compare the performance in specialized scenarios. Specifically, we conduct experiments on the **Monolayer Graphene (MG)** and **Monolayer MoS$_2$ (MM)** datasets, which were released by the **DeepH** series [1,2]. Their statistical information is shown in the table below:
>
> **Table: Overview of the Monolayer Graphene (MG) and Monolayer MoS$_2$ (MM) datasets. $m$: number of samples in the current dataset; $a$: number of atoms per unit cell in the current dataset.**
> | Statistic Types | MG    | MM    |
> |-----------------|-------|-------|
> | **Training**    |       |       |
> | $m$             | 270   | 300   |
> | $a$             | 72    | 75    |
> | **Validation**  |       |       |
> | $m$             | 90    | 100   |
> | $a$             | 72    | 75    |
> | **Testing**     |       |       |
> | $m$             | 90    | 100   |
> | $a$             | 72    | 75    |
>
>
> For both datasets, we use the same training, validation, and testing sets as those used in DeepH-E3. To ensure fairness, we retrain our method under identical conditions, without pre-training, and measure errors using the classic MAE (Mean Absolute Error) metric, in alignment with DeepH-E3.  In addition to DeepH-E3, we also compare our method with the original TraceGrad work [3], which extends the DeepH-E3 backbone network by constructing non-linear equivariant representations from invariant ones.
>
> The version of our method used in these experiments is a cut-down version, denoted as NextHAM$^{\text{cut-down}}$. Since the datasets do not provide the $\mathbf{H}^{(0)}$ (zeroth-step Hamiltonian) label, we modify NextHAM by removing the components related to $\mathbf{H}^{(0)}$. Specifically, we do not use $\mathbf{H}^{(0)}$ as an input descriptor, instead, we use randomly initialized node and edge embeddings, which are trained jointly with the network, similar to those used in DeepH-E3, to represent the elements and their pairwise interactions. The output directly predicts $\mathbf{H}^{(T)}$ rather than $\Delta H = \mathbf{H}^{(T)} - \mathbf{H}^{(0)}$, and because the datasets do not include wavefunction-related data, we retain only the R-space loss function during training, omitting the k-space loss function. What remains is the E(3)-equivariant graph Transformer network architecture developed upon the TraceGrad paradigm.
>
> The comparisons are presented in the following Table, from which the results of DeepH-E3 and the original TraceGrad work are from [3].
>
> **Table: Comparison of MAE values between DeepH-E3, the original TraceGrad work, and NextHAM$^{\text{cut-down}}$. All values are in meV.**
> | **Dataset**            | **DeepH-E3**  | **Original TraceGrad** | **NextHAM$^{\text{cut-down}}$**  |
> |------------------------|--------------------|----------------------------------------|-----------------------------|
> | **MG** | 0.251              | 0.175                                  | **0.102**                   |
> | **MM**     | 0.406              | 0.285                                  | **0.163**                   |
>
> These results show that our method, even after cutting out some modules, still dramatically outperforms both DeepH-E3 and the original TraceGrad work in prediction accuracy for these material systems. While the original TraceGrad work explores effective methods for constructing non-linear equivariant features, it remains a simple graph network based on average pooling. In contrast, our approach extends the TraceGrad paradigm as a more advanced E(3)-symmetry Transformer architecture that can dynamically weight and adapt non-linear equivariant features, which is more expressive than the simple graph network.  **In addition to demonstrating strong generalization across a variety of materials in previous sections, these results  show that our method excels in specialized systems as well, making it a powerful tool for both broad and focused applications in computational materials science.**
>
>     [1] Deep-learning density functional theory Hamiltonian for efficient ab initio electronic-structure calculation. Nature Computational Science, 2022.
>     [2] General framework for E(3)-equivariant neural network representation of density functional theory Hamiltonian. Nature Communications, 2023.
>     [3] Tracegrad: a Framework Learning Expressive SO(3)-equivariant Non-linear Representations for Electronic-Structure Hamiltonian Prediction. ICML 2025

---

### Official Review · Reviewer_Gv72 · 2025-10-30

**Soundness:** 3
**Presentation:** 2
**Contribution:** 2
**Rating:** 4
**Confidence:** 4

**Summary:**

This paper proposes NextHAM, a neural E(3)-equivariant model for predicting materials' electronic structure Hamiltonians. The method incorporates zeroth-step Hamiltonians from the initial DFT charge density as both informative input descriptors and initial estimates for the target. The architecture employs a Transformer designed for strict E(3)-equivariance and high non-linear expressiveness. A novel training objective combining real and reciprocal space losses is introduced to mitigate error amplification and "ghost states" caused by the large condition number of the overlap matrix. Evaluated on a newly curated benchmark, Materials-HAM-SOC, NextHAM demonstrates excellent accuracy in predicting Hamiltonians and band structures, with spin-off-diagonal blocks reaching sub-μeV scale.

**Strengths:**

* The proposed NextHAM method is technically sound and demonstrates state-of-the-art performance on the newly introduced benchmark.
* The release of the Materials-HAM-SOC dataset is a valuable contribution that will facilitate future research in machine learning for materials science.

**Weaknesses:**

* Overall, the technical novelty is limited since the core technical components build heavily upon established concepts:

 * The delta-learning strategy (predicting residuals to a zeroth-order Hamiltonian) is a well-established paradigm in ML-based Hamiltonian and potential energy surface prediction [1, 2].

  * The TraceGrad branch for non-linear expressiveness is a direct adaptation of a prior architecture [3] to this task.

  * The primary novel contribution appears to be the reciprocal space ($k$-space) loss. However, its performance gain over the strong baseline (which already uses the zeroth-step Hamiltonian as an output estimate) seems marginal (Table 4). Furthermore, the authors do not provide sufficient empirical evidence or analysis to demonstrate how this loss concretely alleviates the "ghost states" issue.

* The paper lacks comparisons with several relevant recent methods in the field (e.g., those methods for material systems mentioned in section 2), making it difficult to assess NextHAM's relative advancement. A well-defined, chemically meaningful accuracy threshold is also absent. While the reported metrics (e.g., sub-μeV) are low, it remains unclear whether this precision is sufficient for reliable downstream property predictions (e.g., band gaps, densities of states) in practical computational materials science.


[1] Δ-Machine Learned Potential Energy Surfaces and Force Fields. JCTC 2022

[2] Efficient and Scalable Density Functional Theory Hamiltonian Prediction through Adaptive Sparsity. ICML 2025

[3] Tracegrad: a Framework Learning Expressive SO(3)-equivariant Non-linear Representations for Electronic-Structure Hamiltonian Prediction. ICML 2025

**Questions:**

* Could the authors provide more details on how the sub-matrices of the zeroth-step Hamiltonian are transformed into initial node and edge descriptors? The feature engineering process is crucial for reproducibility.

* The term "universal" in the title is a strong claim, implying robust generalization across diverse chemical and conformational spaces. However, the current experiments primarily evaluate on a static benchmark. Could the authors provide additional evidence, such as out-of-distribution generalization tests, to substantiate this claim?

**Details Of Ethics Concerns:**

No ethics concerns identified.

---

> ### Author Response · Authors · 2025-11-25
> **Response to Reviewer Gv72 (Part 1)**
>
> We would like to sincerely thank the reviewer for their feedback. We greatly appreciate the time and effort spent on evaluating our work.
>
> 1. We appreciate the reviewer’s comment regarding the existing delta-learning approaches using initial DFT guess. However, their methodologies, tasks, and findings differ from ours.
>
>    **First, from a methodological perspective, these existing works [1,2] use the initial guess of DFT solely as a starting point for the output. In contrast, our approach introduces a significant innovation by using the zeroth-step Hamiltonian not only as an output starting point but also as an input descriptor**. Unlike existing methods that rely on randomly initialized atom and edge embeddings lacking physical prior knowledge, $ \mathbf{H}^{(0)} $ efficiently encodes essential information about the system's electronic structure, providing richer physical context by embedding the intrinsic characteristics of diverse elements into a unified representation space, thereby enabling robust generalization across chemically complex material systems. Its effectiveness has been fully validated in our paper, including significant reductions in test errors and the ability to generalize to out-of-distribution elements. This is demonstrated in Section 4.2 and Appendix K.
>
>    Second, regarding the task itself, the focus of their work differs from ours. For instance, the first work [1] applies delta-learning to molecular systems to refine potential energy surfaces and force fields, using traditional quantum chemistry methods (such as DFT) as the initial guess. The second work [2] focuses on electronic structure prediction for molecular systems without considering the SOC (Spin-Orbit Coupling) effect. The challenge addressed in this work are specific to molecular systems and do not directly apply to the periodic nature of materials with SOC effect. In contrast, our work is dedicated to the electronic structure prediction for periodic material systems with explicit modeling of the SOC effect. Nevertheless, prompted by the reviewer’s suggestion, we have explicitly mentioned in a prominent section of the paper that our use of the delta-learning strategy is inspired by previous works [1, 2].
>
>    In our task, **we have made a significant new discovery**. Specifically, $ \mathbf{H}^{(0)} $ and $ \mathbf{H}^{(T)} $ are remarkably close in the spin-off-diagonal block of SOC Hamiltonian blocks of materials. For this part of the block, directly using the matrix elements of $ \mathbf{H}^{(0)} $ achieves accuracy at the sub-µeV level. **This discovery, which is a very interesting finding, has not been reported or observed in prior studies.**
>
>    [1] Δ-Machine Learned Potential Energy Surfaces and Force Fields. JCTC 2022. \
>    [2] Efficient and Scalable Density Functional Theory Hamiltonian Prediction through Adaptive Sparsity. ICML 2025.
>
> 2. We appreciate the reviewer’s comment regarding the TraceGrad mechanism. It is important to clarify that the original TraceGrad work [3] is not an innovation of in this paper, and we have indeed referenced it appropriately. One of our key contributions lies in the development of a highly expressive E(3)-symmetric Transformer architecture, which extends the original TraceGrad work [3] that employed simpler graph networks (Non-Attentive). Additionally, ablation studies in Appendix K have demonstrated that TraceGrad remains effective within our new framework, confirming its continued relevance and usefulness in this context.
>
>     [3] Tracegrad: a Framework Learning Expressive SO(3)-equivariant Non-linear Representations for Electronic-Structure Hamiltonian Prediction. ICML 2025.

---

> ### Author Response · Authors · 2025-11-25
> **Response to Reviewer Gv72 (Part 2)**
>
> 3. The reviewer points out the modest reduction in R-space MAE after introducing the k-space loss, as shown in Table 4. We would like to clarify that the primary purpose of the k-space loss is not to significantly reduce R-space MAE, but to improve the physical reliability of downstream quantities, such as the band structure, and eliminate ghost states.
>
>     **As shown in the case study in Section 4.3, the reduction in R-space MAE after adding the k-space loss is relatively small, from 0.53 meV to 0.49 meV. However, when the k-space loss is removed (i.e., Ablation@Loss-k), ghost states appear in the predicted band structure**. These ghost states manifest as abrupt discontinuities at isolated $\mathbf{k}$-points. Although the majority of $\mathbf{k}$-points are reasonably well-predicted, these isolated points show significant deviations from the true band structure due to the amplification of small R-space errors in momentum space, as mathematically explained in Appendix E. Without the k-space loss, these unphysical anomalies become more pronounced, leading to ghost states. In contrast, the band structure and the derived optical conductivity predicted with the Full Method (which includes the k-space loss) shows a significant improvement in accuracy compared to the ground-truth DFT results. This demonstrates that the k-space loss enhances the physical reliability of the band structure by addressing ghost states, where small errors in the R-space Hamiltonian lead to large deviations in band energies, especially at isolated $\mathbf{k}$-points.
>
>     In summary, the inclusion of the k-space loss significantly mitigates ghost states by addressing the amplification of errors in momentum space. This is consistent with the mathematical explanation in Appendix E and the comprehensive ablation study results shown in Fig. 10, where removing the k-space loss leads to the appearance of ghost states.
>
> 4. We appreciate the reviewer’s comment regarding the need for quantitative comparisons with other recent methods in the field. However, we would like to emphasize that many of the methods referenced in the Related Work Section (Appendix C in the revised paper), such as DeepH-E3, are designed for specific systems and lack generalization across diverse material systems. These approaches typically focus on a narrow range of materials or molecular systems. They develop specialized models, whereas we are developing a universal model, which represents a different technical approach. **To the best of our knowledge, there are only very few methods in the literature that target on universal electronic-structure Hamiltonian prediction for material systems. Although the DeepH-2 method [4] is a representative effort for this task, it does not provide open-source code, data, or pre-trained models, making it challenging for the research community to replicate or compare against.**
>
>     Our work offers a significant advancement in the field by providing a universal framework for electronic structure Hamiltonian prediction that is applicable to a wide variety of material systems. What's more, **we have made our data, code, and pretrained models publicly available, creating a reproducible and comparable benchmark for future work. This openness allows the broader community to validate, build upon, and compare with our work, fostering further advancements in the field of generalized electronic structure prediction. This is one of the key reasons we believe publishing our paper is crucial for the community.**
>
>     [4] Universal materials model of deep-learning density functional theory Hamiltonian. Science Bulletin, 2024.
>
>
> 5. **We believe that achieving SOC predictions with an accuracy of μeV level holds significant value.** Small differences in SOC energy can greatly impact the electronic structures, topological properties, and quantum behaviors of materials. For instance, in recent papers [5], it is noted that in bilayer graphene (BLG), SOC opens a small gap of tens of μeV in the low-energy bands, which plays a crucial role in spin relaxation and decoherence. This example demonstrates that even small SOC energy differences can have notable effects, making μeV precision essential for accurate predictions in these fields. Moreover, our method achieves this high precision while maintaining computational efficiency, representing a significant improvement over traditional approaches. In addition to the performance demonstration in predicting band structures (Appendix K), in Section 4.3 of the revised paper, we show that the proposed method also brings effective predictions of other downstream physical quantities, such as optical conductivity. **These achievements further showcase the effectiveness of our proposed method in practical computational materials science.**
>
>     [5] Electric-field-independent spin-orbit-coupling gap in h-BN-encapsulated bilayer graphene. Physical Review Applied, 2025.

---

> ### Author Response · Authors · 2025-11-25
> **Response to Reviewer Gv72 (Part 3)**
>
> 6. We have added details on how the sub-matrices of the zeroth-step Hamiltonian are transformed into initial node and edge descriptors in Section 2.1.1 and 2.1.2 of the revised paper.
>
> 7. We appreciate the reviewer's comment regarding the evaluation of out-of-distribution generalization. We have performed out-of-distribution tests in our experiments. Specifically, as shown in Figure 7 of the paper, our model is trained on a dataset that does not include structures containing the element Neon (Ne). However, in the testing set, a Ne-containing structure is included. Despite Ne being unseen during training, the model is able to generalize well to this new element, with testing error remaining very small, i.e., 0.1 meV for the R-space MAE, as reported in Figure 4. **This demonstrates the model's ability to extrapolate the knowledge it learned from seen elements to predict properties for unseen elements, showcasing its ability to generalize beyond the training data..**
>
>     **This out-of-distribution generalization capability is largely attributed to the use of the zeroth-step Hamiltonian as an input descriptor.** Unlike traditional methods that rely on randomly initialized embeddings for each element, which cannot generalize to unseen elements not included in the training set, our model integrates physical information about the system's electronic structure into a unified representation space using the zeroth-step Hamiltonian. **This approach enables the model to capture the relationships between different elements' electronic structures, allowing it to generalize more effectively to out-of-distribution elements**, such as Ne.

---

> > ### Comment · Reviewer_Gv72 · 2025-11-25
> >
> > I appreciate the authors' responses, which have clarified several points.
> >
> > The primary contribution of this work lies in its empirical value: providing a reproducible benchmark and a general-purpose pretrained model for universal electronic structure prediction, alongside demonstrating the importance of SOC modeling. While these are valuable to the community, the work aligns more closely with the aims of specialized scientific journals or dataset/benchmarking platforms, which are better suited for such contributions, rather than the core machine learning technique focus of ICLR. Thus, I maintain my original score.

---

> ### Author Response · Authors · 2025-11-26
>
> 8. Thank you very much for your comments. We believe our paper is highly relevant to the "Applications to Physical Sciences" track at ICLR, as it introduces a novel deep learning approach to predicting universal electronic structure Hamiltonians, offering a significant advancement in electronic structure calculations. Our contributions, including a reproducible benchmark and pretrained model, go beyond empirical validation by addressing the critical gap of generalized models for electronic structure prediction across diverse materials.
>
> # Broad Implications of Our Methodology for AI in Science
> A key innovation of our work lies in using $\mathbf{H}^{(0)}$ as an input descriptor. This embeds physical information into the machine learning framework, which is a crucial development in the field of machine learning. Through this approach, we can encode the information of different elements that were unrelated in previous works into a unified latent space, allowing these pieces of information to be interconnected and enhancing the learning process, which yields excellent results. We provide a preliminary example showing that an element absent from the training set can still achieve reasonable performance in the test set. In contrast, **the current research typically relies on randomly initialized element (node) and element pair (edge) embeddings as input descriptors for tasks like Hamiltonian prediction, energy, and force field predictions**. However, the sparse nature of these embeddings poses a problem. Since elements are not uniformly distributed in nature, many element pairs appear infrequently, resulting in poorly trained embeddings that struggle with generalization. What's more, this leads to the need to maintain an embedding cost of **O(M)** and **O(M²)** for nodes and edges, respectively, where **M** is the number of elements included in the dataset. As the number of elements increases, this cost grows rapidly, leading to significant memory overhead. In contrast, **our approach addresses these issues by replacing the embeddings with the zeroth-step Hamiltonian $\mathbf{H}^{(0)}$, which eliminates the sparsity and space complexity problems. This use of an easily accessible DFT initial state tensor as a descriptor highlights an emerging direction in AI for Science: embedding physically grounded information directly into machine learning models, which enhances predictive power and provides a general principle that can be broadly applied across scientific ML tasks beyond electronic structure prediction.**
>
> Finally, we would like to note that the works cited by the reviewer (e.g., [1–2]) and some other works [3-5] share a research style similar to ours—tackling significant challenges in the physical sciences through a blend of machine learning methodology and application-driven insights. These studies have been well received at top-tier machine learning conferences. Similarly, our work offers both substantial application value and methodological contributions, making it well-aligned with the objectives of this ICLR track.
>
> [1] Efficient and Scalable Density Functional Theory Hamiltonian Prediction through Adaptive Sparsity. ICML 2025.
> [2] Tracegrad: a Framework Learning Expressive SO(3)-equivariant Non-linear Representations for Electronic-Structure Hamiltonian Prediction. ICML 2025.
> [3] QH9: A Quantum Hamiltonian Prediction Benchmark for QM9 Molecules. NeurIPS 2023.
> [4] Efficient and Equivariant Graph Networks for Predicting Quantum Hamiltonian. ICML 2023.
> [5] Enhancing the Scalability and Applicability of Kohn-Sham Hamiltonians for Molecular Systems. ICLR 2025.

---

> > ### Comment · Reviewer_Gv72 · 2025-11-28
> >
> > Thank the authors for the further responses, which have resolved several of my concerns. Accordingly, I will be raising the rating to 6.

---

> > > ### Author Response · Authors · 2025-11-28
> > >
> > > I would like to sincerely thank the reviewers for their thorough evaluation and valuable feedback on my paper. I deeply appreciate the time and effort you have invested in reviewing my work!

---

### Official Review · Reviewer_ts7f · 2025-10-31

**Soundness:** 3
**Presentation:** 3
**Contribution:** 3
**Rating:** 6
**Confidence:** 4

**Summary:**

This paper proposes NextHAM, a deep-learning framework for predicting electronic-structure Hamiltonians of materials with near-DFT accuracy.

The method introduces a physically informed prior, the zeroth-step Hamiltonian $H^{(0)}$, derived from the initial electron density (superposition of atomic charge densities). The model learns only the correction $\Delta H = H^{(T)} - H^{(0)}$, effectively simplifying the regression space.

NextHAM uses a Transformer-based E(3)-equivariant architecture extended with the TraceGrad mechanism for non-linear expressiveness under strict SO(3)-equivariance, and jointly optimizes both real-space and reciprocal-space Hamiltonians to reduce “ghost-state” artifacts.
A new dataset, Materials-HAM-SOC, is also released, covering 17 000 crystalline materials across 68 elements, computed with ABACUS + PYATB including explicit spin–orbit coupling.

**Strengths:**

1. Strong Physical–ML Integration: The introduction of the zeroth-step Hamiltonian as both input and output prior is conceptually sound and reduces the learning complexity.
2. Extends TraceGrad to an equivariant Transformer with enhanced edge-centric attention
3. Training target in both real and reciprocal spaces
4. Large-Scale and Diverse Datasets: Materials-HAM-SOC is the first open benchmark with SOC-included Hamiltonians across nearly the full periodic table, offering a valuable resource to the community.

**Weaknesses:**

1. Marginal Conceptual Novelty:
The introduction of the “zeroth-step Hamiltonian” is interesting, but it can be viewed as a variant of the superposition of atomic potentials or the initial DFT guess—concepts that have already appeared in physics-informed ML models. Hence, the main contribution seems to lie in the integration and refinement of known components rather than in introducing a fundamentally new idea.
2. Venue Suitability:
The contribution of this work feels more aligned with a scientific journal in computational materials science than with the core themes of ICLR. The paper emphasizes methodical accuracy, dataset curation, and physical completeness, but it lacks the conceptual abstraction or general ML innovation typically expected at this venue.
3. Limited Evidence for Claimed Efficiency:
The paper claims substantial computational speed-ups, yet it provides no quantitative scaling analyses (e.g., runtime versus system size) or benchmarks against established ML Hamiltonian predictors such as DeepH-2. The reported improvements remain qualitative. I understand that the focus here is on SOC-inclusive Hamiltonians—a domain rarely addressed in prior work—but even so, direct numerical comparisons with existing open-source models would considerably strengthen the experimental section.
4. Unclear Dataset Generation Strategy:
The dataset construction choices appear somewhat arbitrary. For instance, all DFT calculations use a 6×6×6 k-point grid, but no justification is given for this selection. Was this chosen to maintain consistent k-space density, convergence accuracy, or computational feasibility? A systematic benchmark or discussion of how these parameters were determined—either within this work or relative to existing datasets—would improve the scientific rigor.
5. Lack of Physically Meaningful Evaluation:
The reported metrics (e.g., MAE of Hamiltonian matrix elements or visual comparison of band structures) show numerical accuracy but do not necessarily reflect physically interpretable performance. While these are useful from a machine-learning perspective, it would be more convincing to assess the model’s impact on derived physical quantities such as effective masses, band velocities, or even carrier mobilities. These quantities would offer a clearer picture of how well the learned Hamiltonians reproduce observable electronic behavior.

**Questions:**

One point is unclear to me. In NAO-based electronic structure calculations, the reciprocal-space Hamiltonian $H(\mathbf{k})$ is obtained through the Fourier transform of the real-space Hamiltonian $H(\mathbf{R})$. However, the authors mention applying an 8 Å cutoff in their model architecture. This implies that for interatomic distances beyond 8 Å, the predicted matrix elements vanish, i.e., $H(\mathbf{R}) = H^{(0)}(\mathbf{R})$. If so, the Fourier transform would only include short-range interactions, and the long-range components in H(\mathbf{k}) would be incorrect. How, then, does the model reconstruct a physically meaningful $H(\mathbf{k})$ and corresponding band structure? Did I misunderstand the implementation, or is there an additional correction or extrapolation beyond the cutoff?

---

> ### Author Response · Authors · 2025-11-25
> **Response to Reviewer ts7f (Part1)**
>
> We would like to sincerely thank the reviewer for their thoughtful and constructive feedback. We greatly appreciate the time and effort put into evaluating our work. The suggestions and insights provided are extremely valuable and will help us further improve the clarity of our study.
>
> 1. We appreciate the reviewer’s comment about the existing approaches using the initial DFT guess. However, their methodologies, tasks, and findings differ from ours.
>
>    First, from a methodological perspective, existing works [1,2] use the initial guess of DFT solely as a starting point for the output. In contrast, our approach introduces a significant innovation by using the zeroth-step Hamiltonian not only as an output starting point but also as an input descriptor. Unlike existing methods that rely on randomly initialized atom and edge embeddings lacking physical prior knowledge, $ \mathbf{H}^{(0)} $ efficiently encodes essential information about the system's electronic structure, providing richer physical context by embedding the intrinsic characteristics of diverse elements into a unified representation space, thereby enabling robust generalization across chemically complex material systems. $ \mathbf{H}^{(0)} $ as a descriptor significantly reduces prediction errors, and this has been fully validated in the ablation study (Appendix K).
>
>     Moreover, an intriguing aspect of our work is that $ \mathbf{H}^{(0)} $ empowers the model to generalize to elements that were not seen during training, introducing a novel out-of-distribution generalization capability. This is because $ \mathbf{H}^{(0)} $ is a physically grounded operator that encodes cross-element regularities of electronic structure, rather than relying on randomly initialized per-element embeddings. As shown in **Figure 7** of the paper, our model is trained on a training set that does not include structures containing the element Neon (Ne). However, in the test set, a Ne-containing structure is included. Despite Ne being unseen during training, the model is able to generalize well to this new element, with testing error remaining very small, i.e., 0.1 meV for the R-space MAE, as reported in **Figure 4**. This demonstrates the model's ability to extrapolate the knowledge it learned from seen elements to predict properties for unseen elements, showcasing its ability to generalize beyond the training data.. In contrast, traditional methods that rely on randomly initialized embeddings for each element cannot generalize to unseen elements that were not included in the training set.
>
>     Second, regarding the task itself, the focus of their work differs from ours. For instance, the first work [1] applies delta-learning to molecular systems to refine potential energy surfaces and force fields, using traditional quantum chemistry methods (such as DFT) as the initial guess. The second work [2] focuses on electronic structure prediction for molecular systems without considering the SOC (Spin-Orbit Coupling) effect. The challenge addressed in this work are specific to molecular systems and do not directly apply to the periodic nature of materials with SOC effect. In contrast, our work is dedicated to the electronic structure prediction for periodic material systems with explicit modeling of the SOC effect. Nevertheless, prompted by the reviewer’s suggestion, we have explicitly mentioned in the revised paper that our use of the delta-learning strategy is inspired by previous work.
>
>     In our task, **we have made a significant new discovery**. Specifically, $ \mathbf{H}^{(0)} $ and $ \mathbf{H}^{(T)} $ are remarkably close in the spin-off-diagonal block of SOC Hamiltonian blocks of materials. For this part of the block, directly using the matrix elements of $ \mathbf{H}^{(0)} $ achieves accuracy at the sub-µeV level. **This discovery, which is a very interesting finding, has not been reported or observed in prior studies. We believe that achieving SOC predictions with an accuracy of μeV level holds significant value.** Small differences in SOC energy can greatly impact the electronic structures, topological properties, and quantum behaviors of materials. For instance, in recent papers [3], it is noted that in bilayer graphene (BLG), SOC opens a small gap of tens of μeV in the low-energy bands, which plays a crucial role in spin relaxation and decoherence. This example demonstrates that even small SOC energy differences can have notable effects, making μeV precision essential for accurate predictions in these fields.
>
>     [1] Δ-Machine Learned Potential Energy Surfaces and Force Fields. JCTC 2022.
>
>     [2] Efficient and Scalable Density Functional Theory Hamiltonian Prediction through Adaptive Sparsity. ICML 2025.
>
>     [3] Electric-field-independent spin-orbit-coupling gap in h-BN-encapsulated bilayer graphene. Physical Review Applied, 2025.

---

> ### Author Response · Authors · 2025-11-25
> **Response to Reviewer ts7f (Part2)**
>
> 2. In our understanding, the track we submitted to, "Applications to Physical Sciences" at ICLR, is focused on developing and applying machine learning to address important challenges in physical sciences. We believe our work aligns well with this focus. Additionally, our method offers broad implications on AI for Science. The current research typically relies on randomly initialized element (node) and element pair (edge) embeddings as input descriptors for tasks like Hamiltonian prediction, energy, and force field predictions. However, the sparse nature of these embeddings poses a problem. Since elements are not uniformly distributed in nature, many element pairs appear infrequently, resulting in poorly trained embeddings that struggle with generalization. What's more, this leads to the need to maintain an embedding cost of **O(M)** and **O(M²)** for nodes and edges, respectively, where **M** is the number of elements in the dataset. As the number of elements increases, this cost grows rapidly, leading to significant memory overhead. In contrast, **our approach addresses these issues by replacing the embeddings with the zeroth-step Hamiltonian $ \mathbf{H}^{(0)} $, which eliminates the sparsity and space complexity problems. This use of an easily accessible DFT initial state tensor as a descriptor highlights an emerging direction in AI for Science: embedding physically grounded information directly into machine learning models, which enhances predictive power and provides a general principle that can be broadly applied across scientific ML tasks beyond electronic structure prediction.
>
> 3. We greatly appreciate the reviewer’s suggestion regarding the need for quantitative computational efficiency experiments. As detailed in Appendix J, we have already provided comprehensive measurements demonstrating that our method is dramatically faster than DFT methods.
>
>     The reviewer also mentioned comparing against established models such as **DeepH-2**, and we fully agree with this suggestion. However, a major challenge is the limited availability of open-source deep learning models for universal electronic-structure prediction of diverse material systems. **The code and data for DeepH-2 have not been publicly released, preventing direct comparisons.**
>
>     In contrast, one of the key contributions of our work is the establishment of a comprehensive and rich benchmark for universal electronic-structure prediction of materials, which will enable future research to build on and improve it. To facilitate this, we have already open-sourced our full code, pre-trained models, and dataset along with this submission.
>
> 4. We thank the reviewer for this valuable comment. We acknowledge that in the current dataset construction we did not perform systematic convergence tests for parameters such as the 6×6×6 k-point mesh, the charge-density convergence criteria, or the plane-wave energy cutoff, and instead used conservative default values in order to generate a large number of structures efficiently. We agree that this choice was not sufficiently justified in the manuscript. In future updates, we will optimize these DFT parameters according to system sizes and include convergence benchmarks to ensure greater rigor and reproducibility.
>
> 5. We thank the reviewer for this insightful suggestion. We agree that evaluating the predicted Hamiltonians solely through matrix-element MAEs or band-structure comparisons does not fully reflect their physical fidelity. As the reviewer pointed out, the accuracy of derived quantities also depends on the quality of the eigenfunctions. In the revised manuscript, we have added a new subsection (4.3) showing that our mixed real-space and k-space loss not only improves bandstructure agreement but also yields better eigenfunctions, as evidenced by more accurate optical conductivity computed from the predicted Hamiltonians. We believe that, given the simultaneous improvement in both bands and wavefunctions, the Hamiltonians trained with our new loss function are well positioned to achieve reliable performance in predicting other physical properties such as effective masses and carrier dynamics.
>
> 6. We apologize for the confusion caused by the description of the cutoff radius. In our implementation, the model predicts only the Hamiltonian matrix elements within the 8 Å cutoff. For interactions beyond this distance, the Hamiltonian is not set to zero; instead, we reconstruct the long-range part using the zeroth-step Hamiltonian $ \mathbf{H}^{(0)} $ , whose long-range components decay rapidly and closely match the ground-truth behavior. This hybrid scheme greatly reduces the learning burden and computational cost while still preserving the physically meaningful long-range contributions in $H(\mathbf{k})$. As a result, the Fourier-transformed reciprocal-space Hamiltonian remains accurate, and the resulting band structures are consistent with the DFT reference.

---

### Author Response · Authors · 2025-12-02
**Letter to AC: Summary of the Highlights of Our Work**

Dear AC:

Thank you very much for reviewing our paper amidst your busy schedule. We would like to provide a summary of the highlights of our work and our interactions with the reviewers for your reference:

1. **Key innovations of our paper**:
    - First, we design an input structure by introducing the zeroth-step Hamiltonian $\mathbf{H}^{(0)}$ as the **input descriptor**, replacing the traditional randomly initialized node and edge embeddings. Through this approach, we encode information from different elements, previously unrelated in conventional methods, into a unified latent space, allowing these representations to interact and enhancing the overall learning process, which yields excellent results. This physics-grounded descriptor enables robust generalization across chemically complex material systems and demonstrates clear potential for generalization to out-of-distribution elements. To the best of our knowledge, this work is the first to incorporate a physically meaningful initial Hamiltonian as a unified descriptor for  Hamiltonian learning.
    - Second, we find that by introducing $\mathbf{H}^{(0)}$ at the output layer, we can easily achieve accuracy at the sub-µeV level in the spin-off-diagonal blocks of SOC Hamiltonian matrices. This eliminates the need, as in previous works, for the network to specifically predict this part of the matrix (which is also difficult to predict accurately), significantly saving the model capacity.
    - Third, we develop an E(3)-symmetry Transformer framework with high non-linear expressiveness for predicting material Hamiltonians.
    - Last but not least, we incorporate a joint real-space and momentum-space loss that enforces physically consistent band dispersions. This approach greatly suppresses the “ghost states” in the predicted band structures, an issue that has not been addressed in previous works.

  Taken together, these innovations enable our method to achieve excellent accuracy in predicting Hamiltonians and band structures across a rich variety of material systems spanning more than 60 element types, delivering DFT-level precision with dramatically improved computational efficiency, establishing a new state of the art for universal Hamiltonian learning.


2. **Contributions to the Open-Source Community**:  We have made our data, code, and pretrained models publicly available, establishing a reproducible benchmark for future work. This openness enables the broader research community to validate and compare our work, fostering further advancements in the field of universal Hamiltonian prediction of materials. In contrast, existing open-source code and data in this domain are often focused on specialized and narrow chemical domains [1]. Some works targeting more general scenarios (e.g., DeepH-2) do not provide open-source code or data [2], making it challenging for the research community to replicate or compare against. Given the lack of open-source, generalizable models and corresponding benchmarks in this area, we believe our contribution plays a crucial role in advancing the field, offering a valuable resource for continued research.

    [1] General framework for E(3)-equivariant neural network representation of density functional theory Hamiltonian. Nature Communications 2023.

    [2] Universal materials model of deep-learning density functional theory Hamiltonian. Science Bulletin 2024.

3. **Broad Implications of Our Methodology for AI in Science**: Our use of an easily accessible DFT initial state tensor as a descriptor offers many advantages, providing a general principle that can be broadly applied across scientific ML tasks beyond electronic structure prediction. Please refer to our response (Point 8) to Reviewer Gv72 for further details.

---

> ### Author Response · Authors · 2025-12-02
>
> **4. Comparison with Existing Work**: We have engaged in substantive and constructive discussions with the reviewers and have provided detailed clarifications to address their concerns. After that, Reviewers Gv72 and Yiie have clearly indicated their decision to raise their scores to 6. Reviewer hRHV, however, still questions whether the complexity of our model is necessary and suggests that we provide additional comparisons with DeepHE3  [1]  to further demonstrate the advantages of our approach.
>
> As a pioneering effort, DeepH-E3 has played a crucial role in introducing equivariant networks to the calculation of electronic structures of materials. However, we would like to note that our method and DeepH-E3 (and similar recent models) are designed for entirely different levels of physical complexity. Models like DeepH-E3 are designed for specialized and narrow chemical domains, where the element types are limited and the structural motifs are relatively similar. In contrast, our method is explicitly built as a cross–periodic-table Hamiltonian model, capable of handling both light and heavy elements, large variations in structural geometry, and highly diverse bonding chemistries. The difficulty of the prediction problem is significantly higher than in the constrained regimes for which DeepH-E3 was designed. **Experimental results (Appendix L) show that DeepH-E3 performs poorly on our dataset, and even on the specialized datasets it is designed for, our method still dramatically outperforms it.**
>
> Best Regards,
>
> the Authors.

---

### Meta-Review · Area_Chair_HPRx · 2026-01-10

**Summary:**

The authors present a neural network architecture for predicting Hamiltonians, as in density functional theory, that uses an initial "zero-step Hamiltonian" calculated by another method as an initial input. They test this on a novel dataset of calculations that includes spin-orbit coupling and find state-of-the-art results. There is already a large literature of ML methods for accelerating or improving the accuracy of DFT-level theory, so it can be challenging for a particular method to distinguish itself. The novel dataset is valuable in its own right, as spin-orbit coupling is a critical correction for making experimentally accurate predictions, especially for heavier elements, which is often ignored in ML applications. It is a bit more challenging to evaluate the impact of the proposed algorithm, especially since there is only one table of results in the main paper. I would possibly rewrite the paper to focus more on the value of the dataset. For the sake of encouraging the AI-for-science community to focus on more experimentally-relevant benchmarks, I recommend accepting the paper.

**Reviewer Concerns:**

The reviewers had concerns about comparison with other Hamiltonian prediction methods like DeepH-2 and DeepH-E3. The former has not been open-sourced while comparisons against the latter were favorable, and two out of three reviewers who recommended rejection switched their recommendation to acceptance.

**Reviewer Scores:**

Two out of three indicated they would change their score.

---

> ### Public Comment · ~Shi_Yin1 · 2026-03-01
> **Thank you for the acceptance and a note on the methodological evaluations**
>
> Dear Area Chair and Reviewers,
>
> We would like to express our sincere gratitude for your time, constructive feedback, and the final decision to accept our paper.
>
> Regarding the Area Chair's observation that the main text contains only one table of results, we would like to clarify that due to the strict page limit of the ICLR main text, we had to place the comprehensive empirical evaluations of our methodological contributions into the Appendices, presenting only the primary results and conclusions in the main text.
>
> For future readers who are interested in the methodological advancements of NextHAM, we would like to highlight the following sections in the Appendix where the state-of-the-art performance and the effectiveness of our algorithm are thoroughly demonstrated:
>
> * **Appendix J (Band Structure Results):** Extensive visualizations proving that our method achieves high accuracy in the prediction of band structures.
> * **Appendix K (Efficiency Tests):** Rigorous efficiency benchmarks demonstrating the dramatic computational speedups of our method over conventional DFT workflows.
> * **Appendix L (Ablation Studies):** Detailed empirical studies on the contributions of the various components in our framework, proving their effectiveness for achieving accurate Hamiltonian prediction.
> * **Appendix M (Comparison with Related Work):** Experimental results demonstrating that our method dramatically outperforms the pioneering DeepH-E3 work, achieving state-of-the-art results both in the general scenarios evaluated on our diverse dataset, Materials-HAM-SOC, and in specialized scenarios such as Monolayer Graphene and Monolayer MoS$_2$.
>
> Thank you again for your efforts in reviewing our work and for helping us improve this paper.
>
> Best regards,
>
> Authors

---

### Decision · Program_Chairs · 2026-01-26

Accept (Poster)